**Cite this article:** van Bijlert PA, van Soest AJK, Schulp AS. 2021 Natural Frequency Method: estimating the preferred walking speed of *Tyrannosaurus rex* based on tail natural frequency. *R. Soc. Open Sci.* **8**: 201441. https://doi.org/10.1098/rsos.201441

biomechanics/palaeontology/biomechanics

locomotion, cost of transport, theropoda, tetanurae, optimal walking speed

**Author for correspondence:**
Pasha A. van Bijlert
e-mail: pasha.vanbijlert@naturalis.nl

# Natural Frequency Method: estimating the preferred walking speed of *Tyrannosaurus rex* based on tail natural frequency

Pasha A. van Bijlert[1,3], A. J. 'Knoek' van Soest[1,4] and Anne S. Schulp[2,3,5]

[1]Department of Human Movement Sciences, Faculty of Behavioural and Movement Sciences, and [2]Department of Earth Sciences, Faculty of Science, Vrije Universiteit Amsterdam, Amsterdam, The Netherlands
[3]Naturalis Biodiversity Center, Leiden, The Netherlands
[4]Research Institute Amsterdam Movement Sciences, Amsterdam, The Netherlands
[5]Faculty of Geosciences, Utrecht University, Utrecht, The Netherlands

PAvB, 0000-0002-5567-7022; AJKvS, 0000-0002-1959-1061; AS, 0000-0001-9389-1540

Locomotor energetics are an important determinant of an animal's ecological niche. It is commonly assumed that animals minimize locomotor energy expenditure by selecting gait kinematics tuned to the natural frequencies of relevant body parts. We demonstrate that this allows estimation of the preferred step frequency and walking speed of *Tyrannosaurus rex*, using an approach we introduce as the Natural Frequency Method. Although the tail of bipedal dinosaurs was actively involved in walking, it was suspended passively by the caudal interspinous ligaments. These allowed for elastic energy storage, thereby reducing the metabolic cost of transport. In order for elastic energy storage to be high, step and natural frequencies would have to be matched. Using a 3D morphological reconstruction and a spring-suspended biomechanical model, we determined the tail natural frequency of *T. rex* ($0.66\,\mathrm{s}^{-1}$, range 0.41–0.84), and the corresponding walking speed ($1.28\,\mathrm{m\,s}^{-1}$, range 0.80–1.64), which we argue to be a good indicator of preferred walking speed (PWS). The walking speeds found here are lower than earlier estimations for large theropods, but agree quite closely with PWS of a diverse group of extant animals. The results are most sensitive to uncertainties regarding ligament moment arms, vertebral kinematics and ligament composition. However, our model formulation and method for estimation of walking speed are unaffected by assumptions regarding

muscularity, and therefore offer an independent line of evidence within the field of dinosaur locomotion.

## 1. Introduction

Animals display a variety of gaits and walking speeds. Two locomotor extremes are of particular interest: the maximal speed, and the energetically optimal walking speed, at which the metabolic cost of transport (MCOT, in $J\,m^{-1}\,kg^{-1}$) is minimal for walking gait. In the absence of external task constraints, optimal walking speed is very close to preferred walking speed (PWS) in humans [1,2], ratites [3], horses [4] and elephants [5]. It is likely that they are related in all terrestrial animals, and that animals would tend to forage at this speed. During locomotion at preferred speed, animals tend to make use of resonance by matching locomotor frequencies to the (undamped) natural frequencies of their relevant body parts, sometimes tuned through muscular contractions [6–13]. Indeed, it has been demonstrated that mammalian quadrupedal taxa have convergent natural frequencies between fore- and hindlimbs despite differing limb morphologies [14,15]. An important advantage of moving a body part close to its natural frequency is that this reduces mechanical work [16,17]. Based on the commonly adopted idea that PWS is chosen to minimize MCOT [1–5], it has been argued that preferred step frequencies are close to the natural frequencies of relevant body parts [7,11,12]. Determination of the natural frequency of a body part relevant to locomotion can, therefore, constrain preferred (and presumably, optimal) step frequencies and walking speeds of extinct taxa [10].

Theropod dinosaurs had tails that were actively involved in locomotion [18,19], but were passively supported through the caudal interspinous ligaments [20–22], thus forming a mass-spring system. *Tyrannosaurus rex* has been the focus of many locomotor studies [23–32]. Its largest muscle, *M. caudofemoralis longus* (CFL), retracted the femur to produce forward propulsion for locomotion [18,25,33–35] (figure 1a,c). The tail was subject to flexion torque due to gravity and CFL contractions, which was counteracted at zero metabolic cost by the caudal interspinous ligaments, leading to metaplastic adaptations [20–22] (figure 1b). Elastic energy storage in these ligaments improved locomotor efficiency, and like a mass on a spring, the tail would have oscillated at the step frequency. This is conceptually similar to elastic storage in ungulate neck movement [36–38]. The tail would resonate if step frequency was matched to the tail natural frequency, and this would maximize strain and thus elastic storage [39]. The principle of elastic energy storage has previously been used to simulate more realistic running gaits of the smaller theropod *Allosaurus fragilis* [40]. Elastic storage of the caudal interspinous ligaments likely played a smaller role in quadrupedal dinosaurs, due to the less pronounced centre of mass excursions. However, because extant quadrupeds display similar energy-saving mechanisms [36–38], our method should also be applicable to the locomotion of quadrupedal dinosaurs, provided their necks or tails were passively suspended by ligamentous structures.

In recent biomechanical models of non-avian dinosaurs, the focus has been on detailed hindlimb muscular reconstructions, while the tail was simplified to a single rigid structure (for instance: [30,41]). It is our view, however, that tail flexibility is an essential aspect of the locomotion of non-avian dinosaurs. We, therefore, chose to focus on the tail, because investigating the dynamic effects of a compliant tail may provide interesting insights into locomotor capabilities. The primary goal of this study was to develop a reductionist method to estimate PWS for non-avian dinosaurs. To this end, we performed a detailed morphological reconstruction of the tail, including the caudal interspinous ligaments. By incorporating their spring-like properties in a biomechanical model, we subsequently estimated the natural frequency of the vertical swaying of the tail. Using this as an analogue for step frequency, we then combined the tail natural frequency with trackway data to estimate the PWS. We will refer to this approach as the Natural Frequency Method (NFM), and demonstrate its application by determining the PWS for *T. rex*. We have also investigated which morphological features of the tail have the largest impact on tail natural frequency. In doing so, we hope to encourage researchers to incorporate a non-rigid tail into their locomotor simulations, while also demonstrating that the relatively simple NFM can be a valuable expansion of the toolkit of palaeo-biomechanists.

## 2. Material and methods

We estimated tail natural frequency of *T. rex*, by numerically determining the lowest eigenfrequency of a biomechanical model. Essentially, we constructed this model by estimating inertial parameters of the tail,

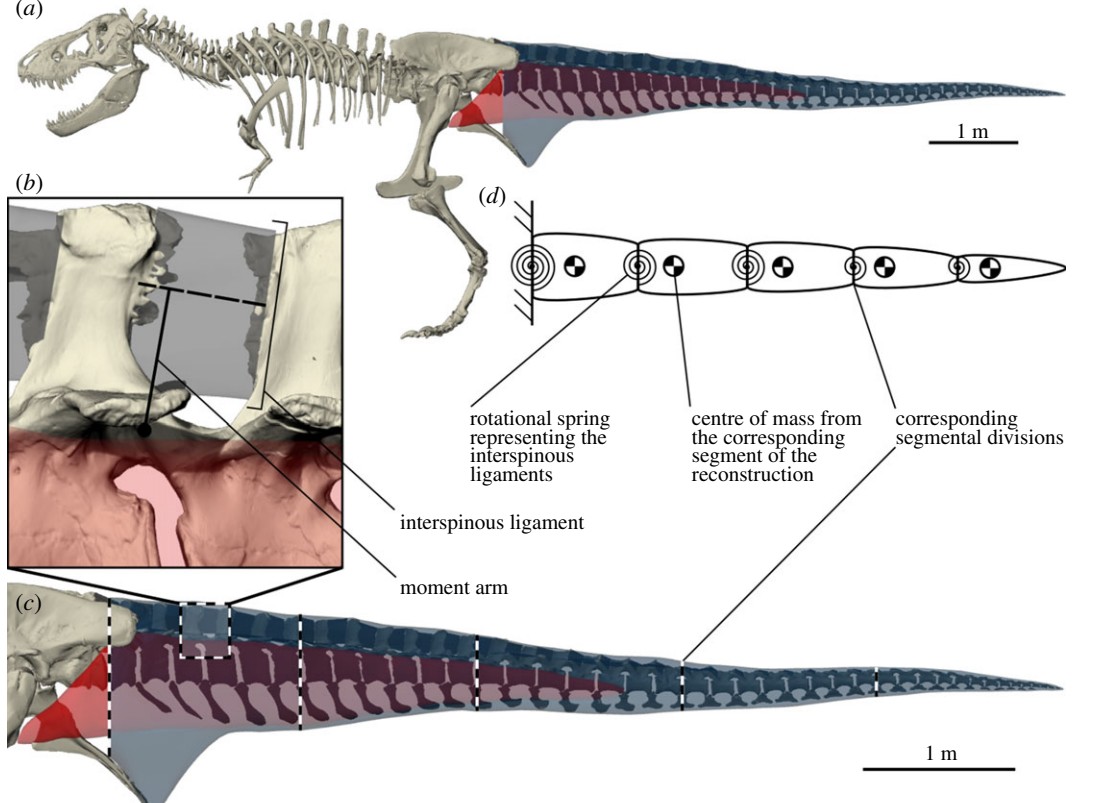

**Figure 1.** Musculoskeletal reconstruction and biomechanical model used to determine the natural frequency of the tail of *Tyrannosaurus rex*. (*a*) Full skeleton of adult specimen RGM.792000, including muscular reconstruction of the tail. *M. caudofemoralis* in red, remaining musculature and soft tissues in blue. (*b*) Reconstruction of the caudal interspinous ligaments (grey). Overlap between the ligament reconstruction and the vertebrae are metaplastic adaptations of the ligament. Our baseline model uses the moment arm from the rotational axis to the dashed line connecting the area centroids of the ligament attachments. Bounds for the moment arms were chosen for the sensitivity analysis. (*c*) Segmental divisions used to partition the tail. Box insert shows (*b*). (*d*) The biomechanical model. Each segment has the length and centre of mass of the corresponding segment in (*c*). Each rotational spring represents the posture-dependent mechanical effect of the interspinous ligaments. The lowest natural frequency (i.e. the frequency corresponding to the fundamental resonant mode) of this system was derived numerically.

dividing it into five segments, and then fitting joint spring parameters based on inverse-dynamic relationships of different postures of the skeleton.

Inertial parameters of this model were based on a 3D volumetric musculoskeletal reconstruction. This reconstruction was performed on adult *T. rex* specimen RGM.792000 (nicknamed 'Trix', figure 1*a*), in the collection of Naturalis Biodiversity Center in Leiden, The Netherlands. This is a large, adult specimen with exceptional surface preservation, making it possible to accurately distinguish the attachment sites of the interspinous ligaments. We articulated 3D scans of the caudal skeleton (figure 1*c*, electronic supplementary material, figure S1), after which we reconstructed the major caudal musculature to estimate inertial parameters (figure 1*c*, electronic supplementary material, figures S2–S4). Using osteological landmarks combined with the physical articulation of 3D prints, we determined the axes of rotation between the vertebrae (electronic supplementary material, figures S5 and S6). Subsequently, we reconstructed the caudal interspinous ligaments (figure 1*b*), which enabled the determination of moment arms and cross-sectional area (figure 1*b*, electronic supplementary material, figure S7). The morphological reconstruction of the ligaments made it possible to quantify the kinematic relation between vertebral flexion/extension and length of the individual ligaments (i.e. strain), which was used to construct the biomechanical model. Parameters of the reconstruction are provided in electronic supplementary material, table S1.

Taking the morphological reconstruction as starting point, we then defined a simplified biomechanical model consisting of five rigid bodies, connected in hinge joints, with a nonlinear rotational spring at each joint (figure 1 and figure 2 for angle definitions). During walking, the most prominent movements and forces occur in the sagittal plane, and the moment arm of the CFL is also the largest in this plane. Furthermore, the exchange between gravitational and elastic potential energy

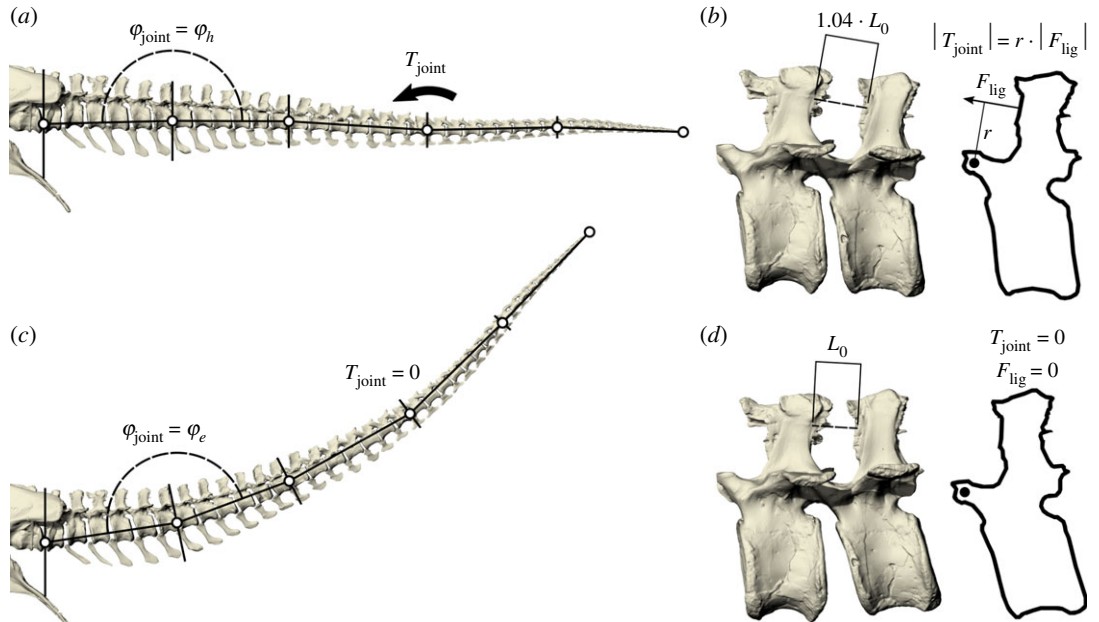

**Figure 2.** Postural mapping between the morphological reconstruction and the biomechanical model. (a) The joint angle ($\varphi_{\text{joint}}$) depends on the posture of the tail. The joint angle in horizontal equilibrium (indicated with $\varphi_h$) is shown for the second rotational spring. The joint torques ($T_{\text{joint}}$) counteract gravity. In the biomechanical model, the tail was placed perfectly horizontal (i.e. $\varphi_{\text{joint, 1–5}} = \pi$ rad). (b) In horizontal equilibrium, the ligaments are strained to 4% of their resting length $L_0$ [39]. The direction of ligament force $F_{\text{lig}}$ is shown, and its moment arm $r$, and its torque about the rotational axis equals $T_{\text{joint}}$. (c) The joint angle in extended posture ($\varphi_e$) is shown for the same rotational spring as in (a). This corresponds to a *T. rex* lying on its side, which implies that $T_{\text{joint}}$ is zero in this plane. (d) In extended posture, the vertebrae are rotated so that the ligaments are at their resting lengths $L_0$. The ligaments are not generating any force, and in the absence of gravity in this plane, this vertebra (and thus the biomechanical model) is in passive equilibrium. The vertebral rotation in this panel has been exaggerated to clarify the difference from (b).

is most meaningfully studied in this plane. Therefore, similar to previous researchers [27,30], we elected to perform a sagittal plane analysis. Inertial properties of the rigid bodies were acquired from the corresponding segments of the morphological reconstruction. Each of the rotational springs was assumed to generate a torque that increased quadratically with joint angle when stretched [42]. As a result, each joint spring has two parameters: joint angle at which the spring torque is zero, and a stiffness parameter. To assign values to these parameters, the biomechanical model was aligned with the morphological reconstruction in two different postures in which the spring torques were known (figure 2). We acquired the first posture by defining passive horizontal equilibrium, which implies that the ligaments are strained in horizontal posture to counteract gravity (figure 2a,b). It has been shown that the interspinous ligaments could generate ample force to maintain this pose [20], which our models confirm (see the electronic supplementary material). In the second posture, all interspinous ligaments were at resting length, and thus, all spring torques equalled zero. We determined this posture on the basis of an assumption regarding the ligament strain in horizontal equilibrium. Tendons and ligaments can be roughly divided into low-stress and high-stress varieties, with the high-stress varieties providing more energy savings at the cost of lower safety factors [39]. We imposed an intermediate 4% strain in horizontal equilibrium ($\varepsilon_{\text{hor}} = 0.04$), and used this to find a pose where the ligaments are not strained (figure 2c,d). This is analogous to passive equilibrium in the absence of gravity, or a *T. rex* lying on its side. After aligning the biomechanical model with the morphological reconstruction in both postures, the mechanical equilibrium conditions were used to calculate the parameter values of each rotational spring.

In the absence of damping, the resonance of the tail would occur at the undamped natural frequency ($f_n$) of the system [43]. In reality, damping is omnipresent. In that case, resonance occurs at a forcing frequency of $f_n \cdot (1 - 2\beta^2)^{0.5}$, with $\beta$ being the damping ratio [43]. Ligaments (like tendons) display relatively little energy loss during work-loop experiments [39,44], so they have low damping ratios. However, damping of dorsoventral oscillations would be dependent on more than just the ligaments, and a structure-based estimate of damping would be difficult, if not impossible. Therefore, we will use the undamped natural frequency of the biomechanical model as a proxy for the resonance frequency. To calculate the natural frequency of the biomechanical model, we first linearized the equations of motion in horizontal equilibrium using standard numerical methods. Next, the five pairs

of conjugated purely imaginary eigenvalues of the system matrix were determined using the 'eig' function in MATLAB. Subsequently, the natural frequency was calculated as the oscillation frequency pertaining to the fundamental resonance mode, i.e. the mode with the lowest eigenfrequency. Finally, walking speed was estimated by multiplying the natural frequency of each model by step length, defined as half the distance between two consecutive footfalls of the same foot. We determined a step length of 1.94 m by scaling a large tyrannosaurid trackway [45] based on footprint length of RGM.792000 (see the electronic supplementary material for details).

As was done in previous studies on dinosaur dimensions [27,33,46,47], we subjected the major inputs to a sensitivity analysis. Since we hope to see future studies of dinosaur locomotion incorporate our tail model, we intentionally chose wide bounds for the sensitivity analysis. This serves to inform researchers which individual steps in the modelling process induce the largest variation in the result. These steps could then be the focus of future research in an attempt to reduce the uncertainty.

To account for uncertainty in inertial estimates, we individually varied length (and thus moment of inertia) and mass. Because we assumed 4% ligament strain in horizontal equilibrium, we investigated the effects of low- and high-stress ligaments on the result, by imposing 3% and 5% strain in horizontal equilibrium, respectively [39]. Although each ligament is well-bounded by the skeleton, the effective point of force application of each ligament was unknown. In the baseline model, we based the moment arms on the area centroids of the ligaments. We used the ligament reconstruction to determine bounds for the moment arms (electronic supplementary material, figures S5 and S7). Lastly, whereas the proximal tail shows pronounced articulations between the zygapophyses, after approximately the 13th caudal this is no longer the case (electronic supplementary material, figure S5). This shifts the rotational axes in the distal tail ventrally, towards the vertebral centra, with the vertical position dependent on whether the tail is in flexion or extension (electronic supplementary material, figure S6 shows the baseline and two extreme possibilities). We, therefore, bounded the rotational axes in tail segments 3 and 4, which affected joint springs 3–5. The dorsal axes are unrealistically high, ensuring a wide bound for the sensitivity analysis. The ventral axes are located at the vertebral articulation with the chevrons, and represent their mechanical effect when the tail is flexed. In total, the aforementioned bounds led to 11 different models.

To encourage the adoption of our compliant tail model for dinosaur locomotion, we have provided in-depth descriptions of reconstruction and biomechanical modelling steps, mathematical derivations, scaling relationships and supplementary results in a combined document titled 'Supplementary texts' as electronic supplementary material. Reconstructions, 3D modelling and measurements were all done in Rhinoceros 6 (McNeel and Associates, Seattle, WA, USA). The biomechanical model was constructed using custom-code written in MATLAB 2019b (MathWorks, Natick, MA, USA). A 3D model of the *iliosacrum* and caudal skeleton of RGM.792000, and all custom MATLAB scripts are provided as electronic supplementary material.

# 3. Results

Natural frequencies and walking speeds determined in this study, as well as key outcomes of the sensitivity analysis, are reported in table 1. The natural frequency of our baseline model of the tail was $0.66 \, s^{-1}$, which we interpreted as the preferred step frequency for *T. rex*. This corresponds to a PWS of $1.28 \, m \, s^{-1}$. To provide the reader with an impression of the dynamic behaviour that results, we have provided a simulation of a lightly damped version of the baseline model (electronic supplementary material, video S1). In this simulation, the only input to the tail was a sinusoidal motion at the base of the tail with an amplitude of 0.08 m. Assuming pendular walking, this is the vertical motion that results from taking 1.94 m long steps at a hip height of 3.1 m. No muscle forces were included, so the phase relationship between the vertical oscillations of the hip and tail may not be representative of a muscle-actuated tail. In this simulation, a small amount of damping was introduced in order to obtain a finite tail amplitude. The damping ratio of the fundamental eigenmode was 0.16, which has a negligible effect on the resonant frequency [43].

The effects of changes in length (and thus inertia), mass and strain are mathematically predictable in our model, and we have thus derived general scaling laws for these parameters in the electronic supplementary material. Mass had zero effect on the natural frequency. Length ($L$) and strain in the horizontal position ($\varepsilon_{hor}$) are related to the natural frequency ($f_n$) as follows: $f_n \propto L^{-0.5}$ and $f_n \propto \varepsilon_{hor}^{-0.5}$. These relations can also be applied to the altered moment arms or distal axes models. For instance, a 6% longer tail, with 10% more strain in horizontal equilibrium (4.4% strain instead of 4%), using the maximal moment arms, would lead to a natural frequency of $0.79 \times 1.06^{-0.5} \times 1.1^{-0.5} = 0.73 \, s^{-1}$. The

**Table 1.** Natural frequencies and walking speeds of baseline and all sensitivity analysis models. Mass, length ($L$) and strain in horizontal position ($\varepsilon_{hor}$) models have a mathematically predictable effect on natural frequency ($f_n$), which is why they are placed together. Mass had zero effect. $f_n \propto L^{-0.5}$ and $f_n \propto \varepsilon_{hor}^{-0.5}$, and these scaling laws can be combined, and also applied to the moment arms or the alternative axes models. The alternative axes models do not represent an entire step cycle, which is why their combination with the moment arms models should be interpreted with caution. We report their combinations in electronic supplementary material, table S2.

| model | relative change in tail natural frequency from baseline (%) | tail natural frequency (s$^{-1}$) | corresponding walking speed (m s$^{-1}$) |
|---|---|---|---|
| baseline | n.a. | 0.66 | 1.28 |
| mass +8% | 0 | 0.66 | 1.28 |
| mass −8% | 0 | 0.66 | 1.28 |
| length +8% | −3.8 | 0.63 | 1.23 |
| length −8% | +4.3 | 0.69 | 1.33 |
| high-strain (+25%) | −10.6 | 0.59 | 1.14 |
| low-strain (−25%) | +15.5 | 0.76 | 1.48 |
| max. moment arms | +20.6 | 0.79 | 1.54 |
| min. moment arms | −14.3 | 0.56 | 1.09 |
| ventral axes[a] | +6.7 | 0.70 | 1.36 |
| dorsal axes[a] | −13.3 | 0.57 | 1.11 |

[a]These models only changed the moment arms within the 3rd–5th segments; segments 1 and 2 remained unchanged from the baseline model.

moment arms models cannot be combined with the distal axes models in this way, because these models are based on measurements of morphological traits, instead of systematic parameter variations. Their combinations are reported in electronic supplementary material, table S2, and the above proportionality relations can be applied to those combination effects as well.

Varying the moment arms had the largest overall effect on tail natural frequency, followed by the models accounting for strain magnitude. Together, these models accounted for uncertainty in the effective point of force application and composition of the caudal interspinous ligaments. We have supplied a spreadsheet as electronic supplementary material that interactively calculates natural frequency and walking speed, depending on parameter changes input by the user.

## 4. Discussion

The primary goal of this study was to develop a reductionist method to estimate the tail natural frequency of *T. rex*, and to combine the results of this method with trackway data for a baseline estimate of PWS (table 1). Our secondary goal was to determine which morphological features of the tail have the largest effect on its natural frequency, and these are represented by all the other models in table 1. Overall, results are most sensitive to uncertainties regarding ligament composition and vertebral kinematics in the tail of *T. rex*, while being minimally sensitive to length (and thus moment of inertia), and not affected by mass.

Our model predicts several scaling laws which provide insight into the locomotion of *T. rex* (derivations are provided in the electronic supplementary material). Natural frequency scales with the inverse square root of tail length (table 1). This implies an ontogenetic decrease in step frequency, but an increase in PWS proportional to $L^{0.5}$.

It has previously been proposed that the distal tail may act as a 'dynamic stabilizer' [19]. Varying the distal rotational axes revealed that even though the distal three segments accounted for only 14.5% of the total caudal mass, they still had a meaningful effect on the natural frequency (table 1). Our results support the notion that the medio-distal tail plays an important role in overall tail dynamics. Furthermore, the ventral axes model suggests that the chevrons increase locomotor speed when the tail is in slight flexion, by increasing the moment arm of the interspinous ligaments. Combining altered moment arms with alternative rotational axes strengthened this effect (electronic

supplementary material, table S2). However, this table should be interpreted with caution because the distal axes models do not represent axes that would have been used throughout the whole step cycle. Instead, the rotational axes would migrate ventrally during flexion, and dorsally during extension of the tail, which could potentially be influenced by contraction of the epaxial and hypaxial musculature. Thus, the bounds in electronic supplementary material, table S2 are unrealistically wide, due to the assumptions underlying this table. Yet, to acknowledge the uncertainties in dinosaur gait reconstruction, we used these as the reported range in our result.

Varying the amount of strain imposed on the ligaments in the horizontal posture ($\varepsilon_{hor}$, expressed as a fraction) had a similar effect as varying the length: $f_n \propto \varepsilon_{hor}^{-0.5}$. This is analogous to increasing or decreasing the stiffness at a constant moment of inertia. Ligaments that are subject to high levels of strain have increased energy-saving properties [39]. Our high-strain model displayed a lower natural frequency, implying that selective pressures for higher PWS may result in less energy-efficient locomotion, and vice versa.

Our baseline model suggests a PWS of 1.28 m s$^{-1}$ for *T. rex*, which we interpret to be near its energetically optimal walking speed. There are no extant analogues for tail-dependent obligate bipedal locomotion, especially combined with such a heavy emphasis on elastic storage. In fact, energetically optimal walking speed has only been measured in a few species, so comparisons are limited to these studies. In the animal kingdom, optimal walking speed is reported to be 1.0 m s$^{-1}$ for ratites [3] (average of the net and total MCOT) and elephants [5], 1.34–1.42 m s$^{-1}$ for humans [1,2] and 1.25 m s$^{-1}$ for horses [4], and these speeds are closely related to their respective PWS. PWS for giraffes [38] (reported as 'semi-selected') and migratory gnus and gazelles [48] have also been determined to lie close to this range. This is a remarkably close distribution, especially given the wide range in body size, shape and locomotor modes. Most of our estimates of *T. rex* walking speed fall within this range, with two of them only slightly exceeding it.

Very little work has been done to directly investigate PWS of dinosaurs. Limb natural frequencies have been used to estimate 'comfortable walking speeds' of several sauropod taxa [10], and dynamic simulations were used to estimate PWS of *Argentinosaurus* [49]. To our knowledge, no such work has been done on theropods. Until now, most estimates for (submaximal) dinosaur walking speeds are based on dynamic similarity (DS). Based on a regression of predominantly mammalian walking data, Alexander proposed an equation to estimate walking speed from trackways [23]. Of importance is the relative stride length, defined as the ratio between the stride length and hip height. This necessitates inferral of the hip height of the trackmaker, leading to uncertainties in the estimate. Despite qualitatively similar scaling predictions, our proposed NFM predicts lower speeds than trackway estimates using DS [23,24,32,45,50]. Such trackway estimates are not strictly energetically optimal, but we would expect them to tend towards the preferred speed if the sample of trackways is large enough. For large theropods, estimates tend to range between 2 and 3 m s$^{-1}$, which is much higher than the narrow range reported for extant animals. These higher speeds are commonly attributed to allometric effects [23,24,32,45,50]. A large source of potential errors, however, lies in the equation that relates relative stride length to walking speed, in which the step frequencies are implicit [24,51]. Alexander himself noted the high variability in his walking data, which leads to errors in speed estimates [51]. Not only is this true for dinosaurs, but even for the walking speed of the animals (including humans) on which the regression is based [10]. It has been shown that limb natural frequencies provided better predictions for elephants and giraffes near the preferred speed [10]. Near preferred speeds, DS consistently overestimated walking speeds, but the accuracy of the predictions improved at speeds substantially higher than the preferred speeds [10]. When compared to walking speeds of extant animals in a variety of sizes and gaits, DS also appears to overestimate the walking speed of bipedal dinosaurs. Neglecting tail dynamics may, therefore, be one of the oversimplifications inherent to DS, when applied to the inverted pendulum model of walking.

Recent amendments have been proposed to improve DS-based estimations, either by focusing on swing phase dynamics [52], or by incorporating taxon-specific morphological parameters to reduce uncertainty [53]. The latter are of course difficult to obtain from a fossilized trackway. Our method can help in this regard, by providing a range of plausible step frequencies for any given taxon. With this goal in mind, it would be most effective to estimate the natural frequencies of taxa that are well-represented in the trackway record. That way, it would be possible to combine step length data from several trackways, increasing confidence that the average result should tend towards preferred (and therefore optimal) gait. Unfortunately, this was not possible in the present study, because large theropod trackways with reasonably close taxonomic proximity to *T. rex* are exceedingly rare [45]. Bipedal dinosaurs seem to have preferred to walk at a relative stride length of 1.3, and it has been suggested that this might have been energetically optimal [24,45]. This relation implicitly relates footprint length to stride length, by first estimating hip height from footprint length, and also assumes

leg kinematics are known. Instead, we preferred to relate footprint length directly to stride length from a trackway that could be ascribed to *T. rex* with reasonable certainty [45]. This ensures sufficient geometric similarity to RGM.792000. The resulting step lengths differed by less than 4% from the suggested preferred relative stride length. Whenever considering trackway data, substrate-related uncertainties will prevent any straightforward interpretation [53,54], although presumably, these would not affect tail natural frequency.

DS does not incorporate MCOT, and therefore cannot be used to estimate PWS of dinosaurs. Therefore, when estimating dinosaur foraging costs, researchers have used DS to scale the walking speed [29], or even kept it at a constant $2 \, \mathrm{m \, s^{-1}}$ for larger taxa [31]. However, methods to calculate MCOT are heavily dependent on both speed and mass [28], so we suggest that any comparison of foraging costs should use the PWS of the respective taxa as a starting point. This could be done using the natural frequency of the legs [10], musculoskeletal simulations [49], or NFM as we propose it.

To our knowledge, musculoskeletal simulation models have not been used to estimate how MCOT varied with walking speed in *T. rex*, although this has been done for *Argentinosaurus* [49]. A simulation model requires estimation of joint rotational axes, body mass and moment arms for both the musculature and the ligaments [27,30,34,41,49]. The muscular reconstruction then provides further uncertainties in fibre composition type and architecture, both of which would significantly influence power output. Simulation studies have generally focused on maximal speed, and have contributed to the consensus that a long flight phase for large theropods would be unlikely [25,27,30,35]. However, the extensive muscular reconstructions often strongly affect the biomechanical simulations [41,49]. The largest uncertainty in the present study is related to the moment arms of the interspinous ligaments. This is an encouraging finding, because their extent is well-bounded by the skeleton. Essentially, one of the more certain parameters provides the most uncertainty. Future research on *in vivo* vertebral kinematics, dynamics and ligament compositions of crocodilian tails could reduce this uncertainty. Similarly encouraging is that our results are minimally sensitive to estimates of muscularity or inertial parameters in general. Indeed, when isolating mass as a free parameter in our sensitivity analysis, we have shown that our predicted speeds are unaffected by mass-estimates (table 1). This is due to the assumption that the ligaments could passively support the tail: adding muscle mass would make the ligaments proportionately stiffer, leading to the same natural frequency (see the scaling relationships in the electronic supplementary material). This assumption mimics the adaptations that naturally would have occurred in the connective tissue, to keep the tail horizontal as the animal gained mass throughout its life. It has been shown that body mass does not affect PWS in humans [1], and this is also in accordance with DS [23]. Minimal sensitivity to muscular and inertial estimates implies that our inverse dynamic approach to constructing a non-rigid tail could be incorporated into more sophisticated hindlimb simulation models, without adding much to the overall uncertainty of the result. This could have implications for maximal running speeds of large taxa like *T. rex*: maximum running speed was shown to be limited by peak stresses on the limbs [30], but a compliant tail may serve to reduce these stresses.

Our method for walking speed estimation is meant to be a reductionist analysis, so we have only incorporated the interspinous ligaments, since we expect their mechanical effect to dominate overall tail dynamics. Natural frequencies play an important role in animal locomotion [6–13], and NFM, therefore, provides a reasonable starting point for investigating how the resonant properties of dinosaur tails affect overall locomotion. However, in reducing the analysis to the mechanics of a single structure, many of the complex interactions are ignored. For instance, animals tune the frequencies of their segments through muscular contractions [6,7]. *Tyrannosaurus rex* could have contracted its epaxial and hypaxial caudal musculature to add rotational stiffness to the tail, which could be beneficial at higher speeds, for instance during a pursuit. Our ventral axes model demonstrates a possibility in this regard: if *T. rex* were to keep its distal tail in slight flexion, thereby migrating the rotational axes of the vertebrae to the articulations with the chevrons, the overall natural frequency of the tail would increase.

Tail musculature could also be employed to enforce beneficial phase relationships between vertical hip and tail oscillations. During walking, vertical oscillations between withers and heads in most ungulates are out of phase, which provides them with an energetic benefit [36–38,55]. This is theorized to be modulated by natural frequencies of the neck segment [37,38], but there is an active component to this behaviour as well [36]. This phase relationship reduces the losses incurred during the step-to-step transitions, which are identified as major sources of energy losses in inverted pendulum models of walking [56]. The implications of energy storage are also further complicated by the serial elasticity in musculotendon complexes: tendon stiffness can substantially impact the mechanical work done by the muscle fibres [57]. Finally, the cost of cyclical muscle contractions could also affect metabolic optima, depending on the task requirements [16,17]. Such complex interactions

would undoubtedly affect PWS, but require extensive reconstructions of muscular contractile properties. Muscular parameters currently provide the biggest uncertainty in most musculoskeletal simulations of dinosaurs [41,49,58], so any investigation of these interactions would require careful consideration. Inclusion of our compliant tail model into fully actuated hindlimb simulations of *T. rex* may shed further light on how the unique tails of non-avian dinosaurs functioned during locomotion. However, in such complex simulations, the intricacies of tail dynamics may be overshadowed by the uncertainties regarding the contractile properties of the muscles.

We set out to develop a method that does not require assumptions regarding contractile properties, so we accept that this limits the predictive power of our method. Given the overall uncertainties we must deal with when investigating the locomotion of extinct animals, we consider the simplicity of the method to be one of its strengths. Provided the neck or tail posture was supported passively, our method could be used to arrive at an estimate of the PWS of any sufficiently complete dinosaur, without the need to estimate contractile properties.

# 5. Conclusion

Gait reconstruction of dinosaurs has numerous inherent uncertainties, and therefore it is important to compare results from different methods, in an attempt to find a convergent point. We have proposed a method based on reconstructing the natural frequency of the vertical swaying of the tail, which we refer to as the Natural Frequency Method. This method requires relatively few assumptions, which are furthermore different assumptions to other methods. Our results for preferred walking speed of *T. rex* are lower than previous estimations for large theropods, but more closely match the preferred walking speeds of a variety of extant animals, regardless of gait pattern and body size. This seems to call in question the oft-cited high walking performance of bipedal dinosaurs due to their cursorial adaptations. Investigating vertebral kinematics and ligament composition in extant archosaurian tails could further constrain predictions based on our method, which could, in turn, be used to improve trackway estimates and foraging costs. Using simple, yet well-established principles from animal locomotion, the Natural Frequency Method provides an independent line of evidence to explore the locomotion of non-avian dinosaurs.

Data accessibility. Supplementary texts, figures S1–S7 and tables S1–S2, video S1, a spreadsheet to calculate all possible parameter effects of our models, custom MATLAB code written for the analyses and 3D scans of the iliosacrum and caudal skeleton of RGM.792000 are provided as electronic supplementary material.

Authors' contributions. P.A.v.B., A.J.K.v.S. and A.S. conceived the project. A.J.K.v.S. designed the biomechanical model. P.A.v.B. performed the reconstructions, measurements, coding. P.A.v.B., A.J.K.v.S. and A.S. wrote the manuscript.

Competing interests. The authors declare no competing interests;

Funding. Nothing to declare.

Acknowledgements. We thank K. Lemaire, M. Bobbert and D. Kistemaker for their vital role in early discussions. Further thanks to V. Vanhecke for 3D scanning, P. Larson for providing excavation maps and D. Tanke for providing photographs. H. Mallison and P. Manning are thanked for in-depth discussion and feedback on earlier versions of this manuscript. Discussions with W. Sellers, K. Bates and F. van Diggelen further helped us to contextualize our work. Lastly, we extend our gratitude to two anonymous reviewers whose constructive feedback and insights helped shape the final manuscript.

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
