## [Peer Review File · Royal Society Open Science]

Review History

RSOS-201441.R0 (Original submission)

Review form: Reviewer 1

Is the manuscript scientifically sound in its present form?

Yes

Are the interpretations and conclusions justified by the results?

Yes

Is the language acceptable?

Yes

Do you have any ethical concerns with this paper?

No

Have you any concerns about statistical analyses in this paper?

No

Recommendation?

Accept with minor revision (please list in comments)

Comments to the Author(s)**SUMMARY:**

The authors propose a new method (called the Natural Frequency Method) for estimating walking speeds of extinct animal locomotion and use the *T. rex* as an example. Using 3D volumetric reconstructions, they estimate input parameters of a biomechanical model that considers the tail as a series of body segments connected by torsional springs. The fundamental resonance of the system is determined and multiplied by step length of a preserved trackway in order to determine optimal walking speed. This method distinguishes itself as an alternative to extrapolating allometric scaling relationships and can provide insights on the energetics of foraging behaviors. Overall, I like this approach and appreciate the focus on mechanistic locomotion energetics. I have provided some comments regarding nuances to the resonance-matching argument outlined here and hope the authors find them useful.

GENERAL COMMENTS:

- The authors argue that metabolic cost of transport should be minimized when strain energy is maximized in the interspinous ligaments – by using a step frequency matched to the fundamental resonance of the tail structure. The authors reference a study that showed the energetic cost of human load carriage is benefited from a compliant suspension backpack (Rome et al., 2006). However, evidence on the energetic advantages of such devices are mixed overall. Kram (1991) and Foissac et al. (2009) found increases in metabolic cost while Castillo et al. (2014) found a decrease in cost. A recent optimization study has shown that the ratio between step frequency and resonance can help explain the different results (Schroeder et al., 2019).

The authors of the current manuscript acknowledge that out-of-phase oscillations are desired to maximize strain energy, however this phase relationship does not necessarily occur when step frequency matches the resonant frequency. For a simple spring mass model stimulated by a sinusoidal forcing function, phase is 90 degrees when frequency equals resonance. At frequencies below resonance, phase is 0 degrees (in phase) and at frequencies above resonance, phase is 180 degrees (out of phase). This transition is impulsive when no damping is present. The more damping that is present, the more gradual the phase transition is (Castillo et al., 2014; Schroeder et al., 2019).

Regardless, it is true that relative amplitude peaks at resonance. Thus, peak strain energy tends to occur at frequencies slightly above resonance, depending on how much damping is present. The optimization model from Schroeder et al. (2019) shows that cost for human load carriage is optimized in this frequency range, as did Castillo et al. (2014) when they found reduced cost. On the other hand, frequencies slightly below resonance can be quite costly (Foissac et al., 2009). Other simulation studies have shown that step frequencies much higher than resonance can still provide an energetic benefit (Ackerman & Seipel, 2014) and these predictions agree with Rome et al.'s findings (2006).

In short, current evidence suggests that locomotion energetics are benefited by interacting with oscillating loads when step frequency is greater than resonant frequency, not equal to. Perhaps this may help to explain why the authors' estimate of optimal walking speed is somewhat lower than the walking speeds predicted by alternative approaches and studies. It is unclear if the amplitude/phase relationships of the spring mass translate to the multi rigid body model of the current manuscript. However, this could be investigated by imposing harmonic motion at the base of the tail.

Ackerman, J., & Seipel, J. (2014). Short communication A model of human walking energetics with an elastically-suspended load. *Journal of Biomechanics*, 47(8), 1922–1927. <https://doi.org/10.1016/j.jbiomech.2014.03.016>

Castillo, E. R., Lieberman, G. M., McCarty, L. S., & Lieberman, D. E. (2014). Effects of pole compliance and step frequency on the biomechanics and economy of pole carrying during human walking. *Journal of Applied Physiology*, 117(5), 507–517. <https://doi.org/10.1152/jappphysiol.00119.2014>

Foissac, M., Millet, G. Y., Geysant, A., Freychat, P., & Belli, A. (2009). Characterization of the mechanical properties of backpacks and their influence on the energetics of walking. *Journal of Biomechanics*, 42(2), 125–130. <https://doi.org/10.1016/j.jbiomech.2008.10.012>

Kram, R. (1991). Carrying loads with springy poles. *Journal of Applied Physiology* (Bethesda, Md. : 1985), 71(3), 1119–1122. <http://www.ncbi.nlm.nih.gov/pubmed/1757307>

Kuo, A. D. (2001). A simple model of bipedal walking predicts the preferred speed-step length relationship. *Journal of Biomechanical Engineering*, 123(3), 264–269. <https://doi.org/10.1115/1.1372322>

Rome, L. C., Flynn, L., & Yoo, T. D. (2006). Rubber bands reduce the cost of carrying loads. *Nature*, 444(7122), 1023–1024. <https://doi.org/10.1038/nature4441023a>

Schroeder, R. T., Bertram, J. E. A., Nguyen, V. S., Hac, V. V., & Croft, J. L. (2019). Load carrying with flexible bamboo poles: optimization of a coupled oscillator system. <https://doi.org/10.1242/jeb.203760>

-It seems reasonable to assume that the dynamics of a large oscillating tail should play a prominent role in determining the energetics of an organism such as the T. rex. However, there are likely many aspects of the locomotory system that influence energetics as a function of frequency. In humans, the choice of step frequency illustrates a trade off between work to redirect the velocity of the center of mass during the step-to-step transition (high cost at low frequencies) and costs associated with swinging the leg (high cost at high frequencies) (Doke et al., 2005; Kuo, 2001; Schroeder et al., 2019). Because of this trade off, humans typically swing their legs at much higher frequencies (approximately 1.5 Hz at a moderate walking speed of 1 m/s; (Bertram & Ruina, 2001)) than the resonant frequency of the leg (0.64 Hz; (Doke et al., 2005). Muscle activation costs have also been shown to increase with frequency (Doke & Kuo, 2007) and may play a role in determining optimal gait frequencies.

In short there may be many determinants of gait energetics besides just resonance and although the resonance of massive tail oscillations could play a large role, the authors should acknowledge that this is simply one mechanism that could influence optimal frequencies.

Bertram, J. E. A., & Ruina, A. (2001). Multiple Walking Speed–frequency Relations are Predicted by Constrained Optimization. *Journal of Theoretical Biology*, 209(4), 445–453. <https://doi.org/10.1006/jtbi.2001.2279>

Doke, J., Donelan, J. M., & Kuo, A. D. (2005). Mechanics and energetics of swinging the human leg. *Journal of Experimental Biology*, 208(3), 439–445. <https://doi.org/10.1242/jeb.01408>

Doke, J., & Kuo, A. D. (2007). Energetic cost of producing cyclic muscle force, rather than work, to swing the human leg. *Journal of Experimental Biology*, 210(13), 2390–2398. <https://doi.org/10.1242/jeb.02782>

Kuo, A. D. (2001). A simple model of bipedal walking predicts the preferred speed-step length relationship. *Journal of Biomechanical Engineering*, 123(3), 264–269.
<https://doi.org/10.1115/1.1372322>

Schroeder, R. T., Bertram, J. E. A., Nguyen, V. S., Hac, V. V., & Croft, J. L. (2019). Load carrying with flexible bamboo poles: optimization of a coupled oscillator system.
<https://doi.org/10.1242/jeb.203760>

-The authors reference evidence that various animals and humans tend to walk at the minimum of their respective metabolic cost of transport curves. Although this may be true in certain circumstances, locomotion speeds in an ecological context likely vary substantially. E.g. people in large cities tend to walk faster than those in small towns (Bornstein & Bornstein, 1976). Individuals move more slowly during short bouts of walking versus long durations (Seethapathi & Srinivasan, 2015). Surely, animals will move at higher speeds to escape or engage in predation, etc. Optimal foraging theory has been applied to determine appropriate movement vigor to show that movement speeds are chosen to match the rate of energy expenditure to the global capture rate of the sought-after reward (Shadmehr et al., 2019; Yoon et al., 2018). Other studies have shown that individuals adjust gait type and speed in response to sensitivities to time constraints and/or costs (Summerside et al., 2018; Tiew & Srinivasan, 2020)

Given that high amplitude, out-of-phase oscillations of the tail should potentially benefit locomotion energetics at step frequencies greater than the resonant frequency, it may be more useful to think of the tail's fundamental resonance as a lower bound frequency for an animal to walk at. Conceptually, this allows an upper range of speeds that should be economical for the T. rex to walk at, depending on the task goal and other relevant circumstances that may affect chosen locomotion speeds.

Bornstein, M. H., & Bornstein, H. G. (1976). The pace of life. *Nature*, 259(5544), 557–559.
<https://doi.org/10.1038/259557a0>

Seethapathi, N., & Srinivasan, M. (2015). The metabolic cost of changing walking speeds is significant, implies lower optimal speeds for shorter distances, and increases daily energy estimates. *Biology Letters*, 11(9), 20150486. <https://doi.org/10.1098/rsbl.2015.0486>

Shadmehr, R., Reppert, T. R., Summerside, E. M., Yoon, T., & Ahmed, A. A. (2019). Movement Vigor as a Reflection of Subjective Economic Utility. In *Trends in Neurosciences* (Vol. 42, Issue 5, pp. 323–336). Elsevier Ltd. <https://doi.org/10.1016/j.tins.2019.02.003>

Summerside, E. M., Kram, R., & Ahmed, A. A. (2018). Contributions of metabolic and temporal costs to human gait selection. *Journal of The Royal Society Interface*, 15(143), 20180197.
<https://doi.org/10.1098/rsif.2018.0197>

Tiew, H., & Srinivasan, M. (2020). Pre-crastination: Time deadlines increase walking speeds even when not constraining. *BioRxiv*, 2020.07.17.208140. <https://doi.org/10.1101/2020.07.17.208140>

Yoon, T., Geary, R. B., Ahmed, A. A., & Shadmehr, R. (2018). Control of movement vigor and decision making during foraging. *Proceedings of the National Academy of Sciences of the United States of America*, 115(44), E10476–E10485. <https://doi.org/10.1073/pnas.1812979115>

MINOR COMMENTS:

Line 39-40: As mentioned in previous comments, natural frequency may be a part of the equation but is unlikely to be the only thing that matters with regards to the determination of optimal

walking speed. Perhaps these lines could be reworded slightly: “Optimal locomotor speeds may be influenced by the natural frequencies of relevant body segments...”. Furthermore, Doke et al. (2005) is referenced here, however, resonant leg frequency reported in this study is very low (0.64 Hz) and does not predict preferred frequencies observed during optimal walking speed in humans.

Figure 1: I like the biomechanics model used in this study. However, I do wonder how much of the model’s complexity matters for the results on fundamental resonant frequencies. Would approximating the dynamics of the tail as a cantilever beam or even as a simple spring mass change the results by very much? I do appreciate the fact that tissue-specific outcomes (e.g. ligament rupture, Young’s modulus, etc.) are more readily evaluated with the more complex model. However, I cannot help but wonder if the gross dynamics of the tail’s oscillation could be approximated with a more accessible, reductionist modeling approach.

Figure 1: It could be helpful to show force vectors expected from muscle contraction and/or ligament spring forces (i.e. a free body diagram) in order to help illustrate the mechanical interaction conceptually

Line 65: This is a little confusing since there is a range of natural frequencies associated with the bounds in the sensitivity analysis. Perhaps consider clarifying your meaning “The lowest natural frequency (i.e. the fundamental resonant mode)...”

Line 72-75: As was mentioned in a previous comment, oscillations may not occur entirely out of phase at resonant frequency where peak relative amplitude is maximized. Can the authors confirm this is the case for their model? Perhaps a Bode plot would be useful in illustrating this.

Line 84-85: Consider word change “the most prominent movements and forces occur in the sagittal plane”. “Importance” is subjective and not clearly defined in this sentence. Another option is “the most energetically relevant movements and forces”

Line 123: Change to “modelling steps, as well as mathematical derivations...”

Line 117: How much confidence do the authors have that this measurement of step length (N=1) is representative of typical gait? I understand that there are inherent limitations to the data available in this field, but are there other trackways to compare with?

Line 113-115: Perhaps add “These bounds account for uncertainties... and led to 11 models (see ESM for details)”

Table 1: The authors explored sensitivity of their model to parameter uncertainties independently, but this assumes that uncertainty cannot accumulate in multiple dimensions simultaneously. Shouldn’t the low/high bounds of natural frequency and resulting optimal walking speed reflect maximal error in all dimensions simultaneously? i.e. the combination of parameter bounds that results in the lowest and highest natural frequency. Further constraints on Young’s modulus, etc. can be applied in case these bounds give unrealistic results.

Line 157: Consider using a different word from “significant” to avoid conflation with statistical significance

ESM, Line 186: Change wording to “Due to the presence of these many unknowns...”

ESM, Line 278: Missing word, change to “the stiffnesses of the ligaments...”

Review form: Reviewer 2

Is the manuscript scientifically sound in its present form?

No

Are the interpretations and conclusions justified by the results?

No

Is the language acceptable?

Yes

Do you have any ethical concerns with this paper?

No

Have you any concerns about statistical analyses in this paper?

No

Recommendation?

Major revision is needed (please make suggestions in comments)

Comments to the Author(s)

The authors present a method for estimating the optimal walking speed of *Tyrannosaurus rex*. They claim that minimal metabolic expenditure should occur when step frequency matches tail resonant frequency. They reconstruct tail morphology of a *T. rex* specimen to estimate inertial parameters, lengths of tail segments, positions of ligaments, and axes of rotation. They perform a sensitivity analysis by shifting each of these elements individually from baseline. By approximating each segment as a rigid body attached to the next segment by a torsional spring, they calculate the resonant frequency of the tail. Using trackways, they then estimate the walking speed corresponding to this frequency, and claim that it is the optimal walking frequency.

The tail reconstruction is detailed and informative. The sensitivity analysis leads to some interesting insights, for example that the distal tail has pronounced effects on the resonant frequency depending on the location of the rotational axis.

The paper provides a method to model tail oscillations in extinct dinosaurs, and another method to estimate walking speeds. Despite these strengths, some major issues should be addressed before publication.

The basic premise- that optimal walking speeds correspond to resonant frequencies- is not well supported. Interaction effects are ignored in the sensitivity analysis. Key aspects of the methods and context for the discussion are relegated to supplemental information. Finally, while the biomechanical model seems sensible, the mathematical derivation involves ambiguous terminology and appears inconsistent as written.

If the authors can address these concerns, I believe this paper would be a useful addition to the literature.

Major revisions

--

Lines 39-41: "Optimal locomotor speeds are to a large extent the result of the natural frequencies of relevant body segments, through a combination of resonance and elastic storage (9-12)."

This is a bold claim, and deserves a little more discussion than simply pointing to references 9-12. What is the specific evidence that natural frequencies determine optimal locomotor speeds? Some examples could be provided.

As it stands, I am not fully convinced that all the references support the claim. For example, (10) show that work is minimized at natural leg swing frequency, but metabolic cost is minimized at a higher frequency, and do not show whether this is the preferred or optimal frequency in walking.

--

Lines 51-52: "Due to resonance, elastic storage would be maximal if step frequency matched the natural frequency of the tail."

Lines 69-70: "Elastic storage in the caudal ligaments would reduce the energy losses at each footfall, thereby reducing MCOT."

Lines 73-75: "Therefore, it would be beneficial to maximize tail amplitude (which occurs when step and natural frequency are matched), as this maximizes ligament strain and minimizes MCOT."

Maximal ligament strain does not necessarily mean minimal metabolic cost. If this were true, humans would stretch Achilles tendon maximally during walking / running, regardless of speed! Damping, active muscle volume, muscle activation all seem to contribute to metabolic cost and call for submaximal tendon/ligament strain (e.g. Collins et al. 2015 doi:10.1038/nature14288; Orselli et al. 2017 doi:10.1016/j.jbiomech.2017.06.022).

--

Lines 115-121: step length from trackways.

The authors already point to issues with substrate affecting walking speed. They claim that the walking speed reported, as determined from this step length is "energetically optimal". How do they know that the trackways were for a preferred (let alone optimal) step length?

Related: Lines 208-210: "DS has provided interesting qualitative observations. Bipedal dinosaurs seem to have preferred to walk at a relative stride length of 1.3..."

If hip height is known for the author's specimen, could they not simply use (hip height)*1.3 to get preferred stride length? How does this method compare to the trackway measurement?

--

The authors have not looked at interaction effects in the sensitivity analysis. I appreciate that this adds a combinatorial level of complexity (and time!) to the analysis, however I do think it is important.

At the very least, I would ask the authors to combine all the parameters that have the greatest increase and decrease in step frequency (e.g. decrease length + max. moment arms + ventral axes + low strain to test whether they combine to increase frequency further. Opposite parameters for decreased frequency). That way, we can at least have an idea of whether these effects are additive.

This minimal level of testing interactions would be sufficient in my view, but will leave room for doubt about whether resonance uncertainty is higher than reported.

If the authors have the time and motivation to look at all combinations, that would be ideal. I think it would make a much stronger paper, put to rest any doubt about untested combinations, and set a much more reliable measure of estimate uncertainty.

I would also note that computational time does not seem to be an issue; their matlab script calculates tail natural frequencies in fractions of a second. There may be some technical hassle in writing the code to iterate through the different combinations, but it should be doable with some nested loops.

--

Re: novelty of the method

Lines 176-177: Until now, estimates for (submaximal) dinosaur walking speeds are all based on Dynamic Similarity (DS).

Lines 212: "The possibility of lower step frequencies than predicted by the equation has only been suggested in passing (14,38). Until now, no steps have been taken to explore this."

Lines 223-225: "Deriving walking speed using Dynamic Similarity has its limitations. However, until now it was the only approach used to estimate (non-optimal) dinosaur walking speeds."

Dynamic Similarity is not the only approach used (though it is most common), the possibility of it being wrong has been explored (a little), and natural frequencies of body segments have been used before (though not the tail as far as I know).

For (extant) avian dinosaurs, much has been done, though I assume the authors meant extinct dinosaurs. They might be particularly interested in Marmol-Guijarro et al. (2020 doi:10.1186/s12983-020-00363-z), who question the use of dynamic similarity by applying it to ptarmigan tracks.

Sellers et al. (2013 doi: 10.1371/journal.pone.0078733) used a full musculoskeletal model to find metabolically optimal walking speeds in a sauropod.

Christian et al. (1999 doi:10.1002/mmng.1999.4860020105) investigated an alternative method from DS, similar to the "Natural Frequency Method" proposed in the submitted paper. They investigated whether the natural pendular period of limbs in giraffe and elephants was predictive of their step (or swing) frequency. Results were mixed, and they show that that dynamic similarity is a better predictor of walking speed from trackways in elephants than using the natural frequency of the limbs.

While they caution that natural pendulum frequencies won't give precise estimates in general, they do use natural pendulum frequency of limbs to estimate "comfortable walking speeds" of four species of dinosaurs.

I would suggest the authors preface their claim of novelty, with reference to this previous work.

--

Some discussion points are confusing without referencing the supplemental information. For example, in the main text there is no mention of how distal rotational axes are changed, or why only the 3rd and 4th segments are altered. Yet this is a key point relevant to the second discussion paragraph.

Similarly, the authors suggest an intriguing insight regarding the function of chevrons on lines 159-160, but the role of the chevrons in the model is lost to any reader who has not referenced the supplement.

Other points in the discussion are similarly confusing. I would encourage the authors to move enough of the supplement to the main text that the key discussion points are understandable without referencing the supplement.

--

The authors appear to be using a small-angle approximation in their derivations in the supplemental information. Is the small-angle approximation justified? What are the maximum relative angles between joints experienced during resonance?

--

While the geometry and physical principles of the biomechanical model are straightforward, some of the mathematics in the supplemental information are confusing, ambiguous or inconsistent. The key issues are

1. The conflation of $\Delta \epsilon$ with ϵ
2. The ambiguity of the meaning of ϕ , and related issues:
 - a. the possible conflation with $\Delta \phi$
 - b. the introduction of an undefined variable ϕ_e in equation 6
 - c. The ambiguity about rest angles
3. The introduction of R_s without its explicit connection to r

Below, I expand on these issues in turn:

1: Conflation of $\Delta \epsilon$ with ϵ

Between equations 5 and 6, the expression $R_s(\phi - \phi_e)$ is substituted for ϵ . Assuming that $(\phi - \phi_e) = \Delta \phi$ (but see point 2), the expression is still problematic since $R_s \Delta \phi = \Delta \epsilon$ by equation 2 and not ϵ .

2: Ambiguity of ϕ

On line 336, ϕ is defined as "the angle that a segment makes with the horizontal axis". This definition is unambiguous on its own. However, only relative angles between segments matter to the torque from a rotational spring (as the authors note on line 338). In particular, the relative angles should deviate from some "resting" relative angle, where there is no spring torque between segments.

The authors introduce a "rest angle" in equation 8. However, here the expression is confusing. If the interspinous ligaments "do not generate any... force" (Line 265), then the strain should be zero by equation 4. However, in equation 8 the strain is $\epsilon_{hor} = 0.04$.

I believe the authors mean $\Delta \phi$ to be a proper "relative angle" from a true resting angle (which is itself a difference between absolute joint angles of adjacent segments). This would make sense given equation (2) and the use of $\Delta \phi$ in equation 6 (besides the issues raised in point 1). But then $\phi - \phi_e = \Delta \phi$, and what is ϕ_e ? Is it different from ϕ_r ? If the authors meant to use ϕ_r instead of ϕ_e , then there is still the issue that ϕ_r is not truly a rest angle (i.e. a relative angle at which there is no torque from the spring).

3: R_s vs r/L_0

As discussed in points 1 and 3, there is ambiguity between the terms ϕ , $\Delta \phi$, and ϕr , as well as $\Delta \varepsilon$ and ε . This leads to ambiguity about R_s , which I hope would be resolved if the authors address points 1 and 2 above.

There is yet another (perhaps more minor) issue with R_s : its dependence on the moment arm r is hidden in the expression 2. Since the cubic relationship between r and T_{lig} is a major point of the paper, it would make more sense to replace R_s with its expanded form using r , or at least define it in terms of r earlier.

In its current form ($R_s = \Delta \varepsilon / \Delta \phi$), or in the form that seems more in line with the useage of R_s in equations 6 and 8 ($R_s = \varepsilon / \Delta \phi$) I believe $R_s = r/L_0$. Simply stating this relationship explicitly would greatly clarify the expressions in the paper.

--

Suppl Line 381: "...stiffness and damping of the tail would be mainly dependent on the ligaments."

I suspect the intervertebral discs would supply more damping. Can the authors comment on this, or even supply an estimate?

Minor revisions / typos

Main Manuscript

--

As the authors have the code to solve 2D dynamics, I would like to know what the tail looks like when oscillating at its natural resonant frequency. Can the authors produce animations of the tail for the different parameter combinations? This would provide another means of assessing whether the biomechanical predictions appear sensible.

--

Line 167: "There are no extant analogues for tail-dependent bipedal locomotion..."

Lizards and some pangolins may be modern extant analogues. See Persons and Curry (2017 doi: 10.1016/j.jtbi.2017.02.032) and its relevant references.

--

Lines 182-183: "Such trackway estimates are not strictly optimal, but on average they should tend towards OWS if the sample of trackways is large enough."

I suggest: "... they should tend towards preferred walking speed (and presumably OWS), if the sample..."

--

Line 188: "This is true for dinosaurs, but even for"

Suggest: "This is true not only for dinosaurs, but even for..."

--

Supplemental material

Line 141: "...tail is in extension of flexion"

Suggest: "... tail is in extension or flexion"

--

Line 303: "Stiffness of the ligaments were recalculated inverse dynamically..."

Suggest: "Stiffness of the ligaments were recalculated using inverse dynamics..."

--

Line 348: "... Segdyn rearranges all the unknowns..."

I believe Segdyn is the authors' custom code, rather than some proprietary software. The authors should make this clear in the text.

--

Figure S7a: the black ligament outline is hard to see. I would suggest a different (lighter) colour.

Decision letter (RSOS-201441.R0)

Dear Mr van Bijlert

The Editors assigned to your paper RSOS-201441 "Natural Frequency Method: reconstructing the tail for optimal walking speed of *Tyrannosaurus rex*" have now received comments from reviewers and would like you to revise the paper in accordance with the reviewer comments and any comments from the Editors. Please note this decision does not guarantee eventual acceptance.

Please submit your revised manuscript and required files (see below) no later than 21 days from today's (ie 26-Oct-2020) date. Note: the ScholarOne system will 'lock' if submission of the revision is attempted 21 or more days after the deadline. If you do not think you will be able to meet this deadline please contact the editorial office immediately.

Please note article processing charges apply to papers accepted for publication in Royal Society Open Science (<https://royalsocietypublishing.org/rsos/charges>). Charges will also apply to papers transferred to the journal from other Royal Society Publishing journals, as well as papers submitted as part of our collaboration with the Royal Society of Chemistry

(<https://royalsocietypublishing.org/rsos/chemistry>). Fee waivers are available but must be requested when you submit your revision (<https://royalsocietypublishing.org/rsos/waivers>).

on behalf of Dr Manoj Srinivasan (Associate Editor) and Kevin Padian (Subject Editor)
openscience@royalsociety.org

Associate Editor Comments to Author (Dr Manoj Srinivasan):

Associate Editor: 1

Comments to the Author:

The reviewers have provided detailed remarks and we invite a revised manuscript that addresses these issues.

I agree with the reviewers that it may not be possible to rigorously claim that the walking speeds are determined by the natural frequencies of the body segments. Superficially, this presumes that swinging body limbs is the primary or dominant energy cost term (assuming animals prefer low effort gaits). For instance, in humans, the walking step frequencies are chosen to be systematically faster than the natural frequency of the swing leg, thought to be because this faster swing lowers other components of the metabolic cost.

Kuo, Arthur D. "A simple model of bipedal walking predicts the preferred speed–step length relationship." *J. Biomech. Eng.* 123.3 (2001): 264-269.

Reviewer comments to Author:

Reviewer: 1

Comments to the Author(s)

SUMMARY:

The authors propose a new method (called the Natural Frequency Method) for estimating walking speeds of extinct animal locomotion and use the *T. rex* as an example. Using 3D volumetric reconstructions, they estimate input parameters of a biomechanical model that considers the tail as a series of body segments connected by torsional springs. The fundamental resonance of the system is determined and multiplied by step length of a preserved trackway in order to determine optimal walking speed. This method distinguishes itself as an alternative to extrapolating allometric scaling relationships and can provide insights on the energetics of foraging behaviors. Overall, I like this approach and appreciate the focus on mechanistic locomotion energetics. I have provided some comments regarding nuances to the resonance-matching argument outlined here and hope the authors find them useful.

GENERAL COMMENTS:

- The authors argue that metabolic cost of transport should be minimized when strain energy is maximized in the interspinous ligaments – by using a step frequency matched to the fundamental resonance of the tail structure. The authors reference a study that showed the energetic cost of

human load carriage is benefited from a compliant suspension backpack (Rome et al., 2006). However, evidence on the energetic advantages of such devices are mixed overall. Kram (1991) and Foissac et al. (2009) found increases in metabolic cost while Castillo et al. (2014) found a decrease in cost. A recent optimization study has shown that the ratio between step frequency and resonance can help explain the different results (Schroeder et al., 2019).

The authors of the current manuscript acknowledge that out-of-phase oscillations are desired to maximize strain energy, however this phase relationship does not necessarily occur when step frequency matches the resonant frequency. For a simple spring mass model stimulated by a sinusoidal forcing function, phase is 90 degrees when frequency equals resonance. At frequencies below resonance, phase is 0 degrees (in phase) and at frequencies above resonance, phase is 180 degrees (out of phase). This transition is impulsive when no damping is present. The more damping that is present, the more gradual the phase transition is (Castillo et al., 2014; Schroeder et al., 2019).

Regardless, it is true that relative amplitude peaks at resonance. Thus, peak strain energy tends to occur at frequencies slightly above resonance, depending on how much damping is present. The optimization model from Schroeder et al. (2019) shows that cost for human load carriage is optimized in this frequency range, as did Castillo et al. (2014) when they found reduced cost. On the other hand, frequencies slightly below resonance can be quite costly (Foissac et al., 2009). Other simulation studies have shown that step frequencies much higher than resonance can still provide an energetic benefit (Ackerman & Seipel, 2014) and these predictions agree with Rome et al.'s findings (2006).

In short, current evidence suggests that locomotion energetics are benefited by interacting with oscillating loads when step frequency is greater than resonant frequency, not equal to. Perhaps this may help to explain why the authors' estimate of optimal walking speed is somewhat lower than the walking speeds predicted by alternative approaches and studies. It is unclear if the amplitude/phase relationships of the spring mass translate to the multi rigid body model of the current manuscript. However, this could be investigated by imposing harmonic motion at the base of the tail.

Ackerman, J., & Seipel, J. (2014). Short communication A model of human walking energetics with an elastically-suspended load. *Journal of Biomechanics*, 47(8), 1922–1927. <https://doi.org/10.1016/j.jbiomech.2014.03.016>

Castillo, E. R., Lieberman, G. M., McCarty, L. S., & Lieberman, D. E. (2014). Effects of pole compliance and step frequency on the biomechanics and economy of pole carrying during human walking. *Journal of Applied Physiology*, 117(5), 507–517. <https://doi.org/10.1152/jappphysiol.00119.2014>

Foissac, M., Millet, G. Y., Geysant, A., Freychat, P., & Belli, A. (2009). Characterization of the mechanical properties of backpacks and their influence on the energetics of walking. *Journal of Biomechanics*, 42(2), 125–130. <https://doi.org/10.1016/j.jbiomech.2008.10.012>

Kram, R. (1991). Carrying loads with springy poles. *Journal of Applied Physiology* (Bethesda, Md. : 1985), 71(3), 1119–1122. <http://www.ncbi.nlm.nih.gov/pubmed/1757307>

Kuo, A. D. (2001). A simple model of bipedal walking predicts the preferred speed-step length relationship. *Journal of Biomechanical Engineering*, 123(3), 264–269. <https://doi.org/10.1115/1.1372322>

Rome, L. C., Flynn, L., & Yoo, T. D. (2006). Rubber bands reduce the cost of carrying loads. *Nature*, 444(7122), 1023–1024. <https://doi.org/10.1038/nature4441023a>

Schroeder, R. T., Bertram, J. E. A., Nguyen, V. S., Hac, V. V., & Croft, J. L. (2019). Load carrying with flexible bamboo poles: optimization of a coupled oscillator system. <https://doi.org/10.1242/jeb.203760>

-It seems reasonable to assume that the dynamics of a large oscillating tail should play a prominent role in determining the energetics of an organism such as the T. rex. However, there are likely many aspects of the locomotory system that influence energetics as a function of frequency. In humans, the choice of step frequency illustrates a trade off between work to redirect the velocity of the center of mass during the step-to-step transition (high cost at low frequencies) and costs associated with swinging the leg (high cost at high frequencies) (Doke et al., 2005; Kuo, 2001; Schroeder et al., 2019). Because of this trade off, humans typically swing their legs at much higher frequencies (approximately 1.5 Hz at a moderate walking speed of 1 m/s; (Bertram & Ruina, 2001)) than the resonant frequency of the leg (0.64 Hz; (Doke et al., 2005). Muscle activation costs have also been shown to increase with frequency (Doke & Kuo, 2007) and may play a role in determining optimal gait frequencies.

In short there may be many determinants of gait energetics besides just resonance and although the resonance of massive tail oscillations could play a large role, the authors should acknowledge that this is simply one mechanism that could influence optimal frequencies.

Bertram, J. E. A., & Ruina, A. (2001). Multiple Walking Speed–frequency Relations are Predicted by Constrained Optimization. *Journal of Theoretical Biology*, 209(4), 445–453. <https://doi.org/10.1006/jtbi.2001.2279>

Doke, J., Donelan, J. M., & Kuo, A. D. (2005). Mechanics and energetics of swinging the human leg. *Journal of Experimental Biology*, 208(3), 439–445. <https://doi.org/10.1242/jeb.01408>

Doke, J., & Kuo, A. D. (2007). Energetic cost of producing cyclic muscle force, rather than work, to swing the human leg. *Journal of Experimental Biology*, 210(13), 2390–2398. <https://doi.org/10.1242/jeb.02782>

Kuo, A. D. (2001). A simple model of bipedal walking predicts the preferred speed–step length relationship. *Journal of Biomechanical Engineering*, 123(3), 264–269. <https://doi.org/10.1115/1.1372322>

Schroeder, R. T., Bertram, J. E. A., Nguyen, V. S., Hac, V. V., & Croft, J. L. (2019). Load carrying with flexible bamboo poles: optimization of a coupled oscillator system. <https://doi.org/10.1242/jeb.203760>

-The authors reference evidence that various animals and humans tend to walk at the minimum of their respective metabolic cost of transport curves. Although this may be true in certain circumstances, locomotion speeds in an ecological context likely vary substantially. E.g. people in large cities tend to walk faster than those in small towns (Bornstein & Bornstein, 1976). Individuals move more slowly during short bouts of walking versus long durations (Seethapathi & Srinivasan, 2015). Surely, animals will move at higher speeds to escape or engage in predation, etc. Optimal foraging theory has been applied to determine appropriate movement vigor to show that movement speeds are chosen to match the rate of energy expenditure to the global capture rate of the sought-after reward (Shadmehr et al., 2019; Yoon et al., 2018). Other studies have shown that individuals adjust gait type and speed in response to sensitivities to time constraints and/or costs (Summerside et al., 2018; Tiew & Srinivasan, 2020)

Given that high amplitude, out-of-phase oscillations of the tail should potentially benefit locomotion energetics at step frequencies greater than the resonant frequency, it may be more useful to think of the tail's fundamental resonance as a lower bound frequency for an animal to walk at. Conceptually, this allows an upper range of speeds that should be economical for the T. rex to walk at, depending on the task goal and other relevant circumstances that may affect chosen locomotion speeds.

Bornstein, M. H., & Bornstein, H. G. (1976). The pace of life. *Nature*, 259(5544), 557-559. <https://doi.org/10.1038/259557a0>

Seethapathi, N., & Srinivasan, M. (2015). The metabolic cost of changing walking speeds is significant, implies lower optimal speeds for shorter distances, and increases daily energy estimates. *Biology Letters*, 11(9), 20150486. <https://doi.org/10.1098/rsbl.2015.0486>

Shadmehr, R., Reppert, T. R., Summerside, E. M., Yoon, T., & Ahmed, A. A. (2019). Movement Vigor as a Reflection of Subjective Economic Utility. In *Trends in Neurosciences* (Vol. 42, Issue 5, pp. 323-336). Elsevier Ltd. <https://doi.org/10.1016/j.tins.2019.02.003>

Summerside, E. M., Kram, R., & Ahmed, A. A. (2018). Contributions of metabolic and temporal costs to human gait selection. *Journal of The Royal Society Interface*, 15(143), 20180197. <https://doi.org/10.1098/rsif.2018.0197>

Tiew, H., & Srinivasan, M. (2020). Pre-crastination: Time deadlines increase walking speeds even when not constraining. *BioRxiv*, 2020.07.17.208140. <https://doi.org/10.1101/2020.07.17.208140>

Yoon, T., Geary, R. B., Ahmed, A. A., & Shadmehr, R. (2018). Control of movement vigor and decision making during foraging. *Proceedings of the National Academy of Sciences of the United States of America*, 115(44), E10476-E10485. <https://doi.org/10.1073/pnas.1812979115>

MINOR COMMENTS:

Line 39-40: As mentioned in previous comments, natural frequency may be a part of the equation but is unlikely to be the only thing that matters with regards to the determination of optimal walking speed. Perhaps these lines could be reworded slightly: "Optimal locomotor speeds may be influenced by the natural frequencies of relevant body segments...". Furthermore, Doke et al. (2005) is referenced here, however, resonant leg frequency reported in this study is very low (0.64 Hz) and does not predict preferred frequencies observed during optimal walking speed in humans.

Figure 1: I like the biomechanics model used in this study. However, I do wonder how much of the model's complexity matters for the results on fundamental resonant frequencies. Would approximating the dynamics of the tail as a cantilever beam or even as a simple spring mass change the results by very much? I do appreciate the fact that tissue-specific outcomes (e.g. ligament rupture, Young's modulus, etc.) are more readily evaluated with the more complex model. However, I cannot help but wonder if the gross dynamics of the tail's oscillation could be approximated with a more accessible, reductionist modeling approach.

Figure 1: It could be helpful to show force vectors expected from muscle contraction and/or ligament spring forces (i.e. a free body diagram) in order to help illustrate the mechanical interaction conceptually

Line 65: This is a little confusing since there is a range of natural frequencies associated with the bounds in the sensitivity analysis. Perhaps consider clarifying your meaning “The lowest natural frequency (i.e. the fundamental resonant mode)...”

Line 72-75: As was mentioned in a previous comment, oscillations may not occur entirely out of phase at resonant frequency where peak relative amplitude is maximized. Can the authors confirm this is the case for their model? Perhaps a Bode plot would be useful in illustrating this.

Line 84-85: Consider word change “the most prominent movements and forces occur in the sagittal plane”. “Importance” is subjective and not clearly defined in this sentence. Another option is “the most energetically relevant movements and forces”

Line 123: Change to “modelling steps, as well as mathematical derivations...”

Line 117: How much confidence do the authors have that this measurement of step length ($N=1$) is representative of typical gait? I understand that there are inherent limitations to the data available in this field, but are there other trackways to compare with?

Line 113-115: Perhaps add “These bounds account for uncertainties... and led to 11 models (see ESM for details)”

Table 1: The authors explored sensitivity of their model to parameter uncertainties independently, but this assumes that uncertainty cannot accumulate in multiple dimensions simultaneously. Shouldn't the low/high bounds of natural frequency and resulting optimal walking speed reflect maximal error in all dimensions simultaneously? i.e. the combination of parameter bounds that results in the lowest and highest natural frequency. Further constraints on Young's modulus, etc. can be applied in case these bounds give unrealistic results.

Line 157: Consider using a different word from “significant” to avoid conflation with statistical significance

ESM, Line 186: Change wording to “Due to the presence of these many unknowns...”

ESM, Line 278: Missing word, change to “the stiffnesses of the ligaments...”

Reviewer: 2

Comments to the Author(s)

The authors present a method for estimating the optimal walking speed of *Tyrannosaurus rex*. They claim that minimal metabolic expenditure should occur when step frequency matches tail resonant frequency. They reconstruct tail morphology of a *T. rex* specimen to estimate inertial parameters, lengths of tail segments, positions of ligaments, and axes of rotation. They perform a sensitivity analysis by shifting each of these elements individually from baseline. By approximating each segment as a rigid body attached to the next segment by a torsional spring, they calculate the resonant frequency of the tail. Using trackways, they then estimate the walking speed corresponding to this frequency, and claim that it is the optimal walking frequency.

The tail reconstruction is detailed and informative. The sensitivity analysis leads to some interesting insights, for example that the distal tail has pronounced effects on the resonant frequency depending on the location of the rotational axis.

The paper provides a method to model tail oscillations in extinct dinosaurs, and another method to estimate walking speeds. Despite these strengths, some major issues should be addressed before publication.

The basic premise- that optimal walking speeds correspond to resonant frequencies- is not well supported. Interaction effects are ignored in the sensitivity analysis. Key aspects of the methods and context for the discussion are relegated to supplemental information. Finally, while the biomechanical model seems sensible, the mathematical derivation involves ambiguous terminology and appears inconsistent as written.

If the authors can address these concerns, I believe this paper would be a useful addition to the literature.

Major revisions

--

Lines 39-41: "Optimal locomotor speeds are to a large extent the result of the natural frequencies of relevant body segments, through a combination of resonance and elastic storage (9-12)."

This is a bold claim, and deserves a little more discussion than simply pointing to references 9-12. What is the specific evidence that natural frequencies determine optimal locomotor speeds? Some examples could be provided.

As it stands, I am not fully convinced that all the references support the claim. For example, (10) show that work is minimized at natural leg swing frequency, but metabolic cost is minimized at a higher frequency, and do not show whether this is the preferred or optimal frequency in walking.

--

Lines 51-52: "Due to resonance, elastic storage would be maximal if step frequency matched the natural frequency of the tail."

Lines 69-70: "Elastic storage in the caudal ligaments would reduce the energy losses at each footfall, thereby reducing MCOT."

Lines 73-75: "Therefore, it would be beneficial to maximize tail amplitude (which occurs when step and natural frequency are matched), as this maximizes ligament strain and minimizes MCOT."

Maximal ligament strain does not necessarily mean minimal metabolic cost. If this were true, humans would stretch Achilles tendon maximally during walking / running, regardless of speed! Damping, active muscle volume, muscle activation all seem to contribute to metabolic cost and call for submaximal tendon/ligament strain (e.g. Collins et al. 2015 doi:10.1038/nature14288; Orselli et al. 2017 doi:10.1016/j.jbiomech.2017.06.022).

--

Lines 115-121: step length from trackways.

The authors already point to issues with substrate affecting walking speed. They claim that the walking speed reported, as determined from this step length is "energetically optimal". How do they know that the trackways were for a preferred (let alone optimal) step length?

Related: Lines 208-210: "DS has provided interesting qualitative observations. Bipedal dinosaurs seem to have preferred to walk at a relative stride length of 1.3..."

If hip height is known for the author's specimen, could they not simply use $(\text{hip height})^{1.3}$ to get preferred stride length? How does this method compare to the trackway measurement?

--

The authors have not looked at interaction effects in the sensitivity analysis. I appreciate that this adds a combinatorial level of complexity (and time!) to the analysis, however I do think it is important.

At the very least, I would ask the authors to combine all the parameters that have the greatest increase and decrease in step frequency (e.g. decrease length + max. moment arms + ventral axes + low strain to test whether they combine to increase frequency further. Opposite parameters for decreased frequency). That way, we can at least have an idea of whether these effects are additive.

This minimal level of testing interactions would be sufficient in my view, but will leave room for doubt about whether resonance uncertainty is higher than reported.

If the authors have the time and motivation to look at all combinations, that would be ideal. I think it would make a much stronger paper, put to rest any doubt about untested combinations, and set a much more reliable measure of estimate uncertainty.

I would also note that computational time does not seem to be an issue; their matlab script calculates tail natural frequencies in fractions of a second. There may be some technical hassle in writing the code to iterate through the different combinations, but it should be doable with some nested loops.

--

Re: novelty of the method

Lines 176-177: Until now, estimates for (submaximal) dinosaur walking speeds are all based on Dynamic Similarity (DS).

Lines 212: "The possibility of lower step frequencies than predicted by the equation has only been suggested in passing (14,38). Until now, no steps have been taken to explore this."

Lines 223-225: "Deriving walking speed using Dynamic Similarity has its limitations. However, until now it was the only approach used to estimate (non-optimal) dinosaur walking speeds."

Dynamic Similarity is not the only approach used (though it is most common), the possibility of it being wrong has been explored (a little), and natural frequencies of body segments have been used before (though not the tail as far as I know).

For (extant) avian dinosaurs, much has been done, though I assume the authors meant extinct dinosaurs. They might be particularly interested in Marmol-Guijarro et al. (2020 doi:10.1186/s12983-020-00363-z), who question the use of dynamic similarity by applying it to ptarmigan tracks.

Sellers et al. (2013 doi: 10.1371/journal.pone.0078733) used a full musculoskeletal model to find metabolically optimal walking speeds in a sauropod.

Christian et al. (1999 doi:10.1002/mmng.1999.4860020105) investigated an alternative method from DS, similar to the "Natural Frequency Method" proposed in the submitted paper. They

investigated whether the natural pendular period of limbs in giraffe and elephants was predictive of their step (or swing) frequency. Results were mixed, and they show that that dynamic similarity is a better predictor of walking speed from trackways in elephants than using the natural frequency of the limbs.

While they caution that natural pendulum frequencies won't give precise estimates in general, they do use natural pendulum frequency of limbs to estimate "comfortable walking speeds" of four species of dinosaurs.

I would suggest the authors preface their claim of novelty, with reference to this previous work.

--

Some discussion points are confusing without referencing the supplemental information. For example, in the main text there is no mention of how distal rotational axes are changed, or why only the 3rd and 4th segments are altered. Yet this is a key point relevant to the second discussion paragraph.

Similarly, the authors suggest an intriguing insight regarding the function of chevrons on lines 159-160, but the role of the chevrons in the model is lost to any reader who has not referenced the supplement.

Other points in the discussion are similarly confusing. I would encourage the authors to move enough of the supplement to the main text that the key discussion points are understandable without referencing the supplement.

--

The authors appear to be using a small-angle approximation in their derivations in the supplemental information. Is the small-angle approximation justified? What are the maximum relative angles between joints experienced during resonance?

--

While the geometry and physical principles of the biomechanical model are straightforward, some of the mathematics in the supplemental information are confusing, ambiguous or inconsistent. The key issues are

1. The conflation of $\Delta \epsilon$ with ϵ
2. The ambiguity of the meaning of ϕ , and related issues:
 - a. the possible conflation with $\Delta \phi$
 - b. the introduction of an undefined variable ϕ_e in equation 6
 - c. The ambiguity about rest angles
3. The introduction of R_s without its explicit connection to r

Below, I expand on these issues in turn:

1: Conflation of $\Delta \epsilon$ with ϵ

Between equations 5 and 6, the expression $R_s(\phi - \phi_e)$ is substituted for ϵ . Assuming that $(\phi - \phi_e) = \Delta \phi$ (but see point 2), the expression is still problematic since $R_s \Delta \phi = \Delta \epsilon$ by equation 2 and not ϵ .

2: Ambiguity of ϕ

On line 336, ϕ is defined as “the angle that a segment makes with the horizontal axis”. This definition is unambiguous on its own. However, only relative angles between segments matter to the torque from a rotational spring (as the authors note on line 338). In particular, the relative angles should deviate from some “resting” relative angle, where there is no spring torque between segments.

The authors introduce a “rest angle” in equation 8. However, here the expression is confusing. If the interspinous ligaments “do not generate any... force” (Line 265), then the strain should be zero by equation 4. However, in equation 8 the strain is $\epsilon_{hor} = 0.04$.

I believe the authors mean $\Delta \phi$ to be a proper “relative angle” from a true resting angle (which is itself a difference between absolute joint angles of adjacent segments). This would make sense given equation (2) and the use of $\Delta \phi$ in equation 6 (besides the issues raised in point 1). But then $\phi - \phi_e = \Delta \phi$, and what is ϕ_e ? Is it different from ϕ_r ? If the authors meant to use ϕ_r instead of ϕ_e , then there is still the issue that ϕ_r is not truly a rest angle (i.e. a relative angle at which there is no torque from the spring).

3: R_s vs r/L_0

As discussed in points 1 and 3, there is ambiguity between the terms ϕ , $\Delta \phi$, and ϕ_r , as well as $\Delta \epsilon$ and ϵ . This leads to ambiguity about R_s , which I hope would be resolved if the authors address points 1 and 2 above.

There is yet another (perhaps more minor) issue with R_s : its dependence on the moment arm r is hidden in the expression 2. Since the cubic relationship between r and T_{lig} is a major point of the paper, it would make more sense to replace R_s with its expanded form using r , or at least define it in terms of r earlier.

In its current form ($R_s = \Delta \epsilon / \Delta \phi$), or in the form that seems more in line with the useage of R_s in equations 6 and 8 ($R_s = \epsilon / \Delta \phi$) I believe $R_s = r/L_0$. Simply stating this relationship explicitly would greatly clarify the expressions in the paper.

--

Suppl Line 381: “...stiffness and damping of the tail would be mainly dependent on the ligaments.”

I suspect the intervertebral discs would supply more damping. Can the authors comment on this, or even supply an estimate?

Minor revisions / typos

Main Manuscript

--

As the authors have the code to solve 2D dynamics, I would like to know what the tail looks like when oscillating at its natural resonant frequency. Can the authors produce animations of the tail for the different parameter combinations? This would provide another means of assessing whether the biomechanical predictions appear sensible.

--

Line 167: “There are no extant analogues for tail-dependent bipedal locomotion...”

Lizards and some pangolins may be modern extant analogues. See Persons and Curry (2017 doi: 10.1016/j.jtbi.2017.02.032) and its relevant references.

--

Lines 182-183: "Such trackway estimates are not strictly optimal, but on average they should tend towards OWS if the sample of trackways is large enough."

I suggest: "... they should tend towards preferred walking speed (and presumably OWS), if the sample..."

--

Line 188: "This is true for dinosaurs, but even for"

Suggest: "This is true not only for dinosaurs, but even for..."

--

Supplemental material

Line 141: "...tail is in extension of flexion"

Suggest: "... tail is in extension or flexion"

--

Line 303: "Stiffness of the ligaments were recalculated inverse dynamically..."

Suggest: "Stiffness of the ligaments were recalculated using inverse dynamics..."

--

Line 348: "... Segdyn rearranges all the unknowns..."

I believe Segdyn is the authors' custom code, rather than some proprietary software. The authors should make this clear in the text.

--

Figure S7a: the black ligament outline is hard to see. I would suggest a different (lighter) colour.

===PREPARING YOUR MANUSCRIPT===

===PREPARING YOUR REVISION IN SCHOLARONE===

-- Ensure that your data access statement meets the requirements at <https://royalsociety.org/journals/authors/author-guidelines/#data>. You should ensure that you cite the dataset in your reference list. If you have deposited data etc in the Dryad repository, please include both the 'For publication' link and 'For review' link at this stage.

-- If you have uploaded ESM files, please ensure you follow the guidance at <https://royalsociety.org/journals/authors/author-guidelines/#supplementary-material> to include a suitable title and informative caption. An example of appropriate titling and captioning may be found at https://figshare.com/articles/Table_S2_from_Is_there_a_trade-off_between_peak_performance_and_performance_breadth_across_temperatures_for_aerobic_sc_ope_in_teleost_fishes_/3843624.

Author's Response to Decision Letter for (RSOS-201441.R0)

See Appendix A.

RSOS-201441.R1 (Revision)

Review form: Reviewer 1

Is the manuscript scientifically sound in its present form?

Yes

Are the interpretations and conclusions justified by the results?

Yes

Is the language acceptable?

Yes

Do you have any ethical concerns with this paper?

No

Have you any concerns about statistical analyses in this paper?

No

Recommendation?

Accept as is

Comments to the Author(s)

I appreciate the authors thoughtful and thorough responses to my comments regarding the initial manuscript. My main concern was that statements regarding the matching of tail resonance

indicating optimal/preferred walking speed (in conjunction with estimations of step length) were overly strong, given complexities and nuances of cost tradeoffs likely to occur. I believe the authors have thoroughly addressed this concern and others listed in the first round of comments. Furthermore, I really appreciate the addition of the animation video and the interactive parameter sensitivity spreadsheet. I feel that both of these items will provide a lot of insight and satisfaction to interested readers.

One note of relevance: the authors reference figures and tables in the main manuscript. However, I am unable to locate any such display items associated with the main manuscript. As such, I am unable to evaluate any modifications or changes made.

%% Later addition by the reviewer via email:

The figures and table all look good to me, although I did notice a small typo in the Table 1 caption. At line 37 of the document "Figure & table captions.docx", it should say "cannot be combined with each other", but the word "be" is missing.

Review form: Reviewer 2

Is the manuscript scientifically sound in its present form?

Yes

Are the interpretations and conclusions justified by the results?

Yes

Is the language acceptable?

Yes

Do you have any ethical concerns with this paper?

No

Have you any concerns about statistical analyses in this paper?

No

Recommendation?

Major revision is needed (please make suggestions in comments)

Comments to the Author(s)

Overall, the authors' changes greatly improve the manuscript in clarity, legibility and rigor. They have adequately addressed most of my concerns.

Further, the improved clarity of the discussion highlights important insights into the effects of the tail anatomy on resonance, as well as motivating some exciting avenues of further research.

There are a few minor issues that should not prevent acceptance of the manuscript for publication, but which I include in the final section below as recommendations.

I'd like to thank the authors for clarifying their approach in their supplementary material. Now that it is more clear, there remains one sticking point for me, which I will expand on in the next section.

Major revision

The authors define a mapping from joint angle to strain as $\epsilon = R_S(\phi - \phi_e)$, where R_S is a constant. They then say that “The equation for RS will provide an adequate approximation as long as the relative joint angles do not deviate far from horizontal posture.”

However, this mapping only appears appropriate when joint angles are close to extended posture. It assumes that the change in strain from extended posture (where strain is zero) to some ϕ is equal to change in strain around ϕ . But there is no reason to believe this a priori unless ϕ is close to ϕ_e . Moreover, moment arm should modify the relationship between strain and joint angle, even close to extended posture- so it seems odd that the moment arm does not appear in the definition of R_S .

Using Taylor expansion, and assuming moment arm remains constant, the relationship between strain (ϵ) and joint angle near horizontal posture (ϕ_h) should be $\epsilon(\phi) \approx \epsilon(\phi_h) + r/L_0^*(\phi - \phi_h)$. While this comes from a quick derivation, and I could have made an error, we would at least expect ϕ_h (or π) to appear in the definition of R_S , but it doesn't.

Can the authors either

1. Explain what I'm misunderstanding, and why their mapping is correct close to ϕ_h , or
2. Use a more carefully derived mapping appropriate for angles close to ϕ_h ?

The authors assume constant moment arms through the motion. This is acceptable, and should be clearly stated. In principle R_S should have a clearly defined basis in the geometry of their model, and if moment arms are constant I suspect that it is relatively simple.

The authors have added simply stated that RS has “the property” of being proportional to moment arm length, but it is not immediately clear why. I think this relationship to model geometry should be explicitly stated. As it stands, the physical basis- and appropriateness- of R_S is not clear.

Minor changes

Main Manuscript

Lines 22-23: “determined tail natural frequency of *T. rex* (0.66 s⁻¹), and the corresponding walking speed (1.28 m s⁻¹)...”

The authors have removed uncertainty ranges in this version. I appreciate the authors' hesitation here. They contend that their lowest and highest estimates for natural frequency are unrealistic, since dorsal and ventral axes of the tail would not be static during locomotion.

Even so, I contend that it's better to be conservative with the estimate and state the lowest and highest frequencies and speeds found during the sensitivity analysis (e.g. “... walking speed (1.28 m/s, range 0.8-1.64)”). This avoids the perception (for the surprising number of readers who only get as far as the abstract) that these are extremely precise estimates.

Even with these conservative uncertainty estimates, the authors ranges are surprisingly precise, especially given the previous estimates of walking speeds of 2 m/s or more for large theropods.

-

Lines 132-134: “However, since ligaments (like tendons) display relatively little energy loss during work-loop experiments [41,46], little damping is present. Therefore, we will use the undamped natural frequency...”

In the response document, the authors instead mention “... a structure-based estimate of damping in the tail is difficult, if not impossible.”

To me, the latter is a more valid reason to ignore damping, as I question that the ligaments would be the dominant source of damping. More likely, all the “squishy stuff” in the tail will provide more damping.

I recommend justifying ignoring damping in the main manuscript with something more similar to the second statement.

-

Lines 41-43: “Both fore- and hindlimb natural frequencies scale similarly with several morphological parameters within several mammalian groups [16,17], suggesting that resonance is beneficial at any body size.”

The sentence is vague, and scaling similarity does not automatically imply “benefit”. I recommend removing the sentence. The authors make their case for estimating PWS at resonance with the other surrounding sentences.

-

Line 178-179: “This corresponds with a preferred walking speed...”.

I recommend: “This corresponds to a preferred walking speed...”

Caption, Table 1: “The moment arms and alternative axes models are not mathematically predictable, and these cannot combined with each other.”

This is confusing and not necessary in the caption. I recommend removing the sentence.

Supplemental text

“Attachment sites visible on the caudal vertebrae were used to reconstruct the CSAs of the interspinous ligaments”

Define CSA (and all acronyms) at first use

-

“Magnitude of ligament strain in horizontal posture was varied by 25% to account for the possible variation in this number, corresponding with strain experienced by low- and high-stress tendons”

Better to say “Magnitude of ligament strain in horizontal posture was also imposed to be 3% and 5%, corresponding with...”

Percentages of percentages are confusing.

Decision letter (RSOS-201441.R1)

Dear Mr van Bijlert

The Editors assigned to your paper RSOS-201441.R1 "Natural Frequency Method: estimating the preferred walking speed of *Tyrannosaurus rex* based on tail natural frequency" have now received comments from reviewers and would like you to revise the paper in accordance with the reviewer comments and any comments from the Editors. Please note this decision does not guarantee eventual acceptance.

Please submit your revised manuscript and required files (see below) no later than 21 days from today's (ie 26-Jan-2021) date. Note: the ScholarOne system will 'lock' if submission of the revision is attempted 21 or more days after the deadline. If you do not think you will be able to meet this deadline please contact the editorial office immediately.

on behalf of Dr Manoj Srinivasan (Associate Editor) and Kevin Padian (Subject Editor)
openscience@royalsociety.org

Associate Editor Comments to Author (Dr Manoj Srinivasan):
Both reviewers agree that the authors have carefully addressed their comments/suggestions and one of them has provided some more comments to be addressed, mostly minor except perhaps

one more substantial one. The reviewer and I agree that these should be addressable, so we kindly invite the authors to revise and resubmit.

Reviewer comments to Author:

Reviewer: 1

Comments to the Author(s)

I appreciate the authors thoughtful and thorough responses to my comments regarding the initial manuscript. My main concern was that statements regarding the matching of tail resonance indicating optimal/preferred walking speed (in conjunction with estimations of step length) were overly strong, given complexities and nuances of cost tradeoffs likely to occur. I believe the authors have thoroughly addressed this concern and others listed in the first round of comments. Furthermore, I really appreciate the addition of the animation video and the interactive parameter sensitivity spreadsheet. I feel that both of these items will provide a lot of insight and satisfaction to interested readers.

One note of relevance: the authors reference figures and tables in the main manuscript. However, I am unable to locate any such display items associated with the main manuscript. As such, I am unable to evaluate any modifications or changes made.

%% Later addition by the reviewer via email:

The figures and table all look good to me, although I did notice a small typo in the Table 1 caption. At line 37 of the document "Figure & table captions.docx", it should say "cannot be combined with each other", but the word "be" is missing.

Reviewer: 2

Comments to the Author(s)

Overall, the authors' changes greatly improve the manuscript in clarity, legibility and rigor. They have adequately addressed most of my concerns.

Further, the improved clarity of the discussion highlights important insights into the effects of the tail anatomy on resonance, as well as motivating some exciting avenues of further research.

There are a few minor issues that should not prevent acceptance of the manuscript for publication, but which I include in the final section below as recommendations.

I'd like to thank the authors for clarifying their approach in their supplementary material. Now that it is more clear, there remains one sticking point for me, which I will expand on in the next section.

Major revision

The authors define a mapping from joint angle to strain as $\epsilon = R_S(\phi - \phi_e)$, where R_S is a constant. They then say that "The equation for RS will provide an adequate approximation as long as the relative joint angles do not deviate far from horizontal posture."

However, this mapping only appears appropriate when joint angles are close to extended posture. It assumes that the change in strain from extended posture (where strain is zero) to some ϕ is equal to change in strain around ϕ_e . But there is no reason to believe this a priori unless ϕ is close to ϕ_e . Moreover, moment arm should modify the relationship between strain and joint angle, even close to extended posture- so it seems odd that the moment arm does not appear in the definition of R_S .

Using Taylor expansion, and assuming moment arm remains constant, the relationship between strain (ϵ) and joint angle near horizontal posture (ϕ_h) should be $\epsilon(\phi) \approx \epsilon(\phi_h) + r/L_0^*(\phi - \phi_h)$. While this comes from a quick derivation, and I could have made an error, we would at least expect ϕ_h (or π) to appear in the definition of R_S , but it doesn't.

Can the authors either

1. Explain what I'm misunderstanding, and why their mapping is correct close to ϕ_h , or
2. Use a more carefully derived mapping appropriate for angles close to ϕ_h ?

The authors assume constant moment arms through the motion. This is acceptable, and should be clearly stated. In principle R_S should have a clearly defined basis in the geometry of their model, and if moment arms are constant I suspect that it is relatively simple.

The authors have simply stated that R_S has "the property" of being proportional to moment arm length, but it is not immediately clear why. I think this relationship to model geometry should be explicitly stated. As it stands, the physical basis- and appropriateness- of R_S is not clear.

Minor changes

Main Manuscript

Lines 22-23: "determined tail natural frequency of *T. rex* (0.66 s⁻¹), and the corresponding walking speed (1.28 m s⁻¹)..."

The authors have removed uncertainty ranges in this version. I appreciate the authors' hesitation here. They contend that their lowest and highest estimates for natural frequency are unrealistic, since dorsal and ventral axes of the tail would not be static during locomotion.

Even so, I contend that it's better to be conservative with the estimate and state the lowest and highest frequencies and speeds found during the sensitivity analysis (e.g. "... walking speed (1.28 m/s, range 0.8-1.64)"). This avoids the perception (for the surprising number of readers who only get as far as the abstract) that these are extremely precise estimates.

Even with these conservative uncertainty estimates, the authors ranges are surprisingly precise, especially given the previous estimates of walking speeds of 2 m/s or more for large theropods.

-

Lines 132-134: "However, since ligaments (like tendons) display relatively little energy loss during work-loop experiments [41,46], little damping is present. Therefore, we will use the undamped natural frequency..."

In the response document, the authors instead mention "... a structure-based estimate of damping in the tail is difficult, if not impossible."

To me, the latter is a more valid reason to ignore damping, as I question that the ligaments would be the dominant source of damping. More likely, all the "squishy stuff" in the tail will provide more damping.

I recommend justifying ignoring damping in the main manuscript with something more similar to the second statement.

-
 Lines 41-43: “Both fore- and hindlimb natural frequencies scale similarly with several morphological parameters within several mammalian groups [16,17], suggesting that resonance is beneficial at any body size.”

The sentence is vague, and scaling similarity does not automatically imply “benefit”. I recommend removing the sentence. The authors make their case for estimating PWS at resonance with the other surrounding sentences.

-
 Line 178-179: “This corresponds with a preferred walking speed...”.

I recommend: “This corresponds to a preferred walking speed...”

Caption, Table 1: “The moment arms and alternative axes models are not mathematically predictable, and these cannot combined with each other.”

This is confusing and not necessary in the caption. I recommend removing the sentence.

Supplemental text

“Attachment sites visible on the caudal vertebrae were used to reconstruct the CSAs of the interspinous ligaments”

Define CSA (and all acronyms) at first use

-
 “Magnitude of ligament strain in horizontal posture was varied by 25% to account for the possible variation in this number, corresponding with strain experienced by low- and high-stress tendons”

Better to say “Magnitude of ligament strain in horizontal posture was also imposed to be 3% and 5%, corresponding with...”

Percentages of percentages are confusing.

===PREPARING YOUR MANUSCRIPT===

Your revised paper should include the changes requested by the referees and Editors of your manuscript. You should provide two versions of this manuscript and both versions must be provided in an editable format:
 one version identifying all the changes that have been made (for instance, in coloured highlight, in bold text, or tracked changes);
 a 'clean' version of the new manuscript that incorporates the changes made, but does not highlight them. This version will be used for typesetting if your manuscript is accepted.
 Please ensure that any equations included in the paper are editable text and not embedded images.

===PREPARING YOUR REVISION IN SCHOLARONE===

<https://royalsociety.org/journals/authors/author-guidelines/#data>. You should ensure that

you cite the dataset in your reference list. If you have deposited data etc in the Dryad repository, please include both the 'For publication' link and 'For review' link at this stage.

Author's Response to Decision Letter for (RSOS-201441.R1)

See Appendix B.

RSOS-201441.R2 (Revision)

Review form: Reviewer 2

Is the manuscript scientifically sound in its present form?

Yes

Are the interpretations and conclusions justified by the results?

Yes

Is the language acceptable?

Yes

Do you have any ethical concerns with this paper?

No

Have you any concerns about statistical analyses in this paper?

No

Recommendation?

Accept as is

Comments to the Author(s)

The authors have satisfactorily addressed my concerns. The model design, assumptions and rationale are now very clear, as is the origin and usefulness of the Rs term. The table of contents in the ESM is a nice touch.

Decision letter (RSOS-201441.R2)

Dear Mr van Bijlert,

It is a pleasure to accept your manuscript entitled "Natural Frequency Method: estimating the preferred walking speed of *Tyrannosaurus rex* based on tail natural frequency" in its current form for publication in Royal Society Open Science. The comments of the reviewer(s) who reviewed your manuscript are included at the foot of this letter.

on behalf of Dr Manoj Srinivasan (Associate Editor) and Kevin Padian (Subject Editor)
openscience@royalsociety.org

Reviewer comments to Author:

Reviewer: 2

Comments to the Author(s)

The authors have satisfactorily addressed my concerns. The model design, assumptions and rationale are now very clear, as is the origin and usefulness of the Rs term. The table of contents in the ESM is a nice touch.

Appendix A

Structure of this document

We have copied comments from the associate editor and reviewers verbatim, and they are marked in red. Below each comment, we first type our response (captioned with “**Response**”). If the comments and response have led to a substantial change in the manuscript wording, we have also copied the new sections from the text (captioned “**Changes to the manuscript**”), in which case the reference numbering (in square brackets) follows the numbering from the main manuscript. Although they were identical, we have supplied a list of references cited in this response to both reviewers individually, because we were unsure whether they will be provided with the entire response letter, or only the sections pertaining to their individual comments.

The response to reviewer 1 starts on page 2. The response to reviewer 2 starts on page 19.

Associate Editor Comments to Author (Dr Manoj Srinivasan):

Associate Editor: 1

Comments to the Author:

The reviewers have provided detailed remarks and we invite a revised manuscript that addresses these issues.

I agree with the reviewers that it may not be possible to rigorously claim that the walking speeds are determined by the natural frequencies of the body segments. Superficially, this presumes that swinging body limbs is the primary or dominant energy cost term (assuming animals prefer low effort gaits). For instance, in humans, the walking step frequencies are chosen to be systematically faster than the natural frequency of the swing leg, thought to be because this faster swing lowers other components of the metabolic cost.

Kuo, Arthur D. "A simple model of bipedal walking predicts the preferred speed–step length relationship." J. Biomech. Eng. 123.3 (2001): 264-269.

Response

The issues brought up by the associate editor are similar to issues brought up by the reviewers, which we will address in more detail in our responses to the reviewers’ comments. While perhaps our wording was too strong in the first version of our manuscript, there is a large body of literature that shows that optimal/preferred locomotor frequencies & natural frequencies of relevant body parts are closely related in many animals, and that they can tune their frequencies to benefit from resonance (Ahlborn and Blake 2002; Ahlborn, Blake, and Megill 2006; Christian et al. 1999; Holt, Hamill, and Andres 1990, 1991; Huat et al. 2004; Lee and Krovi 2008; Wagenaar and Van Emmerik 2000). In response to the associate editor mentioning that our claims may be too rigorous, we have opted to change instances of “optimal walking speed” to “preferred walking speed”, unless referencing a study where energetics were directly measured or estimated. This has also changed the title of our manuscript. We further would like to point out that reconstructing the gait of animals in deep time is a process that inevitably will come with many uncertainties. There are currently only a few methods available, each with their own caveats, assumptions and downsides. Nevertheless, each method has its merits, and we add an independent line of reasoning to the palette of available tools for gait reconstruction of extinct animals.

Reviewer 1

Reviewer comments to Author:

Reviewer: 1

Comments to the Author(s)

SUMMARY:

The authors propose a new method (called the Natural Frequency Method) for estimating walking speeds of extinct animal locomotion and use the T. rex as an example. Using 3D volumetric reconstructions, they estimate input parameters of a biomechanical model that considers the tail as a series of body segments connected by torsional springs. The fundamental resonance of the system is determined and multiplied by step length of a preserved trackway in order to determine optimal walking speed. This method distinguishes itself as an alternative to extrapolating allometric scaling relationships and can provide insights on the energetics of foraging behaviors. Overall, I like this approach and appreciate the focus on mechanistic locomotion energetics. I have provided some comments regarding nuances to the resonance-matching argument outlined here and hope the authors find them useful.

Response

We are pleased that the main message of the manuscript was clear to the reviewer. We thank the reviewer for the encouraging words, and the insightful comments that helped to improve our manuscript. We have responded to each individual comment below, and where applicable, also included the changes to the manuscript below our response.

We also mention to the reviewer that, in response to comments from the associate editor, we have opted to change instances of “optimal walking speed” to “preferred walking speed”, unless referencing a study where energetics were directly measured or estimated. This has also changed the title of our manuscript.

GENERAL COMMENTS:

- The authors argue that metabolic cost of transport should be minimized when strain energy is maximized in the interspinous ligaments – by using a step frequency matched to the fundamental resonance of the tail structure. The authors reference a study that showed the energetic cost of human load carriage is benefited from a compliant suspension backpack (Rome et al., 2006). However, evidence on the energetic advantages of such devices are mixed overall. Kram (1991) and Foissac et al. (2009) found increases in metabolic cost while Castillo et al. (2014) found a decrease in cost. A recent optimization study has shown that the ratio between step frequency and resonance can help explain the different results (Schroeder et al., 2019).

The authors of the current manuscript acknowledge that out-of-phase oscillations are desired to maximize strain energy, however this phase relationship does not necessarily occur when step frequency matches the resonant frequency. For a simple spring mass model stimulated by a sinusoidal forcing function, phase is 90 degrees when frequency equals resonance. At frequencies below resonance, phase is 0 degrees (in phase) and at frequencies above resonance, phase is 180 degrees

(out of phase). This transition is impulsive when no damping is present. The more damping that is present, the more gradual the phase transition is (Castillo et al., 2014; Schroeder et al., 2019).

Response

We agree with the reviewer's assessment, including the insights based on Schroeder et al. (2019), although we maintain that strain energy is maximized when strain is maximized, which occurs during resonance.

Partly based on the reviewers' comments, we have realized that our description of the relation between step frequency on the one hand and elastic storage and metabolic cost of transport on the other hand was overly simplified. Indeed, implying that a passive tail would resonate at the optimal step frequency, concurrently implies a phase difference of 90 degrees between the vertical hip and tail oscillations. Although this would maximize strain energy (and therefore elastic storage in the tail), this may not result in minimal MCOT, because the vertical hip and tail oscillations are not out of phase. In this hypothetical scenario, the only input to the passive tail is the vertical motion of the hip, which enforces the phase difference. In reality, the CFL contractions inherent to walking would also affect the phase of the tail oscillations, and other tail muscles might affect the phase relationship with relatively low muscle forces, due to their mechanical advantage about the base of the tail.

We have decided to build our argument mainly on the observation that step frequency of many animals is close to the natural frequency of important body parts, which is well supported in the literature (Ahlborn and Blake 2002; Ahlborn et al. 2006; Christian et al. 1999; Holt et al. 1990, 1991; Huat et al. 2004; Lee and Krovi 2008; Wagenaar and Van Emmerik 2000). There is of course considerable scope to tune the frequencies depending on task requirements, and there are very interesting trade-offs modulated by contraction- and activation dynamics, tendon elasticity, foot collisions, etc. However, these would only become visible if we increase complexity of our model by adding legs and musculature. We have therefore moved all the subtleties regarding phase relationships to the discussion, because although they are important, their inclusion would require a more complex model with legs and muscle actuation, which goes beyond the scope of this article.

We have expanded our basic premise into several sub-statements, and for each of these statements provided more literature to support it. Subsequently, we combine them to introduce the reasoning behind our method. This has led to the following section, which has replaced the first paragraph of our main text.

Changes to the manuscript (expanded premise, line 32-49)

Animals display a variety of gaits and walking speeds. Two locomotor extremes are of particular interest: the maximal speed, and the energetically optimal walking speed, at which the metabolic cost of transport (MCOT, in $\text{J m}^{-1} \text{kg}^{-1}$) is minimal for walking gait. In the absence of external task constraints, optimal walking speed is very close to preferred walking speed in humans [1,2], ratites [3], horses [4] and elephants [5]. It is likely that they are related in all terrestrial animals, and that animals would tend to forage at this speed. During locomotion at preferred speed, animals tend to make use of resonance by matching locomotor frequencies to the (undamped) natural frequencies of their relevant body parts, sometimes tuned through muscular contractions [6–13]. Indeed, it has been demonstrated that mammalian quadrupedal taxa have convergent natural frequencies between fore- and hindlimbs despite differing limb morphologies [14,15]. Both fore- and hindlimb natural frequencies scale similarly with several morphological parameters within several mammalian

groups [16,17], suggesting that resonance is beneficial at any body size. An important advantage of moving a body part close to its natural frequency is that this reduces mechanical work [18,19]. Based on the commonly adopted idea that preferred walking speed is chosen to minimise MCOT [1–5], it has been argued that preferred step frequencies are close to the natural frequencies of relevant body parts [7,11,12]. Determination of the natural frequency of a body part relevant to locomotion can therefore constrain preferred (and presumably, optimal) step frequencies and walking speeds of extinct taxa [10].

Regardless, it is true that relative amplitude peaks at resonance. Thus, peak strain energy tends to occur at frequencies slightly above resonance, depending on how much damping is present. The optimization model from Schroeder et al. (2019) shows that cost for human load carriage is optimized in this frequency range, as did Castillo et al. (2014) when they found reduced cost. On the other hand, frequencies slightly below resonance can be quite costly (Foissac et al., 2009). Other simulation studies have shown that step frequencies much higher than resonance can still provide an energetic benefit (Ackerman & Seipel, 2014) and these predictions agree with Rome et al.'s findings (2006).

Response

Although this might be a semantic issue, we subtly disagree with the statement that peak strain energy tends to occur above resonance. Strain energy, as we would define it, is the amount of energy stored in a spring as it is strained (or alternatively: the amount work done on a spring to strain it). By definition, this would therefore be maximal when the amplitude of oscillation is maximal, because this would both maximise strain and maximise the force to reach this strain. As we interpret the literature that the reviewer is referencing, there are other interacting energetic cost terms, leading to step frequencies above resonance being optimal, despite there being less amplitude (and thus less strain energy) at these driving frequencies.

As we had mentioned in the supplementary information, adding damping to a second-order mass-spring system results in damped natural frequency that equals $f_n \cdot (1-\beta)^{0.5}$, with f_n being the undamped natural frequency, and β being the damping ratio, or relative damping in the system (Ruina and Pratap 2019). In a damped system, resonance is maximal at a driving frequency of $f_n \cdot (1-2\beta^2)^{0.5}$. We are unable to obtain a structure-based estimate of the damping ratio of the tail of *T. rex*. However, due to the relatively low energy losses in tendons (Alexander 2006; Biewener 2008), we do argue that damping is low (i.e., we expect the damping ratio to be smaller than 0.2). As a result, the differences between the resonant, damped and undamped natural frequency are negligible (Ruina and Pratap 2019). We now explicitly mention in the main manuscript that the resonant frequency is close to the undamped natural frequency at low damping ratios.

Changes (expanded methods line 130-135)

In the absence of damping, resonance of the tail would occur at the undamped natural frequency (f_n) of the system [45]. In reality, damping is omnipresent. In that case, resonance occurs at a forcing frequency of $f_n \cdot (1-2\beta^2)^{0.5}$, with β being the damping ratio [45]. However, since ligaments (like tendons) display relatively little energy loss during work-loop experiments [41,46], little damping is present. Therefore, we will use the undamped natural frequency of the biomechanical model as a proxy for the resonance frequency.

In short, current evidence suggests that locomotion energetics are benefited by interacting with oscillating loads when step frequency is greater than resonant frequency, not equal to. Perhaps this may help to explain why the authors' estimate of optimal walking speed is somewhat lower than the walking speeds predicted by alternative approaches and studies. It is unclear if the amplitude/phase

relationships of the spring mass translate to the multi rigid body model of the current manuscript. However, this could be investigated by imposing harmonic motion at the base of the tail.

*-It seems reasonable to assume that the dynamics of a large oscillating tail should play a prominent role in determining the energetics of an organism such as the *T. rex*. However, there are likely many aspects of the locomotory system that influence energetics as a function of frequency. In humans, the choice of step frequency illustrates a trade off between work to redirect the velocity of the center of mass during the step-to-step transition (high cost at low frequencies) and costs associated with swinging the leg (high cost at high frequencies) (Doke et al., 2005; Kuo, 2001; Schroeder et al., 2019). Because of this trade off, humans typically swing their legs at much higher frequencies (approximately 1.5 Hz at a moderate walking speed of 1 m/s; (Bertram & Ruina, 2001)) than the resonant frequency of the leg (0.64 Hz; (Doke et al., 2005)). Muscle activation costs have also been shown to increase with frequency (Doke & Kuo, 2007) and may play a role in determining optimal gait frequencies.*

Response

The reviewer suggests that our estimates of dinosaur optimal walking speed may be lower than earlier estimates because we do not consider the complex interactions that occur during locomotion. However, our most direct comparisons are with estimates based on dynamic similarity, which use simple inverted pendulum models of walking, and therefore also do not consider these interactions.

Overall, we agree with the reviewer's assessment that in reality there are many interacting trade-offs that would influence the tail's role in locomotor energetics. We also agree that phase relationships are more subtle than we suggested in version 1 of the manuscript. However, we do not agree that this necessarily implies that frequencies higher than the natural frequency are optimal. Unlike in a weighted carry, *T. rex* has several muscles with a significant mechanical advantage that could enforce a different phase relationship while still benefitting from resonance (including the CFL that would already be active during walking). This implies that there may be a possibility to invest energy to maintain a desirable phase relationship, which is returned in the form of energy-saving benefits of resonance.

As mentioned before, however, these considerations are beyond the scope of our method. Our method is intended as a reductionist method, based on what we consider to be an important observation in locomotion: during self-selected locomotion, animals tend to make use of resonance by matching locomotor frequencies to the natural frequencies of their body parts, sometimes tuned through muscular contractions. There is a considerable body of evidence, acquired both experimentally and through simulation studies, to support this (Ahlborn and Blake 2002; Ahlborn et al. 2006; Christian et al. 1999; Holt et al. 1990, 1991; Huat et al. 2004; Lee and Krovi 2008; Wagenaar and Van Emmerik 2000). Convergent natural frequencies between fore- and hindlimbs in a variety of taxa suggests there may even be an evolutionary selective pressure for locomotion that incorporates resonance (Kilbourne and Hoffman 2013, 2015; Myers and Steudel 1997; Raichlen 2004).

Within the context the aforementioned observation, the reviewer mentions that Doke et al. (2015) found a much lower leg natural frequency (0.64 Hz) than the preferred step frequency (1.5 Hz). We do not find this a helpful comparison, because (1) Doke et al. constrained the knee to remain fully extended, which leads to a higher inertia and thus lower natural frequency than when the leg is unimpeded, and (2) during a single step, the leg only swings forward, which represents half of a period. A more apt comparison would be based on the stride frequency (0.75 Hz), which is already much closer to the 0.64 Hz reported by Doke et al. Perhaps even more importantly, in our view Doke et al. never set out to investigate unimpeded locomotion, but other authors have. Wagenaar and Van

Emmerik (2000) have demonstrated that leg natural frequencies and preferred stride frequencies are remarkably close in humans, as have Holt et al. (1990, 1991) and more recently Russell and Apatoczky (2016).

In response to the reviewer's request to impose harmonic motion at the base of the tail: we now supply a video of a lightly damped tail, forced at its undamped natural frequency. As expected, this system shows a 90 degree-phase difference in vertical oscillations between hip and tail. However, as explained above, we expect that muscle actuation would change this phase relationship, without negating the benefits of resonance, so we explicitly mention in the manuscript that this phase relationship may not be representative of true walking gait.

Changes to the manuscript (expanded discussion on phase & frequency tuning, line 319-347)

Our method for walking speed estimation is meant to be a reductionist analysis, so we have only incorporated the interspinous ligaments, since we expect their mechanical effect to dominate overall tail dynamics. Natural frequencies play an important role in animal locomotion [6–13], and NFM therefore provides a reasonable starting point for investigating how the resonant properties of dinosaur tails affect overall locomotion. However, in reducing the analysis to the mechanics of a single structure, many of the complex interactions are ignored. For instance, animals tune the frequencies of their segments through muscular contractions [6,7]. *T. rex* could have contracted its epaxial and hypaxial caudal musculature to add rotational stiffness to the tail, which could be beneficial at higher speeds, for instance during a pursuit. Our ventral axes model demonstrates a possibility in this regard: if *T. rex* were to keep its distal tail in slight flexion, thereby migrating the rotational axes of the vertebrae to the articulations with the chevrons, the overall natural frequency of the tail would increase.

Tail musculature could also be employed to enforce beneficial phase relationships between vertical hip and tail oscillations. During walking, vertical oscillations between withers and heads in most ungulates are out of phase, which provides them with an energetic benefit [38–40,57]. This is theorised to be modulated by natural frequencies of the neck segment [39,40], but there is an active component to this behaviour as well [38]. This phase relationship reduces the losses incurred during the step-to-step transitions, which are identified as major sources of energy losses in inverted pendulum models of walking [58]. The implications of energy storage are also further complicated by the serial elasticity in musculotendon complexes: tendon stiffness can substantially impact the mechanical work done by the muscle fibres [59]. Finally, the cost of cyclical muscle contractions could also affect metabolic optima, depending on the task requirements [18,19]. Such complex interactions would undoubtedly affect PWS, but require extensive reconstructions of muscular contractile properties. Muscular parameters currently provide the biggest uncertainty in most musculoskeletal simulations of dinosaurs [43,51,60], so any investigation of these interactions would require careful consideration. Inclusion of our compliant tail model into fully-actuated hindlimb simulations of *T. rex* may shed further light on how the unique tails of non-avian dinosaurs functioned during locomotion. However, in such complex simulations, the intricacies of tail dynamics may be overshadowed by the uncertainties regarding the contractile properties of the muscles.

Changes to the manuscript (Video S1 mentioned in the results section, line 179-187)

To provide the reader with an impression of the dynamic behaviour that results, we have provided a simulation of a lightly damped version of the baseline model (video S1). In this simulation, the only input to the tail was a sinusoidal motion at the base of the tail with an amplitude of 0.08 m. Assuming pendular walking, this is the vertical motion that results from taking 1.94 m long steps at a hip height of 3.1 m. No muscle forces were included, so the phase relationship between the vertical

oscillations of the hip and tail may not be representative of a muscle-actuated tail. In this simulation, a small amount of damping was introduced in order to obtain a finite tail amplitude. Damping ratio of the fundamental eigenmode was 0.16, which has a negligible effect on the resonant frequency [45].

Ackerman, J., & Seipel, J. (2014). Short communication A model of human walking energetics with an elastically-suspended load. Journal of Biomechanics, 47(8), 1922–1927. <https://doi.org/10.1016/j.jbiomech.2014.03.016>

Castillo, E. R., Lieberman, G. M., McCarty, L. S., & Lieberman, D. E. (2014). Effects of pole compliance and step frequency on the biomechanics and economy of pole carrying during human walking. Journal of Applied Physiology, 117(5), 507–517. <https://doi.org/10.1152/jappphysiol.00119.2014>

Foissac, M., Millet, G. Y., Geysant, A., Freychat, P., & Belli, A. (2009). Characterization of the mechanical properties of backpacks and their influence on the energetics of walking. Journal of Biomechanics, 42(2), 125–130. <https://doi.org/10.1016/j.jbiomech.2008.10.012>

Kram, R. (1991). Carrying loads with springy poles. Journal of Applied Physiology (Bethesda, Md. : 1985), 71(3), 1119–1122. <http://www.ncbi.nlm.nih.gov/pubmed/1757307>

Kuo, A. D. (2001). A simple model of bipedal walking predicts the preferred speed-step length relationship. Journal of Biomechanical Engineering, 123(3), 264–269. <https://doi.org/10.1115/1.1372322>

Rome, L. C., Flynn, L., & Yoo, T. D. (2006). Rubber bands reduce the cost of carrying loads. Nature, 444(7122), 1023–1024. <https://doi.org/10.1038/nature4441023a>

Schroeder, R. T., Bertram, J. E. A., Nguyen, V. S., Hac, V. V., & Croft, J. L. (2019). Load carrying with flexible bamboo poles: optimization of a coupled oscillator system. <https://doi.org/10.1242/jeb.203760>

In short there may be many determinants of gait energetics besides just resonance and although the resonance of massive tail oscillations could play a large role, the authors should acknowledge that this is simply one mechanism that could influence optimal frequencies.

Response

We agree with the reviewer that a more nuanced formulation of the relation between resonant and preferred locomotor frequencies is warranted. Indeed, phase-forcing, frequency-tuning, mechanical work at the level of the muscle-fibres, and minimization of stance-phase losses are highly interesting topics that may be addressed using more sophisticated simulation models. Upon its inclusion in musculoskeletal simulation models, our multi-segment tail model will hopefully aid in the investigation of these topics in future research.

However, the exact phase relationships and kinematics of the tail, hip and legs, are not directly relevant to the main question asked in this study, nor to the novel method proposed here to arrive at an answer to this question. Our method is meant to be a reductionist method based on an empirical relationship between natural frequencies and locomotor tendencies of animals. Inclusion of more factors would require numerous additional assumptions (e.g. regarding muscularity and kinematics), which would not unambiguously increase confidence in the result, but more importantly, it would be counter-productive to the goal of developing a relatively simple method that explores an alternative line of reasoning to established methods.

Therefore, based on the reviewer's comments, we have moved all mention of phase relationships to the discussion. We have expanded upon how different cost parameters that are beyond the scope of our model may influence and interact, if our compliant tail model were to form the basis of a more elaborate model. We also no longer consider weighted-carry studies to be a good analogy for dinosaur tails, because unlike in those studies, dinosaurs could potentially enforce a beneficial phase relationship at low metabolic cost, given the mechanical advantage of the tail musculature. We therefore limit our comparison to ungulate neck-movements, which although theorised to be modulated by natural frequencies (Basu, Wilson, and Hutchinson 2019; Loscher et al. 2016), have an active component as well (Gellman and Bertram 2002).

Changes to the manuscript

See the expanded discussion above (line 319-347).

*Bertram, J. E. A., & Ruina, A. (2001). Multiple Walking Speed–frequency Relations are Predicted by Constrained Optimization. Journal of Theoretical Biology, 209(4), 445–453.
<https://doi.org/10.1006/jtbi.2001.2279>*

Doke, J., Donelan, J. M., & Kuo, A. D. (2005). Mechanics and energetics of swinging the human leg. Journal of Experimental Biology, 208(3), 439–445. <https://doi.org/10.1242/jeb.01408>

*Doke, J., & Kuo, A. D. (2007). Energetic cost of producing cyclic muscle force, rather than work, to swing the human leg. Journal of Experimental Biology, 210(13), 2390–2398.
<https://doi.org/10.1242/jeb.02782>*

*Kuo, A. D. (2001). A simple model of bipedal walking predicts the preferred speed-step length relationship. Journal of Biomechanical Engineering, 123(3), 264–269.
<https://doi.org/10.1115/1.1372322>*

*Schroeder, R. T., Bertram, J. E. A., Nguyen, V. S., Hac, V. V., & Croft, J. L. (2019). Load carrying with flexible bamboo poles: optimization of a coupled oscillator system.
<https://doi.org/10.1242/jeb.203760>*

-The authors reference evidence that various animals and humans tend to walk at the minimum of their respective metabolic cost of transport curves. Although this may be true in certain circumstances, locomotion speeds in an ecological context likely vary substantially. E.g. people in large cities tend to walk faster than those in small towns (Bornstein & Bornstein, 1976). Individuals move more slowly during short bouts of walking versus long durations (Seethapathi & Srinivasan, 2015). Surely, animals will move at higher speeds to escape or engage in predation, etc. Optimal foraging theory has been applied to determine appropriate movement vigor to show that movement speeds are chosen to match the rate of energy expenditure to the global capture rate of the sought-after reward (Shadmehr et al., 2019; Yoon et al., 2018). Other studies have shown that individuals adjust gait type and speed in response to sensitivities to time constraints and/or costs (Summerside et al., 2018; Tiew & Srinivasan, 2020)

Response

We agree that animals can choose different locomotor speeds depending on the situations at hand. However, our goal was to find the energetically optimal walking speed based on arguments of

mechanical nature. How external factors of non-mechanical nature affect locomotion are beyond the scope of our manuscript.

Our statement on line 37 regarding foraging costs references (Dececchi et al. 2020; Kane et al. 2016), where singular walking speeds for each taxon have been chosen for ecological comparisons. This is a less sophisticated ecological comparison than the research pointed out by the reviewer. Within the framework of a single-speed foraging costs comparison, it seems reasonable to state that one should choose the preferred/optimal walking speed of the respective taxa. This means speed estimates using our method can form a useful starting point for future, more elaborate, comparisons of dinosaur foraging costs. We have amended our statement regarding foraging costs to be worded less strongly.

Changes to the manuscript (reworded discussion, line 286-292)

DS does not incorporate MCOT, and therefore cannot be used to estimate preferred walking speed of dinosaurs. Therefore, when estimating dinosaur foraging costs, researchers have used DS to scale the walking speed [31], or even kept it at a constant 2 m s⁻¹ for larger taxa [33]. However, methods to calculate MCOT are heavily dependent on both speed and mass [30], so we suggest that any comparison of foraging costs should use the PWS of the respective taxa as a starting point. This could be done using the natural frequency of the legs [10], musculoskeletal simulations [51], or NFM as we propose it.

Given that high amplitude, out-of-phase oscillations of the tail should potentially benefit locomotion energetics at step frequencies greater than the resonant frequency, it may be more useful to think of the tail's fundamental resonance as a lower bound frequency for an animal to walk at. Conceptually, this allows an upper range of speeds that should be economical for the T. rex to walk at, depending on the task goal and other relevant circumstances that may affect chosen locomotion speeds.

Response

We agree, but note that we are unable to bracket the upper end of this frequency range without further assumptions regarding active musculature, mechanical advantages and locomotor kinematics. We set out to explore the preferred walking speed, as it relates to natural frequency. In the animal kingdom, there is in general a singular optimal speed for each gait (for instance: Hoyt and Taylor (1981)). In response to the reviewer, we have expanded the discussion. We now also include the frequency-tuning effects of added stiffness through muscular contraction, which may influence energy expenditure at speeds above the preferred speed.

Changes to the manuscript

Expanded discussion, lines 319-347, see our response to the comment above.

Bornstein, M. H., & Bornstein, H. G. (1976). The pace of life. Nature, 259(5544), 557–559. <https://doi.org/10.1038/259557a0>

Seethapathi, N., & Srinivasan, M. (2015). The metabolic cost of changing walking speeds is significant, implies lower optimal speeds for shorter distances, and increases daily energy estimates. Biology Letters, 11(9), 20150486. <https://doi.org/10.1098/rsbl.2015.0486>

Shadmehr, R., Reppert, T. R., Summerside, E. M., Yoon, T., & Ahmed, A. A. (2019). Movement Vigor as a Reflection of Subjective Economic Utility. In Trends in Neurosciences (Vol. 42, Issue 5, pp. 323–336). Elsevier Ltd. <https://doi.org/10.1016/j.tins.2019.02.003>

Summerside, E. M., Kram, R., & Ahmed, A. A. (2018). Contributions of metabolic and temporal costs to human gait selection. *Journal of The Royal Society Interface*, 15(143), 20180197. <https://doi.org/10.1098/rsif.2018.0197>

Tiew, H., & Srinivasan, M. (2020). Pre-crastination: Time deadlines increase walking speeds even when not constraining. *BioRxiv*, 2020.07.17.208140. <https://doi.org/10.1101/2020.07.17.208140>

Yoon, T., Geary, R. B., Ahmed, A. A., & Shadmehr, R. (2018). Control of movement vigor and decision making during foraging. *Proceedings of the National Academy of Sciences of the United States of America*, 115(44), E10476–E10485. <https://doi.org/10.1073/pnas.1812979115>

MINOR COMMENTS:

Line 39-40: As mentioned in previous comments, natural frequency may be a part of the equation but is unlikely to be the only thing that matters with regards to the determination of optimal walking speed. Perhaps these lines could be reworded slightly: "Optimal locomotor speeds may be influenced by the natural frequencies of relevant body segments...". Furthermore, Doke et al. (2005) is referenced here, however, resonant leg frequency reported in this study is very low (0.64 Hz) and does not predict preferred frequencies observed during optimal walking speed in humans.

Response

Doke et al. (2005) was referenced to support the claim that resonance leads to lower mechanical work requirements. This is supported by their result that metabolic power was minimal at a swinging frequency that was very close to the leg natural frequency in their experimental setup. We agree that the current wording and reference placement should be amended. We have split up the premise into smaller statements with references supporting each individual claim, and then combine them to propose our method.

Changes to the manuscript

Reworded premise (line 32-49, see above). In particular, specified why Doke et al. was referenced (line 43-44):

An important advantage of moving a body part close to its natural frequency is that this reduces mechanical work [18,19].

Figure 1: I like the biomechanics model used in this study. However, I do wonder how much of the model's complexity matters for the results on fundamental resonant frequencies. Would approximating the dynamics of the tail as a cantilever beam or even as a simple spring mass change the results by very much? I do appreciate the fact that tissue-specific outcomes (e.g. ligament rupture, Young's modulus, etc.) are more readily evaluated with the more complex model. However, I cannot help but wonder if the gross dynamics of the tail's oscillation could be approximated with a more accessible, reductionist modeling approach.

Response

We thank the reviewer for their kind words regarding our model. We agree that a reductionist approach is desirable, which is why we have reduced a structure built up of 88 skeletal elements and numerous passive and active structures down to 5 rigid bodies connected by torsional springs. We were therefore somewhat surprised by the suggestion that the model was too complex. Indeed, both reviewers have pointed that the model description seems to be quite complicated/confusing,

whereas we had thought that the process, as described, was relatively straightforward. Essentially, we are dividing the tail into 5 segments, and then fitting joint spring parameters using kinematic relationships measured in two different postures of the skeleton.

Since our overarching goal is a simple, reductionist method that is easily to adopt, both for frequency analysis and full-scale dynamic simulation models, we have re-evaluated our modelling steps to see at what points we could further streamline the process (and description). The changes carried through have resulted in natural frequencies that are slightly lower than those reported in version 1 of the manuscript. It has also slightly raised average Young's Modulus per segment, but none of the models show unrealistic/impossible values.

We have reworded the summary of our method in the main manuscript in a manner that we hope clarifies the method (and indeed shows how straightforward it is to implement). To this end, we have also added a second figure (figure 2) to the main manuscript.

The reviewer suggests a cantilever beam approach. This would at least involve assigning the entire tail a constant cross-section (and thus moment of area), and estimating the modulus of elasticity of the tail as a whole. Since these parameters would be very specific to the morphology and tissue parameters of any taxon, it in fact seems to be a more challenging approach than our method, which is contrary to the suggestion that a simpler approach is desirable.

It is challenging to define a model so that it simplifies the original complex structure, while still providing meaningful predictions. The reviewer suggests reducing our model even further to a simple mass on a spring. In our view, the sensitivity analysis indicates that even though most of the mass is concentrated in the first two-fifths of the tail, the comparatively light remainder still appreciably affects overall tail dynamics. Reducing the tail to 1 segment would therefore remove much of the biological signal we are hoping to explore, namely: how a compliant tail capable of elastic storage would influence locomotion. Partly based on this argument, we would consider a 1 degree-of-freedom mass-spring model to be an oversimplification, likely leading to unrealistic results.

Changes to the manuscript (expanded, simplified methods section, line 104-128)

Taking the morphological reconstruction as starting point, we then defined a simplified biomechanical model consisting of 5 rigid bodies, connected in hinge joints, with a nonlinear rotational spring at each joint (figure 1, figure 2 for angle definitions). During walking, the most prominent movements and forces occur in the sagittal plane, and the moment arm of the CFL is also largest in this plane. Furthermore, exchange between gravitational and elastic potential energy is most meaningfully studied in this plane. Therefore, similar to previous researchers [29,32], we elected to perform a sagittal plane analysis. Inertial properties of the rigid bodies were acquired from the corresponding segments of the morphological reconstruction. Each of the rotational springs was assumed to generate a torque that increased quadratically with joint angle when stretched [44]. As a result, each joint spring has two parameters: joint angle at which the spring torque is zero, and a stiffness parameter. To assign values to these parameters, the biomechanical model was aligned with the morphological reconstruction in two different postures in which the spring torques were known (figure 2). We acquired the first posture by defining passive horizontal equilibrium, which implies that the ligaments are strained in horizontal posture to counteract gravity (figure 2 A,B). It has been shown that the interspinous ligaments could generate ample force to maintain this pose [22], which our models confirm (see the supplementary information). In the second posture, all interspinous ligaments were at resting length, and thus, all spring torques equalled zero. We determined this posture on the basis of an assumption regarding the ligament strain in horizontal equilibrium. Tendons and ligaments can be roughly divided into low-stress and high-stress varieties, with the

high-stress varieties providing more energy savings at the cost of lower safety factors [41]. We imposed an intermediate 4% strain in horizontal equilibrium ($\epsilon_{\text{hor}} = 0.04$), and used this to find a pose where the ligaments are not strained (figure 2 C,D). This is analogous to passive equilibrium in the absence of gravity, or a *T. rex* lying on its side. After aligning the biomechanical model with the morphological reconstruction in both postures, the mechanical equilibrium conditions were used to calculate the parameter values of each rotational spring.

Other changes to the manuscript

We have added a new figure to the main manuscript (figure 2). We have removed figure S8 from the ESM, because the simplified angle definitions have made it redundant. Because we only roughly estimated Young's Modulus to ensure none of the models led to a YM that exceeded 1200 MPa, the section reporting the average YM per segment has been removed from the ESM. It has been summarised with the following eight lines in the ESM (page 8):

To ensure that this did not lead to unrealistic ligament properties, we performed a rough estimate of average YM and ligament pressures per segment in a post hoc analysis. Because reports vary so widely, models were deemed acceptable if YM was below 1200 MPa, and pressure below 50 MPa, because in physiological levels of strain those maximal values are not reached. All of the models abided by these criteria, and we do not report them because they are not directly relevant to the natural frequency of the tail, nor the preferred walking speed. For the interested reader, our MATLAB code (specifically, `Trex_tail_main_script.m`) produces tables of YM and pressures in horizontal equilibrium for all the models.

Figure 1: It could be helpful to show force vectors expected from muscle contraction and/or ligament spring forces (i.e. a free body diagram) in order to help illustrate the mechanical interaction conceptually

Response

The new figure 2 includes ligament forces, and the resultant joint torques. We hope that this makes it conceptually clear what the joint torques represent. We have opted not to draw a formal free body diagram of the vertebra, because that would require inclusion of many distinct forces that would be internal forces in the biomechanical model and are therefore irrelevant to our analysis. We have opted not to include muscle contraction forces, since in this paper we are not modelling any muscle contractions.

Line 65: This is a little confusing since there is a range of natural frequencies associated with the bounds in the sensitivity analysis. Perhaps consider clarifying your meaning "The lowest natural frequency (i.e. the fundamental resonant mode)..."

Response

We agree with this suggestion and have changed the wording to:

The lowest natural frequency (i.e. the frequency corresponding to the fundamental resonant mode)

Line 72-75: As was mentioned in a previous comment, oscillations may not occur entirely out of phase at resonant frequency where peak relative amplitude is maximized. Can the authors confirm this is the case for their model? Perhaps a Bode plot would be useful in illustrating this.

Response

As mentioned in our response to the previous comment regarding the phase difference, we agree that we cannot make the claim that resonance would occur at a 180-degree phase difference, as this is inconsistent with the passive system we consider here. We have corrected the manuscript accordingly.

Changes to manuscript

Discussion has been expanded (see above, lines 319-347)

Line 84-85: Consider word change “the most prominent movements and forces occur in the sagittal plane”. “Importance” is subjective and not clearly defined in this sentence. Another option is “the most energetically relevant movements and forces”

Response

We thank the reviewer the suggestion and have reworded the manuscript accordingly.

Line 123: Change to “modelling steps, as well as mathematical derivations...”

Response

It is not clear to us what change the reviewer is proposing. Do note that we would like to distinguish between the reconstruction of inertial parameters, and the construction of the biomechanical model as separate steps in the process. We expect that many colleagues will be differentially interested in the musculoskeletal reconstruction versus the biomechanical modelling steps.

Line 117: How much confidence do the authors have that this measurement of step length (N=1) is representative of typical gait? I understand that there are inherent limitations to the data available in this field, but are there other trackways to compare with?

Response

The trackway (i.e. a series of multiple successive prints) record from the late Cretaceous of North America is notoriously sparse with regards to large theropods, and there is furthermore the issue that theropod and ornithopod tracks can be difficult to distinguish. For instance, (Manning, Ott, and Falkingham 2008) described one of the first singular footprints that could likely be ascribed to *T. rex*, and a trackway initially ascribed to a large theropod by Thulborn (1984) was later diagnosed to belong to a large ornithopod (Romilio and Salisbury 2011). We chose to use data from McCrea et al. (2014) because it currently represents the only trackway that is reasonably ascribed to a large tyrannosaur. We concede that this is the lowest possible sample size. This is because there are no other, sufficiently large (to ensure geometric and dynamic similarity) theropod trackways from the late Cretaceous of North America. Unfortunately, low sample-size observations are part of the harsh reality that palaeontologists have to deal with. If we were to widen our search to other time periods and landmasses (reminding the reviewer that these landmasses were not connected in the Cretaceous), we would be including different taxa with considerably different proportions to *T. rex*. That being said, we do agree that this issue should be highlighted in the discussion, and have done so.

Changes to the manuscript (expanded discussion, line 268-285):

Recent amendments have been proposed to improve DS-based estimations, either by focusing on swing phase dynamics [54], or by incorporating taxon-specific morphological parameters to reduce uncertainty [55]. The latter are of course difficult to obtain from a fossilised trackway. Our method can help in this regard, by providing a range of plausible step frequencies for any given taxon. With this goal in mind, it would be most effective to estimate the natural frequencies of taxa that are well-represented in the trackway record. That way, it would be possible to combine step length data from several trackways, increasing confidence that the average result should tend towards preferred (and therefore optimal) gait. Unfortunately, this was not possible in the present study, because large theropod trackways with reasonably close taxonomic proximity to *T. rex* are exceedingly rare [47]. Bipedal dinosaurs seem to have preferred to walk at a relative stride length of 1.3, and it has been suggested that this might have been energetically optimal [26,47]. This relation implicitly relates footprint length to stride length, by first estimating hip height from footprint length, and also assumes leg kinematics are known. We therefore preferred to simply relate footprint length directly to stride length from a trackway that could be ascribed to *T. rex* with reasonable certainty [47]. This ensures sufficient geometric similarity to RGM.792000. The resulting step lengths differed by less than 4% from the suggested preferred relative stride length. Whenever considering trackway data, substrate-related uncertainties will prevent any straightforward interpretation [55,56], although presumably these would not affect tail natural frequency.

Line 113-115: Perhaps add "These bounds account for uncertainties... and led to 11 models (see ESM for details)"

Response

In response to both reviewers, we have expanded this section, providing a more in-depth explanation of all the sensitivity analysis models.

Changes to the manuscript (expanded methods section, line 151-165)

To account for uncertainty in inertial estimates, we individually varied length (and thus moment of inertia) and mass. Because we assumed 4% ligament strain in horizontal equilibrium, we investigated the effects of low- and high-stress ligaments on the result, by imposing 3% and 5% strain in horizontal equilibrium, respectively [41]. Although each ligament is well-bounded by the skeleton, the effective point of force application of each ligament was unknown. In the baseline model, we based the moment arms on the area centroids of the ligaments. We used the ligament reconstruction to determine bounds for the moment arms (figures S5 & S7). Lastly, whereas the proximal tail shows pronounced articulations between the zygapophyses, after approximately the 13th caudal this is no longer the case (figure S5). This shifts the rotational axes in the distal tail ventrally, towards the vertebral centra, with the vertical position dependent on whether the tail is in flexion or extension. (figure S6 shows the baseline and two extreme possibilities). We therefore bounded the rotational axes in tail segments 3 and 4, which affected joint springs 3 -5. The dorsal axes are unrealistically high, ensuring a wide bound for the sensitivity analysis. The ventral axes are located at the vertebral articulation with the chevrons, and represent their mechanical effect when the tail is flexed. In total, the aforementioned bounds led to 11 different models.

Table 1: The authors explored sensitivity of their model to parameter uncertainties independently, but this assumes that uncertainty cannot accumulate in multiple dimensions simultaneously. Shouldn't the low/high bounds of natural frequency and resulting optimal walking speed reflect maximal error in all dimensions simultaneously? i.e. the combination of parameter bounds that results in the lowest

and highest natural frequency. Further constraints on Young's modulus, etc. can be applied in case these bounds give unrealistic results.

Response

To our knowledge, development of a multi-segment tail model for a dinosaur has not been attempted before. Therefore, aside from introducing a new method for walking speed estimation, we also developed a method for parametrising a tail model for use in dynamic simulation models of gait. Our goal for the sensitivity analysis was to quantify which step in the entire reconstruction/modelling process provides the largest uncertainty in the estimate. This is because inertial and exact length parameters, for instance, are more uncertain than the extent of the ligaments, since these are bounded by the skeleton. Our hope was that by identifying which steps provide the most uncertainty, future research can focus on improving confidence in these steps (for instance, by determining *in vivo* tail rotational axes in a crocodylian). By choosing relatively large bounds for these, we found that despite their uncertainty, they have less effect on the outcome than the ligaments themselves, which is an encouraging result because unlike the musculature, the extent of the ligaments are well bounded by the skeleton. Our sensitivity analysis suggests that future research should therefore be focused on vertebral kinematics and ligament composition in crocodylian tails.

Lastly: the combinations between Moment arms & distal axes models were initially not reported, because the distal axes models do not represent full step cycles (i.e.: the ventral axes represent the tail during flexion, and the dorsal axis during extension). We would therefore caution the interpretation of their combined effects.

In response to the reviewer, we now explicitly describe our motivation for the inclusion of the sensitivity analysis. We have also highlighted the mathematically predictable nature of the effect of some of the parameters, and added the 4 relevant omitted combination effects as supplementary table S2. Lastly, we have now provided a spreadsheet that can be used to interactively calculate the natural frequency as predicted by any combination of our input parameters. For reasons mentioned above, we prefer not to report the combined effects of all the parameters as bounds, and have instead opted to remove the reported bounds in the abstract. We now explicitly mention in the abstract that the largest uncertainty was due to vertebral kinematics and ligament composition.

Changes to the manuscript (modified abstract, line 21-25):

Using a 3D morphological reconstruction and a spring-suspended biomechanical model, we determined tail natural frequency of *T. rex* (0.66 s^{-1}), and the corresponding walking speed (1.28 m s^{-1}), which we argue to be a good indicator of preferred walking speed. The results are most sensitive to uncertainties regarding ligament moment arms, vertebral kinematics and composition.

Changes to the manuscript (elaboration on secondary goal of the manuscript, line 78-81):

We have also investigated which morphological features of the tail have the largest impact on tail natural frequency. In doing so, we hope to encourage researchers to incorporate a non-rigid tail into their locomotor simulations, while also demonstrating that the relatively simple Natural Frequency Method can be a valuable expansion of the toolkit of palaeo-biomechanists.

Changes to the manuscript (elaboration on goal of the sensitivity analysis, line 145-149):

As was done in previous studies on dinosaur dimensions [29,35,48,49], we subjected the major inputs to a sensitivity analysis. Since we hope to see future studies of dinosaur locomotion incorporate our tail model, we intentionally chose wide bounds for the sensitivity analysis. This

serves to inform researchers which individual steps in the modelling process induce the largest variation in the result. These steps could then be the focus of future research in an attempt to reduce the uncertainty.

Changes to the manuscript (added discussion on combination effects, line 221-226):

Combining altered moment arms with alternative rotational axes strengthened this effect (table S2). However, this table should be interpreted with caution because the distal axes models do not represent axes that would have been used throughout the whole step cycle. Instead, the rotational axes would migrate ventrally during flexion, and dorsally during extension of the tail, which could potentially be influenced by contraction of the epaxial and hypaxial musculature.

Line 157: Consider using a different word from “significant” to avoid conflation with statistical significance

Response

We have changed the wording to “meaningful”.

ESM, Line 186: Change wording to “Due to the presence of these many unknowns...”

Response

We have changed the wording to “Due to the numerous unknowns”.

ESM, Line 278: Missing word, change to “the stiffnesses of the ligaments...”

Response

Thank you, we have changed the wording.

References cited in our response

- Ahlborn, Boye K. and Robert W. Blake. 2002. “Walking and Running at Resonance.” *Zoology* 105(2):165–74.
- Ahlborn, Boye K., Robert W. Blake, and William M. Megill. 2006. “Frequency Tuning in Animal Locomotion.” *Zoology* 109(1):43–53.
- Alexander, R. McN. 2006. *Principles of Animal Locomotion*. Princeton NJ: Princeton University Press.
- Basu, Christopher, Alan M. Wilson, and John R. Hutchinson. 2019. “The Locomotor Kinematics and Ground Reaction Forces of Walking Giraffes.” *Journal of Experimental Biology* 222(2).
- Biewener, A. A. 2008. “Tendons and Ligaments: Structure, Mechanical Behavior and Biological Function.” *Collagen: Structure and Mechanics* 269–84.
- Casius, L. J. Richard, Maarten F. Bobbert, and Arthur J. Van Soest. 2004. “Forward Dynamics of Two-Dimensional Skeletal Models . A Newton-Euler Approach.” 421–49.
- Christian, A., R. H. G. Müller, G. Christian, and H. Preuschoft. 1999. “Limb Swinging in Elephants and Giraffes and Implications for the Reconstruction of Limb Movements and Speed Estimates in Large Dinosaurs.” *Fossil Record* 2(1):81–90.
- Dececchi, T. Alexander, Aleksandra M. Mloszewska, Thomas R. Holtz, Michael Bruce Habib, and Hans C. E. Larsson. 2020. “The Fast and the Frugal: Divergent Locomotory Strategies Drive Limb Lengthening in Theropod Dinosaurs” edited by A. Cuff. *PLOS ONE* 15(5):e0223698.

- Doke, J., J. M. Donelan, and A. D. Kuo. 2005. "Mechanics and Energetics of Swinging the Human Leg." *Journal of Experimental Biology* 208(3):439–45.
- Gellman, Karen S. and J. E. A. Bertram. 2002. "The Equine Nuchal Ligament 2: Passive Dynamic Energy Exchange in Locomotion." *Veterinary and Comparative Orthopaedics and Traumatology* 15(1):7–14.
- Hengst, Richard. 2004. "Gravity and the T. Rex Backbone." Pp. 69A--70A in *Journal of Vertebrate Paleontology*. Vol. 24.
- Holt, Kenneth G., Joseph Hamill, and Robert O. Andres. 1990. "The Force-Driven Harmonic Oscillator as a Model for Human Locomotion." *Human Movement Science* 9(1):55–68.
- Holt, Kenneth G., Joseph Hamill, and Robert O. Andres. 1991. "Predicting the Minimal Energy Costs of Human Walking." *Medicine and Science in Sports and Exercise* 23(4):491–98.
- Hoyt, Donald F. and Richard Taylor. 1981. "Gait and the Energetics of Locomotion in Horses." *Nature* 292:239–40.
- Huat, Ong Jor, Dhanjoo N. Ghista, Ng Kok Beng, and Tan Cher Chay John. 2004. "Optimal Stride Frequency Computed from the Double-Compound Pendulum of the Leg, and Verified Experimentally as the Preferred Stride Frequency of Jogging." *International Journal of Computer Applications in Technology* 21(1–2):46–51.
- Kane, Adam, Kevin Healy, Graeme D. Ruxton, and Andrew L. Jackson. 2016. "Body Size as a Driver of Scavenging in Theropod Dinosaurs." *American Naturalist* 187(6):706–16.
- Kilbourne, Brandon M. and Louwrens C. Hoffman. 2013. "Scale Effects between Body Size and Limb Design in Quadrupedal Mammals." *PLoS ONE* 8(11).
- Kilbourne, Brandon M. and Louwrens C. Hoffman. 2015. "Energetic Benefits and Adaptations in Mammalian Limbs : Scale Effects." (June 2015).
- Lee, Leng Feng and Venkat N. Krovi. 2008. "Musculoskeletal Simulation-Based Parametric Study of Optimal Gait Frequency in Biped Locomotion." *Proceedings of the 2nd Biennial IEEE/RAS-EMBS International Conference on Biomedical Robotics and Biomechatronics, BioRob 2008* (November 2008):354–59.
- Loscher, David M., Fiete Meyer, Kerstin Kracht, and John A. Nyakatura. 2016. "Timing of Head Movements Is Consistent with Energy Minimization in Walking Ungulates." *Proceedings of the Royal Society B: Biological Sciences* 283(1843).
- Manning, Phillip L., Christopher Ott, and Peter L. Falkingham. 2008. "A Probable Tyrannosaurid Track from the Hell Creek Formation (Upper Cretaceous), Montana, United States." *Palaios* 23(10):645–47.
- McCrea, Richard T., Lisa G. Buckley, James O. Farlow, Martin G. Lockley, Philip J. Currie, Neffra A. Matthews, and S. George Pemberton. 2014. "A 'Terror of Tyrannosaurs': The First Trackways of Tyrannosaurids and Evidence of Gregariousness and Pathology in Tyrannosauridae." *PLoS ONE* 9(7).
- Myers, Marcella J. and Karen Steudel. 1997. "Morphological Conservation of Limb Natural Pendular Period in the Domestic Dog (*Canis Familiaris*): Implications for Locomotor Energetics." *Journal of Morphology* 234(2):183–96.
- Raichlen, D. A. 2004. "Convergence of Forelimb and Hindlimb Natural Pendular Period in Baboons (*Papio Cynocephalus*) and Its Implication for the Evolution of Primate Quadrupedalism." *Journal of Human Evolution* 46(6):719–38.

- Romilio, Anthony and Steven W. Salisbury. 2011. "A Reassessment of Large Theropod Dinosaur Tracks from the Mid-Cretaceous (Late Albian-Cenomanian) Winton Formation of Lark Quarry, Central-Western Queensland, Australia: A Case for Mistaken Identity." *Cretaceous Research* 32(2):135–42.
- Rothschild, Bruce M., Robert A. Depalma, David A. Burnham, and Larry Martin. 2020. "Anatomy of a Dinosaur—Clarification of Vertebrae in Vertebrate Anatomy." *Journal of Veterinary Medicine Series C: Anatomia Histologia Embryologia* 49(4):571–74.
- Ruina, Andy and Rudra Pratap. 2019. *Mechanics Toolset, Statics and Dynamics*.
- Russell, Daniel M. and Dylan T. Apatoczky. 2016. "Gait & Posture Walking at the Preferred Stride Frequency Minimizes Muscle Activity." *Gait & Posture* 45:181–86.
- Schroeder, Ryan T., John E. A. Bertram, Van Son Nguyen, Van Vinh Hac, and James L. Croft. 2019. "Load Carrying with Flexible Bamboo Poles: Optimization of a Coupled Oscillator System." *Journal of Experimental Biology* 22(23).
- Sellers, William Irvin, Lee Margetts, Rodolfo Aníbal Coria, and Phillip Lars Manning. 2013. "March of the Titans: The Locomotor Capabilities of Sauropod Dinosaurs" edited by D. Carrier. *PLoS ONE* 8(10):e78733.
- Thulborn, Richard. 1984. "Dinosaur Trackways in the Winton Formation (Mid-Cretaceous) of Queensland." *Memoirs of the Queensland Museum*. 21(2):413–517.
- Wagenaar, R. C. and R. E. A. Van Emmerik. 2000. "Resonant Frequencies of Arms and Legs Identify Different Walking Patterns." *Journal of Biomechanics* 33(7):853–61.
- Wintrich, Tanja, Martin Scaal, Christine Böhmer, Rico Schellhorn, Ilja Kogan, Aaron van der Reest, and P. Martin Sander. 2020. "Palaeontological Evidence Reveals Convergent Evolution of Intervertebral Joint Types in Amniotes." *Scientific Reports* 10(1):1–14.

Reviewer 2

Reviewer: 2

Comments to the Author(s)

The authors present a method for estimating the optimal walking speed of Tyrannosaurus rex. They claim that minimal metabolic expenditure should occur when step frequency matches tail resonant frequency. They reconstruct tail morphology of a T. rex specimen to estimate inertial parameters, lengths of tail segments, positions of ligaments, and axes of rotation. They perform a sensitivity analysis by shifting each of these elements individually from baseline. By approximating each segment as a rigid body attached to the next segment by a torsional spring, they calculate the resonant frequency of the tail. Using trackways, they then estimate the walking speed corresponding to this frequency, and claim that it is the optimal walking frequency.

The tail reconstruction is detailed and informative. The sensitivity analysis leads to some interesting insights, for example that the distal tail has pronounced effects on the resonant frequency depending on the location of the rotational axis.

The paper provides a method to model tail oscillations in extinct dinosaurs, and another method to estimate walking speeds. Despite these strengths, some major issues should be addressed before publication.

The basic premise- that optimal walking speeds correspond to resonant frequencies- is not well supported. Interaction effects are ignored in the sensitivity analysis. Key aspects of the methods and context for the discussion are relegated to supplemental information. Finally, while the biomechanical model seems sensible, the mathematical derivation involves ambiguous terminology and appears inconsistent as written.

If the authors can address these concerns, I believe this paper would be a useful addition to the literature.

Response

We agree with the reviewer's summary of our method, and thank the reviewer for the time spent reviewing our manuscript and the encouraging comments regarding our findings. We would also especially like to thank the reviewer for bringing interesting literature to our attention that we were not previously aware of. We will address the reviewer's concerns on a point-by-point basis below.

We also mention to the reviewer that, in response to comments from the associate editor, we have opted to change instances of "optimal walking speed" to "preferred walking speed", unless referencing a study where energetics were directly measured or estimated. This has also changed the title of our manuscript.

Major revisions

--

Lines 39-41: "Optimal locomotor speeds are to a large extent the result of the natural frequencies of relevant body segments, through a combination of resonance and elastic storage (9–12)."

This is a bold claim, and deserves a little more discussion than simply pointing to references 9-12. What is the specific evidence that natural frequencies determine optimal locomotor speeds? Some examples could be provided.

As it stands, I am not fully convinced that all the references support the claim. For example, (10) show that work is minimized at natural leg swing frequency, but metabolic cost is minimized at a higher frequency, and do not show whether this is the preferred or optimal frequency in walking.

Response

We thank the reviewer for pointing out that our basic premise was not sufficiently supported, and that our claims were formulated too boldly. We also agree that the inclusion of (Doke et al. 2005) needs to be formulated more carefully: we referenced this paper to support the claim that moving body parts at frequencies close to their natural frequencies, minimizes mechanical work. Quantitatively, the minimum in metabolic cost is indeed at a frequency that is slightly higher than the natural frequency. However, given the uncertainties we must overcome within palaeontology, we consider the difference to be small.

We agree that the claims in this section should be formulated more carefully and elaborated upon with more relevant citations and examples from the literature. We have split our basic premise, that the empiric relationship between natural frequencies and locomotor frequencies allows for estimation of preferred walking speed, into several sub-claims. Furthermore, we have collected more references to support each of these claims.

Changes to the manuscript (expanded premise, line 32-49)

Animals display a variety of gaits and walking speeds. Two locomotor extremes are of particular interest: the maximal speed, and the energetically optimal walking speed, at which the metabolic cost of transport (MCOT, in $\text{J m}^{-1} \text{kg}^{-1}$) is minimal for walking gait. In the absence of external task constraints, optimal walking speed is very close to preferred walking speed in humans [1,2], raities [3], horses [4] and elephants [5]. It is likely that they are related in all terrestrial animals, and that animals would tend to forage at this speed. During locomotion at preferred speed, animals tend to make use of resonance by matching locomotor frequencies to the (undamped) natural frequencies of their relevant body parts, sometimes tuned through muscular contractions [6–13]. Indeed, it has been demonstrated that mammalian quadrupedal taxa have convergent natural frequencies between fore- and hindlimbs despite differing limb morphologies [14,15]. Both fore- and hindlimb natural frequencies scale similarly with several morphological parameters within several mammalian groups [16,17], suggesting that resonance is beneficial at any body size. An important advantage of moving a body part close to its natural frequency is that this reduces mechanical work [18,19]. Based on the commonly adopted idea that preferred walking speed is chosen to minimise MCOT [1–5], it has been argued that preferred step frequencies are close to the natural frequencies of relevant body parts [7,11,12]. Determination of the natural frequency of a body part relevant to locomotion can therefore constrain preferred (and presumably, optimal) step frequencies and walking speeds of extinct taxa [10].

--

Lines 51-52: "Due to resonance, elastic storage would be maximal if step frequency matched the natural frequency of the tail."

Lines 69-70: “Elastic storage in the caudal ligaments would reduce the energy losses at each footfall, thereby reducing MCOT.”

Lines 73-75: “Therefore, it would be beneficial to maximize tail amplitude (which occurs when step and natural frequency are matched), as this maximizes ligament strain and minimizes MCOT.”

Maximal ligament strain does not necessarily mean minimal metabolic cost. If this were true, humans would stretch Achilles tendon maximally during walking / running, regardless of speed! Damping, active muscle volume, muscle activation all seem to contribute to metabolic cost and call for submaximal tendon/ligament strain (e.g. Collins et al. 2015 doi:10.1038/nature14288; Orsell et al. 2017 doi:10.1016/j.jbiomech.2017.06.022).

Response

The reviewer points to interesting interactions that occur when adding springs across the ankle joint to reduce mechanical joint work. In this specific situation, excursion of the muscle fibres increases because tendon force (and therefore excursion) decreases once the spring is added. The interactive effects of muscle fibre excursion and tendon excursion are indeed very complex, but go beyond the scope (and goal) of our manuscript. Our method simply relates the empirical finding of resonance in locomotion (see above) to extinct dinosaurs, without delving into the more subtle dynamic interactions that occur at the level of individual muscles, in which serial elasticity of the tendons affects work done by the muscle fibres.

We agree that more subtle interactions would occur, at the cost of higher model complexity, which is counterproductive to the goal of the current manuscript. Instead, the possible trade-offs that may be investigated when using a more sophisticated model have been elaborated upon in the discussion.

Changes to the manuscript (expanded discussion on complex interactive effects, line 319-347)

Our method for walking speed estimation is meant to be a reductionist analysis, so we have only incorporated the interspinous ligaments, since we expect their mechanical effect to dominate overall tail dynamics. Natural frequencies play an important role in animal locomotion [6–13], and NFM therefore provides a reasonable starting point for investigating how the resonant properties of dinosaur tails affect overall locomotion. However, in reducing the analysis to the mechanics of a single structure, many of the complex interactions are ignored. For instance, animals tune the frequencies of their segments through muscular contractions [6,7]. *T. rex* could have contracted its epaxial and hypaxial caudal musculature to add rotational stiffness to the tail, which could be beneficial at higher speeds, for instance during a pursuit. Our ventral axes model demonstrates a possibility in this regard: if *T. rex* were to keep its distal tail in slight flexion, thereby migrating the rotational axes of the vertebrae to the articulations with the chevrons, the overall natural frequency of the tail would increase.

Tail musculature could also be employed to enforce beneficial phase relationships between vertical hip and tail oscillations. During walking, vertical oscillations between withers and heads in most ungulates are out of phase, which provides them with an energetic benefit [38–40,57]. This is theorised to be modulated by natural frequencies of the neck segment [39,40], but there is an active component to this behaviour as well [38]. This phase relationship reduces the losses incurred during

the step-to-step transitions, which are identified as major sources of energy losses in inverted pendulum models of walking [58]. The implications of energy storage are also further complicated by the serial elasticity in musculotendon complexes: tendon stiffness can substantially impact the mechanical work done by the muscle fibres [59]. Finally, the cost of cyclical muscle contractions could also affect metabolic optima, depending on the task requirements [18,19]. Such complex interactions would undoubtedly affect PWS, but require extensive reconstructions of muscular contractile properties. Muscular parameters currently provide the biggest uncertainty in most musculoskeletal simulations of dinosaurs [43,51,60], so any investigation of these interactions would require careful consideration. Inclusion of our compliant tail model into fully-actuated hindlimb simulations of *T. rex* may shed further light on how the unique tails of non-avian dinosaurs functioned during locomotion. However, in such complex simulations, the intricacies of tail dynamics may be overshadowed by the uncertainties regarding the contractile properties of the muscles.

--

Lines 115-121: step length from trackways.

The authors already point to issues with substrate affecting walking speed. They claim that the walking speed reported, as determined from this step length is “energetically optimal”. How do they know that the trackways were for a preferred (let alone optimal) step length?

Response

Our initial choice for using a single trackway (i.e. a series of multiple successive prints), was motivated by the fact that large theropod trackway data from the late Cretaceous of North America is notoriously sparse (Manning et al. 2008; McCrea et al. 2014). McCrea et al. have described the only trackway that can reasonably be ascribed to an adult *T. rex* (or a very close relative), which is why we only used that trackway. The reviewer correctly points out that it is impossible to know if a trackmaker was walking at its optimal or preferred speed when making the track. Given that animals tend to walk at preferred walking speed, trackway data should reflect this, once enough for any given taxon have been described (perhaps containing a relatively constant bias induced by a nonrigid substrate).

We concede that our estimate of step length is based on the lowest possible sample size, but there are no other sufficiently large (to ensure geometric and dynamic similarity) theropod trackways from the late Cretaceous of North America. Unfortunately, low sample-size observations are part of the harsh reality that palaeontologists have to deal with. If we were to widen our search to other time periods and landmasses (reminding the reviewer that these landmasses were not connected in the Cretaceous), we would be including different taxa with considerably different proportions to *T. rex*. That being said, we do agree that this issue should be highlighted in the discussion, and have done so.

Changes to the manuscript (expanded discussion, line 268-285):

Recent amendments have been proposed to improve DS-based estimations, either by focusing on swing phase dynamics [54], or by incorporating taxon-specific morphological parameters to reduce uncertainty [55]. The latter are of course difficult to obtain from a fossilised trackway. Our method can help in this regard, by providing a range of plausible step frequencies for any given taxon. With

this goal in mind, it would be most effective to estimate the natural frequencies of taxa that are well-represented in the trackway record. That way, it would be possible to combine step length data from several trackways, increasing confidence that the average result should tend towards preferred (and therefore optimal) gait. Unfortunately, this was not possible in the present study, because large theropod trackways with reasonably close taxonomic proximity to *T. rex* are exceedingly rare [47]. Bipedal dinosaurs seem to have preferred to walk at a relative stride length of 1.3, and it has been suggested that this might have been energetically optimal [26,47]. This relation implicitly relates footprint length to stride length, by first estimating hip height from footprint length, and also assumes leg kinematics are known. We therefore preferred to simply relate footprint length directly to stride length from a trackway that could be ascribed to *T. rex* with reasonable certainty [47]. This ensures sufficient geometric similarity to RGM.792000. The resulting step lengths differed by less than 4% from the suggested preferred relative stride length. Whenever considering trackway data, substrate-related uncertainties will prevent any straightforward interpretation [55,56], although presumably these would not affect tail natural frequency.

Related: Lines 208-210: “DS has provided interesting qualitative observations. Bipedal dinosaurs seem to have preferred to walk at a relative stride length of 1.3...”

*If hip height is known for the author’s specimen, could they not simply use (hip height)*1.3 to get preferred stride length? How does this method compare to the trackway measurement?*

The reviewer makes an interesting suggestion, that we have indeed considered. However, Thulborn’s “preferred relation” is not based on hip height measurements. Instead, hip height has been estimated from the footprint lengths using allometric equations. Essentially, we believe that this induces more error than simply assuming that the footprint length is a good predictor of stride length. Hip height estimation makes the same assumption, but also assumes that footprint length is a good predictor of hip height. It therefore makes two assumptions whereas the stride-footprint ratio (SFR) only makes one. It further presupposes that we actually know the leg kinematics of *T. rex*, which would influence effective hip height even further. The relation found by Thulborn is an average of all observed tracks, including very small taxa. We would expect *T. rex* to walk with shorter relative strides than average, because at higher masses it becomes more costly to take long strides. Using Thulborn’s relation, we came to a step length of 2.015 m (versus 1.94 m using SFR, less than 4% difference), and therefore concluded that SFR was more desirable since it makes less assumptions to find similar conclusions. We now mention this explicitly in the manuscript

Changes to the manuscript (elaboration on step/stride lengths in the discussion, line (318-324))

See the response above.

The authors have not looked at interaction effects in the sensitivity analysis. I appreciate that this adds a combinatorial level of complexity (and time!) to the analysis, however I do think it is important.

At the very least, I would ask the authors to combine all the parameters that have the greatest increase and decrease in step frequency (e.g. decrease length + max. moment arms + ventral axes + low strain to test whether they combine to increase frequency further. Opposite parameters for decreased frequency). That way, we can at least have an idea of whether these effects are additive.

Response

Mass does not affect the result, and was included only to demonstrate that our derivations and predictions based on a 1-segment tail are also applicable to a 5-segment tail. Similarly, changes in length (L) and strain in horizontal position (ϵ_{hor}), have an analytically calculable effect on the baseline natural frequency (f_n) for small amplitudes. In essence, $f_n \propto L^{-0.5}$ and $f_n \propto \epsilon_{hor}^{-0.5}$. This makes it unnecessary to report their combined effects, because these scaling laws (derived in the ESM) still hold true when combining them (also when combining them with the changed moment arms or rotational axes models). For instance: if the desired model has a 12% longer tail, and 10% less strain in horizontal equilibrium (both from baseline), one can multiply our baseline natural frequency by $1.12^{-0.5} * 0.9^{-0.5}$. We have now highlighted this in the manuscript, and reordered table 1 to group together the parameters for which we have derived general scaling laws.

The previous is true because parameter changes for m, L and ϵ_{hor} are mathematically recalculated for each segment. For the moment arms & rotational axes models, this is not the case: the parameter changes are based on the measured tail morphology. This means that their effect on the outcome cannot be easily predicted, and therefore combinations between these parameters represent the only “real” combination effects. There are two different parameters (moment arms and rotational axes), with each three different possibilities (increased, baseline, decreased). This leads to 3^2 possible combinations, five of which were already reported as main results. We have now added the remaining 4 combinations as supplementary results (table S2). We remind the reviewer that the rotational axes models are different from all the other models: the rotational axes in the distal tail should migrate ventrally during flexion, and dorsally during extension. While the bounds are the absolute extremes, we consider it likely they are both represented in a step-cycle (to some degree). In our view, combining an extreme axis (which only occurs during a certain proportion of the step cycle) with an extreme moment arm (which quantifies the uncertainty in where the effective point of force application would be between any two ligaments), does not adequately represent reality. As is now mentioned in the text, caution should be exercised when interpreting these combinations.

Changes to the manuscript (expanded results, line 188-198):

The effects of changes in length (and thus inertia), mass and strain are mathematically predictable in our model, and we have thus derived general scaling laws for these parameters in the supplementary text. Mass had zero effect on the natural frequency. Length (L) and strain in the horizontal position (ϵ_{hor}) are related to the natural frequency (f_n) as follows: $f_n \propto L^{-0.5}$ and $f_n \propto \epsilon_{hor}^{-0.5}$. These relations can also be applied to the altered moment arms or distal axes models. For instance, a 6% longer tail, with 10% more strain in horizontal equilibrium (4.4% strain instead of 4%), using the maximal moment arms, would lead to a natural frequency of $0.79 \cdot 1.06^{-0.5} \cdot 1.1^{-0.5} = 0.73 \text{ s}^{-1}$. The moment arms models cannot be combined with the distal axes models in this way, because these models are based on measurements of morphological traits, instead of systematic parameter variations. Their combinations are reported in Table S2, and the above proportionality relations can be applied to those combination effects as well.

This minimal level of testing interactions would be sufficient in my view, but will leave room for doubt about whether resonance uncertainty is higher than reported.

If the authors have the time and motivation to look at all combinations, that would be ideal. I think it would make a much stronger paper, put to rest any doubt about untested combinations, and set a much more reliable measure of estimate uncertainty.

I would also note that computational time does not seem to be an issue; their matlab script calculates tail natural frequencies in fractions of a second. There may be some technical hassle in writing the code to iterate through the different combinations, but it should be doable with some nested loops.

Response

There are four input parameters that affect the result (excluding mass), and in the sensitivity analysis performed, each of these has three possible values (baseline, lower or upper). This leads to 3^4 , or 81 total combinations. We have provided general scaling laws (that can be combined), and now explicitly state in the manuscript that the reader can use these to calculate the changes from baseline for any given model. To be able to do this for all 81 possible combinations, we have provided the 4 combinations that were omitted previously as ESM table 2 (but see our reservations in the response above). To make this easier for the reader, we have developed a spreadsheet that can be used interactively to calculate natural frequency & walking speed as a function of parameter changes, which we have now also supplied as ESM.

Our main goal of the sensitivity analysis was to quantify which of the steps in reconstruction and parametrisation of the tail contribute most to the uncertainty of the predicted natural frequency. We hope to encourage researchers to incorporate our mechanistic tail model into locomotor simulations, since the locomotor studies we have seen only incorporate a rigid tail. We now mention this in the introduction and methods section.

The largest uncertainty seems to be due to the moment arms of the ligaments, which are well bounded by the skeleton. This is an encouraging result, and in highlighting this, we intended to 1) signify that future research should focus on tail vertebral dynamics and ligament composition in crocodylians, and 2) highlight that uncertainty in inertial & muscular parameters which tend to dominate the effect in simulation studies do not play a large role in parametrizing the tail in such a way. This is now explicitly mentioned in the discussion.

Changes to the manuscript (elaboration on secondary goal of the manuscript, line 78-81):

We have also investigated which morphological features of the tail have the largest impact on tail natural frequency. In doing so, we hope to encourage researchers to incorporate a non-rigid tail into their locomotor simulations, while also demonstrating that the relatively simple Natural Frequency Method can be a valuable expansion of the toolkit of palaeo-biomechanists.

Changes to the manuscript (elaboration on goal of the sensitivity analysis, line 145-149):

As was done in previous studies on dinosaur dimensions [29,35,48,49], we subjected the major inputs to a sensitivity analysis. Since we hope to see future studies of dinosaur locomotion incorporate our tail model, we intentionally chose wide bounds for the sensitivity analysis. This serves to inform researchers which individual steps in the modelling process induce the largest variation in the result. These steps could then be the focus of future research in an attempt to reduce the uncertainty.

Changes to the manuscript (added discussion on combination effects, line 221-226):

Combining altered moment arms with alternative rotational axes strengthened this effect (table S2). However, this table should be interpreted with caution because the distal axes models do not represent axes that would have been used throughout the whole step cycle. Instead, the rotational axes would migrate ventrally during flexion, and dorsally during extension of the tail, which could potentially be influenced by contraction of the epaxial and hypaxial musculature.

Changes to the manuscript (expanded discussion, line 301-308):

The largest uncertainty in the present study is related to the moment arms of the interspinous ligaments. This is an encouraging finding, because their extent is well-bounded by the skeleton. Essentially, one of the more certain parameters provides the most uncertainty. Future research on in vivo vertebral kinematics, dynamics and ligament compositions of crocodylian tails could reduce this uncertainty. Similarly encouraging is that our results are minimally sensitive to estimates of muscularity or inertial parameters in general. Indeed, when isolating mass as a free parameter in our sensitivity analysis, we have shown that our predicted speeds are unaffected by mass-estimates (Table 1).

Changes to the manuscript (expanded discussion, line 313-318):

Minimal sensitivity to muscular & inertial estimates implies that our inverse dynamic approach to constructing a non-rigid tail could be incorporated into more sophisticated hindlimb simulation models, without adding much to the overall uncertainty of the result. This could have implications for maximal running speeds of large taxa like *T. rex*: maximum running speed was shown to be limited by peak stresses on the limbs [32], but a compliant tail may serve to reduce these stresses.

--

Re: novelty of the method

Lines 176-177: Until now, estimates for (submaximal) dinosaur walking speeds are all based on Dynamic Similarity (DS).

Lines 212: "The possibility of lower step frequencies than predicted by the equation has only been suggested in passing (14,38). Until now, no steps have been taken to explore this."

Lines 223-225: "Deriving walking speed using Dynamic Similarity has its limitations. However, until now it was the only approach used to estimate (non-optimal) dinosaur walking speeds."

Dynamic Similarity is not the only approach used (though it is most common), the possibility of it being wrong has been explored (a little), and natural frequencies of body segments have been used before (though not the tail as far as I know).

Response

In light of the research that has been brought to our attention, we agree that our statements should be amended to reflect earlier work. We will deal with these in turn below.

For (extant) avian dinosaurs, much has been done, though I assume the authors meant extinct dinosaurs. They might be particularly interested in Marmol-Guijarro et al. (2020 doi:10.1186/s12983-020-00363-z), who question the use of dynamic similarity by applying it to ptarmigan tracks.

Response

We thank the reviewer for bringing this interesting work to our attention, and agree that it is relevant to our manuscript. The authors suggest that (morphological) parameters not derived from the trackways themselves are necessary to improve accuracy of any trackway speed estimate. This is in line with our own suggestion that a relationship between footprint lengths and preferred step frequencies (found using NFM) could help to constrain trackway speed estimates. The authors also mention that difficulty in relating a trackway footprint length to osteological data could still add uncertainty, which our method cannot resolve. Tail natural frequency, however, would not change in response to different substrates. We have added these considerations to the discussion.

Changes to the manuscript (expanded discussion, line 268-285)

See our response above, with regards to trackway estimates.

Sellers et al. (2013 doi: 10.1371/journal.pone.0078733) used a full musculoskeletal model to find metabolically optimal walking speeds in a sauropod.

Response

We were familiar with Sellers et al. (2013), but upon re-evaluation conclude that they did indeed investigate optimal walking speed, although they did not refer to it as such. We thank the reviewer for pointing this out, and now mention this explicitly in our manuscript.

Changes to the manuscript (expanded discussion, line 293-294):

To our knowledge, musculoskeletal simulation models have not been used to estimate how MCOT varied with walking speed in *T. rex*, although this has been done for *Argentinosaurus* [51]

Christian et al. (1999 doi:10.1002/mmng.1999.4860020105) investigated an alternative method from DS, similar to the “Natural Frequency Method” proposed in the submitted paper. They investigated whether the natural pendular period of limbs in giraffe and elephants was predictive of their step (or swing) frequency. Results were mixed, and they show that that dynamic similarity is a better predictor of walking speed from trackways in elephants than using the natural frequency of the limbs.

While they caution that natural pendulum frequencies won't give precise estimates in general, they do use natural pendulum frequency of limbs to estimate “comfortable walking speeds” of four species of dinosaurs.

Response

We would like to extend our sincere gratitude to the reviewer for bringing this research to our attention. We were unaware of this work, and agree that the approach of these authors is very similar to ours, albeit that they do not involve tail natural frequencies nor do they reconstruct any spring-like properties of passive structures. After studying the claims in Christian et al. (1999), we subtly differ from the reviewer's interpretation. Christian et al. state “Elephants as well as giraffes tended to prefer stride frequencies close to or a little bit below the estimated natural pendulum frequencies of the hindlimbs, if they walked undisturbed over a long distance”. They also state that other stride frequencies were exhibited by the elephants, but these animals were motivated to do so by their caretaker while the giraffes were not. They later go on to state that the pendular period seemed to be a superior predictor near preferred gaits, whereas Dynamic Similarity outperformed it at higher speeds. Since our goal is to estimate the former, we interpret this as lending support to

our method, and therefore now reference them in our premise. We have also expanded several sections of the discussion to reflect their work in this regard.

Changes to manuscript (explicit reference to Christian et al. (1999) in the premise, line 47-49):

Determination of the natural frequency of a body part relevant to locomotion can therefore constrain preferred (and presumably, optimal) step frequencies and walking speeds of extinct taxa [10].

Changes to the manuscript (expanded discussion, line 244-247):

Very little work has been done to directly investigate PWS of dinosaurs. Limb natural frequencies have been used to estimate “comfortable walking speeds” of several sauropod taxa [10], and dynamic simulations were used to estimate preferred walking speed of *Argentinosaurus* [51]. To our knowledge, no such work has been done on theropods.

Changes to the manuscript (expanded discussion, line 261-264):

It has been shown that limb natural frequencies provided better predictions for elephants and giraffes near the preferred speed [10]. Near preferred speeds, DS consistently overestimated walking speeds, but accuracy of the predictions improved at speeds substantially higher than the preferred speeds [10].

Changes to the manuscript (expanded discussion, line 288-292):

However, methods to calculate MCOT are heavily dependent on both speed and mass [30], so we suggest that any comparison of foraging costs should use the PWS of the respective taxa as a starting point. This could be done using the natural frequency of the legs [10], musculoskeletal simulations [51], or NFM as we propose it.

I would suggest the authors preface their claim of novelty, with reference to this previous work.

Response

We agree, and now mention the work of Sellers et al. (2013) and Christian et al. (1999) explicitly in multiple sections of our manuscript, and refrain from claiming to have made the first estimate of PWS in a dinosaur.

Changes to the manuscript

See our responses above.

--

Some discussion points are confusing without referencing the supplemental information. For example, in the main text there is no mention of how distal rotational axes are changed, or why only the 3rd and 4th segments are altered. Yet this is a key point relevant to the second discussion paragraph.

Similarly, the authors suggest an intriguing insight regarding the function of chevrons on lines 159-160, but the role of the chevrons in the model is lost to any reader who has not referenced the supplement.

Other points in the discussion are similarly confusing. I would encourage the authors to move enough of the supplement to the main text that the key discussion points are understandable without referencing the supplement.

Response

We thank the reviewer for their kind words, and agree with this suggestion. We have expanded the methods section so that the reader doesn't need to reference the ESM for vital information. We have also added a second figure (figure 2) to the main manuscript, which hopefully adds further clarity.

Changes to the manuscript (expanded methods section, line 85-165):

We estimated tail natural frequency of *T. rex*, by numerically determining the lowest eigenfrequency of a biomechanical model. Essentially, we constructed this model by estimating inertial parameters of the tail, dividing it into 5 segments, and then fitting joint spring parameters based on inverse-dynamic relationships of different postures of the skeleton.

Inertial parameters of this model were based on a 3D volumetric musculoskeletal reconstruction. This reconstruction was performed on adult *T. rex* specimen RGM.792000 (nicknamed "Trix", figure 1 A), in the collection of Naturalis Biodiversity Center in Leiden, the Netherlands. This is a large, adult specimen with exceptional surface preservation, making it possible to accurately distinguish the attachment sites of the interspinous ligaments. We articulated 3D scans of the caudal skeleton (figure 1C, figure S1), after which we reconstructed the major caudal musculature to estimate inertial parameters (figure 1C, figure S2-4). Using osteological landmarks combined with physical articulation of 3D prints, we determined the axes of rotation between the vertebrae (figure S5-6). Subsequently, we reconstructed the caudal interspinous ligaments (figure 1B), which enabled determination of moment arms and cross-sectional area (figure 1B, figure S7). The morphological reconstruction of the ligaments made it possible to quantify the kinematic relation between vertebral flexion/extension and length of the individual ligaments (i.e. strain), which was used to construct the biomechanical model. Parameters of the reconstruction are provided in Table S1.

Taking the morphological reconstruction as starting point, we then defined a simplified biomechanical model consisting of 5 rigid bodies, connected in hinge joints, with a nonlinear rotational spring at each joint (figure 1, figure 2 for angle definitions). During walking, the most prominent movements and forces occur in the sagittal plane, and the moment arm of the CFL is also largest in this plane. Furthermore, exchange between gravitational and elastic potential energy is most meaningfully studied in this plane. Therefore, similar to previous researchers [29,32], we elected to perform a sagittal plane analysis. Inertial properties of the rigid bodies were acquired from the corresponding segments of the morphological reconstruction. Each of the rotational springs was assumed to generate a torque that increased quadratically with joint angle when stretched [44]. As a result, each joint spring has two parameters: joint angle at which the spring torque is zero, and a stiffness parameter. To assign values to these parameters, the biomechanical model was aligned with the morphological reconstruction in two different postures in which the spring torques were known (figure 2). We acquired the first posture by defining passive horizontal equilibrium, which implies that the ligaments are strained in horizontal posture to counteract gravity (figure 2 A,B). It has been shown that the interspinous ligaments could generate ample force to maintain this pose [22], which our models confirm (see the supplementary information). In the second posture, all interspinous ligaments were at resting length, and thus, all spring torques equalled zero. We determined this posture on the basis of an assumption regarding the ligament strain in horizontal equilibrium. Tendons and ligaments can be roughly divided into low-stress and high-stress varieties, with the high-stress varieties providing more energy savings at the cost of lower safety factors [41]. We imposed an intermediate 4% strain in horizontal equilibrium ($\epsilon_{hor} = 0.04$), and used this to find a pose where the ligaments are not strained (figure 2 C,D). This is analogous to passive equilibrium in the absence of gravity, or a *T. rex* lying on its side. After aligning the biomechanical model with the

morphological reconstruction in both postures, the mechanical equilibrium conditions were used to calculate the parameter values of each rotational spring.

In the absence of damping, resonance of the tail would occur at the undamped natural frequency (f_n) of the system [45]. In reality, damping is omnipresent. In that case, resonance occurs at a forcing frequency of $f_n \cdot (1-2\beta^2)^{0.5}$, with β being the damping ratio [45]. However, since ligaments (like tendons) display relatively little energy loss during work-loop experiments [41,46], little damping is present. Therefore, we will use the undamped natural frequency of the biomechanical model as a proxy for the resonance frequency. To calculate the natural frequency of the biomechanical model, we first linearised the equations of motion in horizontal equilibrium using standard numerical methods. Next, the five pairs of conjugated purely imaginary eigenvalues of the system matrix were determined using the “eig” function in MATLAB. Subsequently, the natural frequency was calculated as the oscillation frequency pertaining to the fundamental resonance mode, i.e. the mode with the lowest eigenfrequency. Finally, walking speed was estimated by multiplying natural frequency of each model by step length, defined as half the distance between two consecutive footfalls of the same foot. We determined a step length of 1.94 m by scaling a large tyrannosaurid trackway [47] based on footprint length of RGM.792000 (see the supplementary text for details).

As was done in previous studies on dinosaur dimensions [29,35,48,49], we subjected the major inputs to a sensitivity analysis. Since we hope to see future studies of dinosaur locomotion incorporate our tail model, we intentionally chose wide bounds for the sensitivity analysis. This serves to inform researchers which individual steps in the modelling process induce the largest variation in the result. These steps could then be the focus of future research in an attempt to reduce the uncertainty.

To account for uncertainty in inertial estimates, we individually varied length (and thus moment of inertia) and mass. Because we assumed 4% ligament strain in horizontal equilibrium, we investigated the effects of low- and high-stress ligaments on the result, by imposing 3% and 5% strain in horizontal equilibrium, respectively [41]. Although each ligament is well-bounded by the skeleton, the effective point of force application of each ligament was unknown. In the baseline model, we based the moment arms on the area centroids of the ligaments. We used the ligament reconstruction to determine bounds for the moment arms (figures S5 & S7). Lastly, whereas the proximal tail shows pronounced articulations between the zygapophyses, after approximately the 13th caudal this is no longer the case (figure S5). This shifts the rotational axes in the distal tail ventrally, towards the vertebral centra, with the vertical position dependent on whether the tail is in flexion or extension. (figure S6 shows the baseline and two extreme possibilities). We therefore bounded the rotational axes in tail segments 3 and 4, which affected joint springs 3-5. The dorsal axes are unrealistically high, ensuring a wide bound for the sensitivity analysis. The ventral axes are located at the vertebral articulation with the chevrons, and represent their mechanical effect when the tail is flexed. In total, the aforementioned bounds led to 11 different models.

--

The authors appear to be using a small-angle approximation in their derivations in the supplemental information. Is the small-angle approximation justified? What are the maximum relative angles between joints experienced during resonance?

Response

The reviewer is correct that we use a small angle approximation. Put more generally, we calculate eigenfrequencies on the basis of a linearization of the tail dynamics in the neutral position. We

neglect damping in the tail as ligaments are generally assumed to introduce very little damping (Alexander 2006; Biewener 2008). Yet, damping is omnipresent and in reality, damping will (fortunately!) limit the amplitude of oscillations when a periodic input (such as hip vertical motion) is applied. We have determined a realistic ligament stiffness by assuming a reasonable amount of strain in the ligaments and using inverse dynamics. Unfortunately, we see no way to estimate damping in the tail based on structural properties (see also our response below to the comment about intervertebral discs).

However, when relative damping (also known as “damping ratio”) is small, the differences between undamped natural frequency, damped natural frequency and resonance frequency are negligible (Ruina and Pratap 2019). To give readers an impression of the steady state behaviour that results when vertical oscillations of the hip are imposed to a lightly damped tail, we have provided a video as supplementary material (video S1). We mention these considerations explicitly in our expanded methods section, and mention the video in the results section.

Changes to the manuscript (explicit treatment on oscillations in the methods section, line (130-135):

In the absence of damping, resonance of the tail would occur at the undamped natural frequency (f_n) of the system [45]. In reality, damping is omnipresent. In that case, resonance occurs at a forcing frequency of $f_n \cdot (1-2\beta^2)^{0.5}$, with β being the damping ratio [45]. However, since ligaments (like tendons) display relatively little energy loss during work-loop experiments [41,46], little damping is present. Therefore, we will use the undamped natural frequency of the biomechanical model as a proxy for the resonance frequency.

Changes to the manuscript (expanded results section, referencing video S1, line 179-187):

To provide the reader with an impression of the dynamic behaviour that results, we have provided a simulation of a lightly damped version of the baseline model (video S1). In this simulation, the only input to the tail was a sinusoidal motion at the base of the tail with an amplitude of 0.08 m. Assuming pendular walking, this is the vertical motion that results from taking 1.94 m long steps at a hip height of 3.1 m. No muscle forces were included, so the phase relationship between the vertical oscillations of the hip and tail may not be representative of a muscle-actuated tail. In this simulation, a small amount of damping was introduced in order to obtain a finite tail amplitude. Damping ratio of the fundamental eigenmode was 0.16, which has a negligible effect on the resonant frequency [45].

--

While the geometry and physical principles of the biomechanical model are straightforward, some of the mathematics in the supplemental information are confusing, ambiguous or inconsistent. The key issues are

- 1. The conflation of $\Delta\epsilon$ with ϵ*
- 2. The ambiguity of the meaning of φ , and related issues:*
 - a. the possible conflation with $\Delta\varphi$*
 - b. the introduction of an undefined variable $\varphi\epsilon$ in equation 6*
 - c. The ambiguity about rest angles*

3. The introduction of Rs without its explicit connection to r

Response

We thank the reviewer for this in-depth consideration of our derivations and appreciate the effort that will help us improve how they are explained. Both reviewers have pointed that the model description was quite complicated/confusing, whereas we had thought that the process, as described, was relatively straightforward. Essentially, we are dividing the tail into 5 segments, and then fitting joint spring parameters using kinematic relationships measured in two different postures of the skeleton. Also, in response to reviewer 2's comments regarding confusing angle definitions, we realised that some steps were not adequately explained, but perhaps were also unnecessarily complicated.

Since our overarching goal is a simple, reductionist method that is easily to adopt, both for frequency analysis and full-scale dynamic simulation models, we have re-evaluated our modelling steps to see at what points we could further simplify and streamline the process (and description). The changes carried through have resulted in natural frequencies that are slightly lower than those reported in version 1 of the manuscript. It has also slightly raised average Young's Modulus per segment, but none of the models show unrealistic/impossible values.

We have reworded the summary of our method in the main manuscript in a manner that we hope clarifies the method (and indeed shows how straightforward it is to implement). To this end, we have also added a second figure to the main manuscript, which hopefully will also help to resolve the issues regarding the angle definitions below. Apart from these changes and additions, we have also incorporated the suggestions of the reviewer regarding the derivations, and will respond to them below.

Changes to the manuscript

Modified and expanded methods section (see the response above).

Other changes to the manuscript

We have added a new figure to the main manuscript (figure 2). We have removed figure S8 from the ESM, because the simplified angle definitions have made it redundant. Because we only roughly estimated Young's Modulus to ensure none of the models led to a YM that exceeded 1200 MPa, the section reporting the average YM per segment has been removed from the ESM. It has been summarised with the following eight lines in the ESM (ESM page 8):

To ensure that this did not lead to unrealistic ligament properties, we performed a rough estimate of average YM and ligament pressures per segment in a post hoc analysis. Because reports vary so widely, models were deemed acceptable if YM was below 1200 MPa, and pressure below 50 MPa, because in physiological levels of strain those maximal values are not reached. All of the models abided by these criteria, and we do not report them because they are not directly relevant to the natural frequency of the tail, nor the preferred walking speed. For the interested reader, our MATLAB code (specifically, `Trex_tail_main_script.m`) produces tables of YM and pressures in horizontal equilibrium for all the models.

Below, I expand on these issues in turn:

1: Conflation of $\Delta\epsilon$ with ϵ

Between equations 5 and 6, the expression $R_s(\phi - \phi_e)$ is substituted for ϵ . Assuming that $(\phi - \phi_e) = \Delta\phi$ (but see point 2), the expression is still problematic since $R_s \Delta\phi = \Delta\epsilon$ by equation 2 and not ϵ .

Response

The reviewer is correct, and indeed the confusion arises in equation 2). We believe the situation would be resolved by redefining equation 2. Upon rereading, we noticed that ϕ_e was introduced but its nature was not explained. ϕ_e is now defined in figure 2 of the main manuscript, and we have further added ϕ_h . R_s , as we define it, is a simple kinematic relation that defines the mapping between the morphological reconstruction and the biomechanical model.

Changes to the ESM (ESM page 4, changes/additions marked in yellow)

As the tail flexes and extends, the amount of strain in the interspinous ligaments changes. We assume that the amount of ligament strain in horizontal equilibrium results in ligament forces that suffice to counteract the flexion torque of gravity (figure 2 A,B) (Hengst 2004). This implies that the posture of the tail in which the ligaments are at rest length is in slight extension (figure 2 C,D). The rotational springs in our model represent this posture-dependent strain and force, and therefore we have to define a mapping between segmental angles and the amount of strain. Attachment sites visible on the caudal vertebrae were used to reconstruct the CSAs of the interspinous ligaments (figure S7 A). This was done on the cranial and caudal side of each spinous process, and connecting these provided the extent of the ligaments (figure 1, figure S7 B). Ligament length was measured as the line between the area-centroids of two sequential vertebrae (see the next section on ligament moment arms for an elaboration on this choice). Strain was calculated as:

$$\epsilon = (L - L_0) / L_0 \quad (1)$$

Where ϵ is strain (expressed as a ratio), L is the ligament length and L_0 the resting length of the ligament. We assumed that ligament strain was 4% in horizontal posture for the baseline model. This is equivalent to resting lengths L_0 being 3.84% shorter than their ligament lengths in horizontal posture. Using this definition of L_0 , the tail was extended until all the ligament lengths were equal to their respective L_0 (leading to the “extended posture”, figure 2 C,D). We now introduce the rotational strain constant R_s (in strain/rad). For each segment, this is calculated using:

$$R_s = \epsilon / (\phi - \phi_e) \quad (2)$$

In equation 2, ϵ represents the instantaneous strain in the morphological model, while ϕ is the corresponding joint angle in the biomechanical model. The “e” in ϕ_e stands for “extended posture” (figure 2 C). Essentially, ϕ_e is the joint angle at which $L=L_0$, so this is conceptually similar to the rest angles (ϕ_r) of the biomechanical model as will be defined later. For each joint, R_s defines a kinematic coupling between relative joint angle and average ligament strain. This allows us to map the postural

dependent ligament strain (and thus torques) from the morphological reconstruction onto the rotational springs that cross the joints of our biomechanical model.

2: Ambiguity of φ

On line 336, φ is defined as “the angle that a segment makes with the horizontal axis”. This definition is unambiguous on its own. However, only relative angles between segments matter to the torque from a rotational spring (as the authors note on line 338). In particular, the relative angles should deviate from some “resting” relative angle, where there is no spring torque between segments.

Response

We agree with this statement.

The authors introduce a “rest angle” in equation 8. However, here the expression is confusing. If the interspinous ligaments “do not generate any... force” (Line 265), then the strain should be zero by equation 4. However, in equation 8 the strain is $\epsilon_{hor} = 0.04$.

Response

We agree that the expression (and explanation before it) is confusing. The rest angle is meant to be a “true” resting angle as the reviewer describes it below. We caught a further error: in horizontal equilibrium, $\phi = \pi$ (see figure S8), whereas in equation 8 & 9 the implication was that it is 0. We will correct the equations and expand the explanation in the ESM surrounding it.

Changes to the ESM (ESM page 6-7, changes marked in yellow):

To fit any of the sensitivity analysis models to RGM.792000, all that remains is to calculate the rest angles of each joint under the new assumptions. These are the relative joint angles where the rotational springs do not generate any torque. The baseline model simply uses ϕ_e as determined from the extended posture as defined in figure 2. All the other models used recalculated rest angles (ϕ_r , in rad), if the moment arms, rotational axes, or imposed strain in horizontal equilibrium was changed. This is equivalent to rewriting equation 2 to solve for ϕ_e . However, in equation 2, ϕ_e is a fixed parameter, because it is determined from the skeleton. We are now doing the inverse calculation: calculating ϕ_e as a free parameter, which is dependent on ϕ , ϵ , and the mapping determined by R_s . In horizontal equilibrium, $\phi = \pi$ (see figure 2). ϵ is replaced with ϵ_{hor} : this is the magnitude of strain we wish to impose during horizontal equilibrium (in the baseline model this is 0.04). This leads to the following equation to calculate the rest angles:

$$\phi_r = \pi - \epsilon_{hor}/R_s \quad (8)$$

ϵ_{hor} has the same dimension as strain, and R_s has the dimension strain/rad (or %/rad), and all the angles are in radians. They are both positive, which leads to a ϕ_r that is always smaller than π , which is the desired result (figure 2). ϕ_r is numerically identical to ϕ_e for the baseline, mass and length change models. In the other models, ϕ_r is different than ϕ_e , because of the scaling of R_s (equation 7) or by imposing a different magnitude of strain: In our baseline model, ϵ_{hor} was 0.04 (for 4% strain), and the effect of this choice was varied in the sensitivity analysis (i.e. ϵ_{hor} becomes 0.03 or 0.05).

Equation 8 allows us to investigate parameter changes on the tail, while only positioning it in the 2 postures visible in figure 2, instead of a different extended posture for each different model.

The moment arms have a scaling effect on R_s , thereby directly affecting the rest angles and thus natural frequencies of the model. The same is true for the alternative rotational axes in segments 3 and 4. ϕ_r can be used to calculate the spring constant of the rotational springs (c_s , in Nm/rad²):

$$c_s = T_{hor} / (\pi - \phi_r)^2 \quad (9)$$

T_{hor} (in Nm) is defined as the torque required to keep the segments horizontal, and was calculated using inverse dynamics. The spring constant c_s here is equal to $c \cdot R_s^2 \cdot L_0^2 \cdot r$ from the expression of T_{lig} provided in Eq. 6.

I believe the authors mean $\Delta\phi$ to be a proper “relative angle” from a true resting angle (which is itself a difference between absolute joint angles of adjacent segments). This would make sense given equation (2) and the use of $\Delta\phi$ in equation 6 (besides the issues raised in point 1). But then $\phi - \phi_e = \Delta\phi$, and what is ϕ_e ? Is it different from ϕ_r ? If the authors meant to use ϕ_r instead of ϕ_e , then there is still the issue that ϕ_r is not truly a rest angle (i.e. a relative angle at which there is no torque from the spring).

Response

We make the distinction between ϕ_e and ϕ_r because the former was determined from the skeleton (once) to find R_s , whereas the latter was recalculated for each model that was different from the baseline model (by scaling R_s). In the baseline, mass and length change models, ϕ_e and ϕ_r are numerically equivalent. In the other models (moment arms, rotational axes and strain change), ϕ_r was calculated by scaling R_s according to equation 7. We believe the above change resolves this issue.

3: R_s vs r/L_0

As discussed in points 1 and 3, there is ambiguity between the terms ϕ , $\Delta\phi$, and ϕ_r , as well as $\Delta\epsilon$ and ϵ . This leads to ambiguity about R_s , which I hope would be resolved if the authors address points 1 and 2 above.

Response

We have addressed points 1 and 2 and hope it indeed resolves the ambiguity mentioned.

There is yet another (perhaps more minor) issue with R_s : its dependence on the moment arm r is hidden in the expression 2. Since the cubic relationship between r and T_{lig} is a major point of the paper, it would make more sense to replace R_s with its expanded form using r , or at least define it in terms of r earlier.

In its current form ($R_s = \Delta\epsilon / \Delta\phi$), or in the form that seems more in line with the useage of R_s in equations 6 and 8 ($R_s = \epsilon / \Delta\phi$) I believe $R_s = r/L0$. Simply stating this relationship explicitly would greatly clarify the expressions in the paper.

Response

While R_s does not equal $r/L0$, R_s is proportional to r . We have now mentioned this explicitly after its introduction in the ESM (see our comment above).

We hope that the slightly altered method (leading to the lower results), combined with the elaborations in response to the previous comments, serve to remove the confusing nature regarding R_s .

--

Suppl Line 381: "...stiffness and damping of the tail would be mainly dependent on the ligaments."

I suspect the intervertebral discs would supply more damping. Can the authors comment on this, or even supply an estimate?

Response

The nature of the intervertebral connection between non-avian dinosaurs is still unclear. In general, the caudal vertebrae of RGM.792000 are deeply concave on the rostral side, and have a more shallow concavity on the caudal surface (amphicoelous to platycoelous). Going by (Wintrich et al. 2020), we could then infer a true (but perhaps primitive) intervertebral disc for dinosaurs, which would provide some damping. However, there is conflicting evidence in the form of tooth fragments scattered throughout the intervertebral space of a hadrosaurid (Rothschild et al. 2020), which the authors interpreted as evidence for an open space between the vertebrae (i.e. a synovial joint), which we would expect to have very little damping effects.

As such, a structure-based estimate of damping in the tail is difficult, if not impossible. For simplicity, we would like to limit our analysis to the passive structures that would dominate the tail's dynamic behaviour, which are the interspinous ligaments. Ligaments do not have much damping (Alexander 2006; Biewener 2008), and as such we simply assume that damping is low enough to not substantially lower the resonant frequency that would result (Ruina and Pratap 2019). The video (video S1) we have provided as ESM assumes a low damping ratio ($\beta = 0.16$) in the eigenmode with the lowest imaginary component (i.e. the lowest natural frequency).

Changes to the manuscript

We now provide a video of a lightly damped resonating tail, but we refrain from making any structure-based estimates of damping in the tail (see our response above).

Minor revisions / typos

Main Manuscript

--

As the authors have the code to solve 2D dynamics, I would like to know what the tail looks like when oscillating at its natural resonant frequency. Can the authors produce animations of the tail for the different parameter combinations? This would provide another means of assessing whether the biomechanical predictions appear sensible.

Response

We have provided a video (video S1) as ESM for a lightly damped tail resonating at its natural frequency (see above).

--

Line 167: "There are no extant analogues for tail-dependent bipedal locomotion..."

Lizards and some pangolins may be modern extant analogues. See Persons and Curry (2017 doi: 10.1016/j.jtbi.2017.02.032) and its relevant references.

Response

While certain squamates do indeed display (facultative) bipedalism, there are numerous issues with comparisons to non-avian dinosaurs. One important point (that Persons and Curry also mention) is that these squamates tend to only display these speeds when fleeing. It represents a top speed with high exertion, it does not at all represent an energetic minimum. Further issues are that these squamates have a sprawling gait, even when displaying bipedalism, and indeed are not obligate bipeds. Lastly, we find it difficult to justify their comparison given a mass difference of four to five orders of magnitude.

We were unaware that pangolins can display bipedal gait, and indeed were pleasantly surprised to learn this. As far as we can judge, however, pangolins actively lift their tail off the ground to walk bipedally, so it is not a valid comparison with tetanuran theropods, or indeed any non-avian dinosaur. Furthermore, they are not obligate bipeds.

Changes to the manuscript (addition marked in yellow)

"There are no extant analogues for tail-dependent **obligate** bipedal locomotion"

--

Lines 182-183: "Such trackway estimates are not strictly optimal, but on average they should tend towards OWS if the sample of trackways is large enough."

I suggest: "... they should tend towards preferred walking speed (and presumably OWS), if the sample..."

Response

Not directly in response to the reviewers, we have chosen to change almost all instances of "optimal" to "preferred" walking speed, unless explicitly referring to a study where energy expenditure was measured. This changes the title of our manuscript, but also covers the above suggestion.

--

Line 188: "This is true for dinosaurs, but even for"

Suggest: "This is true not only for dinosaurs, but even for..."

Response

Changed to "Not only is this true for dinosaurs, but even for.."

--

Supplemental material

Line 141: "...tail is in extension of flexion"

Suggest: "... tail is in extension or flexion"

Response

Thank you for the close attention, we have corrected this mistake

--

Line 303: "Stiffness of the ligaments were recalculated inverse dynamically..."

Suggest: "Stiffness of the ligaments were recalculated using inverse dynamics..."

Response

We have changed the manuscript accordingly

--

Line 348: "... Segdyn rearranges all the unknowns..."

I believe Segdyn is the authors' custom code, rather than some proprietary software. The authors should make this clear in the text.

Response

Segdyn is the algorithm that is extensively described in Casius, Bobbert, and van Soest (2004). We agree that this is unclear from the current wording, and have modified the manuscript to reflect this.

Changes to the ESM (ESM page 8):

This model was built in MATLAB using Segdyn, which is an algorithm described by Casius et al. (81) to derive the Newton-Euler equations of motion.

--

Figure S7a: the black ligament outline is hard to see. I would suggest a different (lighter) colour.

Response

We agree, and have adjusted the outline so it is more easily discernible, the outline is now dashed red and black.

References cited in our response

- Ahlborn, Boye K. and Robert W. Blake. 2002. "Walking and Running at Resonance." *Zoology* 105(2):165–74.
- Ahlborn, Boye K., Robert W. Blake, and William M. Megill. 2006. "Frequency Tuning in Animal Locomotion." *Zoology* 109(1):43–53.
- Alexander, R. McN. 2006. *Principles of Animal Locomotion*. Princeton NJ: Princeton University Press.
- Basu, Christopher, Alan M. Wilson, and John R. Hutchinson. 2019. "The Locomotor Kinematics and Ground Reaction Forces of Walking Giraffes." *Journal of Experimental Biology* 222(2).
- Biewener, A. A. 2008. "Tendons and Ligaments: Structure, Mechanical Behavior and Biological Function." *Collagen: Structure and Mechanics* 269–84.
- Casius, L. J. Richard, Maarten F. Bobbert, and Arthur J. Van Soest. 2004. "Forward Dynamics of Two-Dimensional Skeletal Models . A Newton-Euler Approach." 421–49.
- Christian, A., R. H. G. Müller, G. Christian, and H. Preuschoft. 1999. "Limb Swinging in Elephants and Giraffes and Implications for the Reconstruction of Limb Movements and Speed Estimates in Large Dinosaurs." *Fossil Record* 2(1):81–90.
- Dececchi, T. Alexander, Aleksandra M. Mloszewska, Thomas R. Holtz, Michael Bruce Habib, and Hans C. E. Larsson. 2020. "The Fast and the Frugal: Divergent Locomotory Strategies Drive Limb Lengthening in Theropod Dinosaurs" edited by A. Cuff. *PLOS ONE* 15(5):e0223698.
- Doke, J., J. M. Donelan, and A. D. Kuo. 2005. "Mechanics and Energetics of Swinging the Human Leg." *Journal of Experimental Biology* 208(3):439–45.
- Gellman, Karen S. and J. E. A. Bertram. 2002. "The Equine Nuchal Ligament 2: Passive Dynamic Energy Exchange in Locomotion." *Veterinary and Comparative Orthopaedics and Traumatology* 15(1):7–14.
- Hengst, Richard. 2004. "Gravity and the T. Rex Backbone." Pp. 69A--70A in *Journal of Vertebrate Paleontology*. Vol. 24.
- Holt, Kenneth G., Joseph Hamill, and Robert O. Andres. 1990. "The Force-Driven Harmonic Oscillator as a Model for Human Locomotion." *Human Movement Science* 9(1):55–68.
- Holt, Kenneth G., Joseph Hamill, and Robert O. Andres. 1991. "Predicting the Minimal Energy Costs of Human Walking." *Medicine and Science in Sports and Exercise* 23(4):491–98.
- Hoyt, Donald F. and Richard Taylor. 1981. "Gait and the Energetics of Locomotion in Horses." *Nature* 292:239–40.
- Huat, Ong Jor, Dhanjoo N. Ghista, Ng Kok Beng, and Tan Cher Chay John. 2004. "Optimal Stride Frequency Computed from the Double-Compound Pendulum of the Leg, and Verified

- Experimentally as the Preferred Stride Frequency of Jogging." *International Journal of Computer Applications in Technology* 21(1–2):46–51.
- Kane, Adam, Kevin Healy, Graeme D. Ruxton, and Andrew L. Jackson. 2016. "Body Size as a Driver of Scavenging in Theropod Dinosaurs." *American Naturalist* 187(6):706–16.
- Kilbourne, Brandon M. and Louwrens C. Hoffman. 2013. "Scale Effects between Body Size and Limb Design in Quadrupedal Mammals." *PLoS ONE* 8(11).
- Kilbourne, Brandon M. and Louwrens C. Hoffman. 2015. "Energetic Benefits and Adaptations in Mammalian Limbs : Scale Effects." (June 2015).
- Lee, Leng Feng and Venkat N. Krovi. 2008. "Musculoskeletal Simulation-Based Parametric Study of Optimal Gait Frequency in Biped Locomotion." *Proceedings of the 2nd Biennial IEEE/RAS-EMBS International Conference on Biomedical Robotics and Biomechatronics, BioRob 2008* (November 2008):354–59.
- Loscher, David M., Fiete Meyer, Kerstin Kracht, and John A. Nyakatura. 2016. "Timing of Head Movements Is Consistent with Energy Minimization in Walking Ungulates." *Proceedings of the Royal Society B: Biological Sciences* 283(1843).
- Manning, Phillip L., Christopher Ott, and Peter L. Falkingham. 2008. "A Probable Tyrannosaurid Track from the Hell Creek Formation (Upper Cretaceous), Montana, United States." *Palaios* 23(10):645–47.
- McCrea, Richard T., Lisa G. Buckley, James O. Farlow, Martin G. Lockley, Philip J. Currie, Neffra A. Matthews, and S. George Pemberton. 2014. "A 'Terror of Tyrannosaurs': The First Trackways of Tyrannosaurids and Evidence of Gregariousness and Pathology in Tyrannosauridae." *PLoS ONE* 9(7).
- Myers, Marcella J. and Karen Steudel. 1997. "Morphological Conservation of Limb Natural Pendular Period in the Domestic Dog (*Canis Familiaris*): Implications for Locomotor Energetics." *Journal of Morphology* 234(2):183–96.
- Raichlen, D. A. 2004. "Convergence of Forelimb and Hindlimb Natural Pendular Period in Baboons (*Papio Cynocephalus*) and Its Implication for the Evolution of Primate Quadrupedalism." *Journal of Human Evolution* 46(6):719–38.
- Romilio, Anthony and Steven W. Salisbury. 2011. "A Reassessment of Large Theropod Dinosaur Tracks from the Mid-Cretaceous (Late Albian-Cenomanian) Winton Formation of Lark Quarry, Central-Western Queensland, Australia: A Case for Mistaken Identity." *Cretaceous Research* 32(2):135–42.
- Rothschild, Bruce M., Robert A. Depalma, David A. Burnham, and Larry Martin. 2020. "Anatomy of a Dinosaur—Clarification of Vertebrae in Vertebrate Anatomy." *Journal of Veterinary Medicine Series C: Anatomia Histologia Embryologia* 49(4):571–74.
- Ruina, Andy and Rudra Pratap. 2019. *Mechanics Toolset, Statics and Dynamics*.
- Russell, Daniel M. and Dylan T. Apatoczky. 2016. "Gait & Posture Walking at the Preferred Stride Frequency Minimizes Muscle Activity." *Gait & Posture* 45:181–86.
- Schroeder, Ryan T., John E. A. Bertram, Van Son Nguyen, Van Vinh Hac, and James L. Croft. 2019. "Load Carrying with Flexible Bamboo Poles: Optimization of a Coupled Oscillator System." *Journal of Experimental Biology* 22(23).
- Sellers, William Irvin, Lee Margetts, Rodolfo Aníbal Coria, and Phillip Lars Manning. 2013. "March of the Titans: The Locomotor Capabilities of Sauropod Dinosaurs" edited by D. Carrier. *PLoS ONE*

8(10):e78733.

Thulborn, Richard. 1984. "Dinosaur Trackways in the Winton Formation (Mid-Cretaceous) of Queensland." *Memoirs of the Queensland Museum*. 21(2):413–517.

Wagenaar, R. C. and R. E. A. Van Emmerik. 2000. "Resonant Frequencies of Arms and Legs Identify Different Walking Patterns." *Journal of Biomechanics* 33(7):853–61.

Wintrich, Tanja, Martin Scaal, Christine Böhmer, Rico Schellhorn, Ilja Kogan, Aaron van der Reest, and P. Martin Sander. 2020. "Palaeontological Evidence Reveals Convergent Evolution of Intervertebral Joint Types in Amniotes." *Scientific Reports* 10(1):1–14.

Appendix B

Response to reviewer 2

We once again extend our gratitude to the reviewer for the kind words regarding our manuscript. We were pleased to read that our revised manuscript resolved most of the concerns, and appreciate the evident care with which the reviewer has re-evaluated our model.

As a preface to our response regarding the reviewer's main point, we now suspect that the misunderstanding was caused by the fact that we did not make it completely clear in our supplementary material which equations apply to a single ligament spanning an actual intervertebral joint, and which equations apply to our reductionist model in which several vertebrae together were modelled as a single rigid body (in the spirit of lowering the barrier of adoption).

We have split up the main concern of the reviewer (marked in red), and our response, into sub-statements. Because most of our changes are to the ESM, we have included it in its entirety (with the changes marked in yellow), at the end of the tracked changes version of the main manuscript.

The authors define a mapping from joint angle to strain as $\epsilon = R_s (\phi - \phi_e)$, where R_s is a constant. They then say that "The equation for R_s will provide an adequate approximation as long as the relative joint angles do not deviate far from horizontal posture."

However, this mapping only appears appropriate when joint angles are close to extended posture. It assumes that the change in strain from extended posture (where strain is zero) to some ϕ is equal to change in strain around ϕ . But there is no reason to believe this a priori unless ϕ is close to ϕ_e . Moreover, moment arm should modify the relationship between strain and joint angle, even close to extended posture- so it seems odd that the moment arm does not appear in the definition of R_s .

While the main argument of the reviewer is below, we mention here that for the purposes of a small angle approximations, ϕ indeed remains "close to" ϕ_e in our model: during the (damped) simulation of our baseline model, the differences between ϕ and ϕ_e do not exceed 0.3 rad, corresponding to a discrepancy of less than 4.5 %, and the differences between ϕ and ϕ_h are of course lower.

Using Taylor expansion, and assuming moment arm remains constant, the relationship between strain (ϵ) and joint angle near horizontal posture (ϕ_h) should be $\epsilon(\phi) \approx \epsilon(\phi_h) + r/L_0 (\phi - \phi_h)$. While this comes from a quick derivation, and I could have made an error, we would at least expect ϕ_h (or π) to appear in the definition of R_s , but it doesn't.

The above Taylor expansion is valid for strain in a single ligament, where the angle ϕ represents the relative angle between the vertebrae that the ligament spans. It can be derived using the insight that ligament length (L) is dependent on joint angle (ϕ) between two vertebrae, moment arm (r), and resting length (L_0 , where $\phi = \phi_e$). When $\phi - \phi_e$ is small, or when r is constant, this expression is:

$$L(\phi) = L_0 + r (\phi - \phi_e)$$

This shows how change in ligament length depends on the moment arm and change in joint angle, which can also be derived using the principle of virtual work (An, Takahashi, Harrigan, & Chao, 1984). The above expression for L can be combined with ESM equation (1) to acquire a general expression for $\epsilon(\phi)$:

$$\epsilon(\phi) \approx r/L_0 (\phi - \phi_e)$$

If this expression is compared to ESM equation (2), it is clear why the reviewer expects R_s to be equivalent to r/L_0 , and indeed this is true when a single ligament spanning an actual intervertebral joint is considered. However, it should be noted that in our reductionist model, ϕ does not represent an actual joint angle between two individual vertebrae, but instead represents the angle in the hinge joint between segments, so it represents a combined rotation at several intervertebral joints. We expect that this is where the confusion arises: R_s is only equal to r/L_0 if the biomechanical model would incorporate each vertebra and ligament individually. Constructing a model of the tail which incorporates each individual vertebra greatly increases the barrier to adoption, and is counter-productive to the goal of a reductionist method. However, in dividing the tail into larger functional segments, the moment arm's effect on strain of each individual ligament is lost.

We sought to incorporate this effect of the moment arms, without repositioning the entire skeleton for each individual model of the sensitivity analysis (because this greatly increases the barrier to adoption: rearticulating a skeleton 8 times for each specimen is time consuming, which would greatly reduce the number of specimens one could incorporate into the analysis). As such, we introduce R_s , which we use to map the strain dependency of the ligaments onto the more abstract rotational spring. This is also why we make the distinction between ϕ_e , the joint angles acquired by repositioning the skeleton in extended posture, and ϕ_r (rest angles), the joint angles at which the rotational springs generate no torque. For the baseline, mass & length change models, $\phi_e = \phi_r$ because R_s is unchanged. R_s comes into play with the moment arms & distal axes models, where we use the moment arm ratios to scale R_s (circumventing the need to reposition the skeleton using the different moment arms or axes). L_0 is unchanged in these models, because we are isolating the effect of the moment arms on strain. In other words: in the maximal or minimal moment arms models, the ligament is still at $1.04 L_0$ in horizontal equilibrium. The effects of varying ϵ_{hor} are investigated in the alternate strain models using ESM equation (8), while keeping R_s unchanged.

It is important to point out that R_s in the reduced model is not geometrically the same as r/L_0 for the individual vertebrae. R_s is used as a means to incorporate relative moment arm variation throughout the skeleton in a reduced model where these moment arms are no longer present. Upon reevaluation, we understand the reviewer's suggestion that R_s should also scale with L_0^{-1} . This would affect the strain-change models, because the moment arms/rotational axes models assume constant L_0 . However, it would make no difference to our result (within reporting accuracy). For instance, in the increased strain model, we set ϵ_{hor} to 0.05 (up from 0.04, so a 25% increase). ESM equations (8) and (9) imply that natural frequency $\propto \epsilon_{hor}^{-0.5}$. Therefore, increased strain decreases baseline natural frequency:

$$0.66 * 1.25^{-0.5} = 0.59.$$

We could scale R_s according to the change in L_0 this implies, which amounts to a scale factor of $1.04/1.05 = 1.0096$. This still leads to:

$$0.66 * (1.25 * 1.04/1.05)^{-0.5} = 0.59.$$

We confirmed the above prediction by editing the code to see if it affected the results (within reporting accuracy), which it did not. Incorporation of this extra scale factor has the undesirable effect of narrowing the bounds of the sensitivity analysis (at larger changes in ϵ_{hor}), so we prefer not to incorporate it in the spirit of maintaining wide bounds.

Changes to the manuscript (combined document, lines 751-802, 902-907)

Given the nature of this misunderstanding, we believe that it is prudent to highlight the conceptual similarity, but practical difference, between R_s and r/L_0 . We therefore have split up ESM equation 2

into two variants, and rewritten the original equation to ease comparison. We also now make a distinction between ϕ (joint angle between two vertebrae) and φ (joint angle in the reduced linked chain model):

$$\varepsilon = r/L_0 \cdot (\phi - \phi_0) \quad (2a)$$

$$\varepsilon_{hor} = R_S \cdot (\varphi_h - \varphi_e) \quad (2b)$$

We now specifically mention that while this implies that $R_S \propto r/L_0$, for the moment arms/rotational axes models this makes no difference because L_0 was kept constant. We also explicitly mention that for the strain change models we opted not to include this proportionality, because it does not affect our result within reporting accuracy, and its exclusion leads to wider bounds for the sensitivity analysis.

Combined document, lines 835-853

Because equation 6 was based on substituting equation 2, we have split equation 6 into two parallel equations as well (again signifying the abstraction step between a single ligament and a joint spring).

Can the authors either

1. Explain what I'm misunderstanding, and why their mapping is correct close to ϕ_h or

Assuming the moment arm stays constant, our mapping would be accurate for all angles, not just ϕ_h , using the relation between joint angle, moment arm, and ligament length (An et al., 1984). In reality, moment arms likely vary with joint angle, so the relation is only accurate for angles near the angle at which the moment arm was determined. In our case, this was chosen to be ϕ_h , on the ground that this is the posture at which we intend to estimate natural frequency. As stated above, deviations of ϕ from ϕ_h are not large, so the moment arms do not change substantially.

Changes to the manuscript (combined document, lines 756-758)

We now explicitly state, before introduction equation 2a in the ESM, that we assume moment arms to remain constant due to the small angle deviations, and that they were determined in horizontal posture.

2. Use a more carefully derived mapping appropriate for angles close to ϕ_h ?

The authors assume constant moment arms through the motion. This is acceptable, and should be clearly stated. In principle R_S should have a clearly defined basis in the geometry of their model, and if moment arms are constant I suspect that it is relatively simple.

As indicated above, the assumption that moment arms are constant is now mentioned in the ESM.

The authors have added simply stated that R_S has "the property" of being proportional to moment arm length, but it is not immediately clear why. I think this relationship to model geometry should be explicitly stated. As it stands, the physical basis- and appropriateness- of R_S is not clear.

The most important issue, namely the abstract nature of R_S , has been addressed above. Essentially: it does not have a clear geometric basis, because the geometric relationships between moment arms and strain are lost when the complex morphology of the tail is reduced to 5 rigid bodies. Because we recognize that the moment arms play an important role in the rotational stiffness of the tail, we sought to incorporate them without resorting to repositioning the skeleton for each different moment arm. This led to the introduction of R_S , which should be interpreted as a way to investigate

how sensitive a parametrization of the tail is to choices in moment arms or rotational axes, while avoiding unnecessary complexity.

Minor changes

The authors have removed uncertainty ranges in this version. I appreciate the authors' hesitation here. They contend that their lowest and highest estimates for natural frequency are unrealistic, since dorsal and ventral axes of the tail would not be static during locomotion. Even so, I contend that it's better to be conservative with the estimate and state the lowest and highest frequencies and speeds found during the sensitivity analysis (e.g. "... walking speed (1.28 m/s, range 0.8-1.64)"). This avoids the perception, for the surprising number of readers who only get as far as the abstract, that these are extremely precise estimates.

Response/Changes

We had removed the bounds because we interpret the sensitivity analysis as a guide for future research directions. We consider the source of the uncertainty to be more interesting than the actual ranges they imply, especially since we selected overly large bounds. However, we understand the position of the reviewer, and now report natural frequency as (0.66 s^{-1} , range 0.41 – 0.84) and walking speed as (1.28 m s^{-1} , range 0.80 – 1.64) in the abstract. We opted to include the following statement in our discussion (lines 231-233):

Thus, the bounds in supplementary table S2 are unrealistically wide, due to the assumptions underlying this table. Yet, to acknowledge the uncertainties in dinosaur gait reconstruction, we used these as the reported range in our result.

Lines 132-134: "However, since ligaments (like tendons) display relatively little energy loss during work-loop experiments [41,46], little damping is present. Therefore, we will use the undamped natural frequency..."

In the response document, the authors instead mention "... a structure-based estimate of damping in the tail is difficult, if not impossible."

To me, the latter is a more valid reason to ignore damping, as I question that the ligaments would be the dominant source of damping. More likely, all the "squishy stuff" in the tail will provide more damping.

I recommend justifying ignoring damping in the main manuscript with something more similar to the second statement.

Response

Because energy storage in the ligaments plays a prominent role in our reasoning, we prefer to keep the references regarding energy losses in ligaments. However, we now follow it with a statement in line with the reviewer's request

Changes to the manuscript (lines 135-139).

However, damping of dorsoventral oscillations would be dependent on more than just the ligaments, and a structure-based estimate of damping would be difficult, if not impossible. Therefore, we will use the undamped...

Lines 41-43: “Both fore- and hindlimb natural frequencies scale similarly with several morphological parameters within several mammalian groups [16,17], suggesting that resonance is beneficial at any body size.”

The sentence is vague, and scaling similarity does not automatically imply “benefit”. I recommend removing the sentence. The authors make their case for estimating PWS at resonance with the other surrounding sentences.

Response & changes

We have removed the entire quoted sentence, and corresponding two references, from the manuscript.

Caption, Table 1: “The moment arms and alternative axes models are not mathematically predictable, and these cannot be combined with each other.”

This is confusing and not necessary in the caption. I recommend removing the sentence.

Changes

We have replaced the quoted sentence with: “The alternative axes models do not represent an entire step cycle, which is why their combinations with the moment arms models should be interpreted with caution.”

Other reviewer comments on the supplementary text

We have now defined cross-sectional area before using the acronym, and changed the wording when describing the “percentages of percentages” as per the reviewer’s suggestion.

Other Changes

We have opted to add the following sentence to our abstract, to place our results into context:

“The walking speeds found here are lower than earlier estimations for large theropods, but agree quite closely with preferred walking speeds of a diverse group of extant animals.”

We have also included a table of contents to our ESM, to improve the clarity of the document.

References cited in this response:

An, K. N., Takahashi, K., Harrigan, T. P., & Chao, E. Y. (1984). Determination of muscle orientations and moment arms. *Journal of Biomechanical Engineering*, 106(3), 280–282.
<https://doi.org/10.1115/1.3138494>